# The enhancement of droplet collision by electric charges and atmospheric electric fields

Shian Guo, Huiwen Xue

School of Physics, Peking University, Beijing, China

*Correspondence to*: Huiwen Xue (hxue@pku.edu.cn)

**Abstract.**

The effects of electric charges and fields on droplet collision-coalescence and the evolution of cloud droplet size distribution are studied numerically. Collision efficiencies for droplet pairs with radii from 2 to 1024 μm and charges from -32 $r^2$ to +32 $r^2$ (in unit of elementary charge, droplet radius $r$ in unit of μm) in different strengths of downward electric fields (0, 200 and

400 V cm$^{-1}$) are computed by solving the equations of motion for the droplets. It is seen that collision efficiency is increased by electric charges and fields, especially for pairs of small droplets. These can be considered as electrostatic effects.

The evolution of cloud droplet size distribution with the electrostatic effects is simulated using the stochastic collection equation. Results show that the electrostatic effect is not notable for clouds with the initial mean droplet radius $\bar{r}$ =15 μm or larger. For clouds with the initial $\bar{r}$ = 9 μm, the electric charge without field could evidently accelerate raindrop formation

compared to the uncharged condition, and the existence of electric fields further accelerates it. For clouds with the initial $\bar{r}$ = 6.5 μm, it is difficult for gravitational collision to occur, and the electric field could significantly enhance the collision process. Results of this study indicate that electrostatic effects can accelerate raindrop formation in natural conditions, particularly for polluted clouds. It is seen that the aerosol effect on the suppression of raindrop formation is significant in polluted clouds, when comparing the three cases with $\bar{r}$ = 15, 9, and 6.5 μm. However, the electrostatic effects can accelerate raindrop formation

in polluted clouds and mitigate the aerosol effect to some extent.

## 1 Introduction

Clouds are usually electrified (Pruppacher and Klett 1997). For thunderstorms, several theories of electrification have been

proposed in the past decades. The proposed theories assume that the electrification involves the collision of graupel or hailstones with ice crystals or supercooled cloud droplets, based on radar observational result that the onset of strong electrification follows the formation of graupel or hailstones within the cloud (Wallace and Hobbs, 2006). However, the exact conditions and mechanisms are still under debate. One charging process could be due to the thermoelectric effect between the relatively warm, rimed graupel or hailstones and the relatively cold ice crystals or supercooled cloud droplets. Another charging

process could be due to the polarization of particles by the downward atmospheric electric field. The thunderstorm electrification can increase the electric fields to several thousand V cm$^{-1}$, while the magnitude of electric fields in fair weather air is only about 1 V cm$^{-1}$ (Pruppacher and Klett 1997). Droplet charges can reach $|q| \approx 42r^2$ in unit of elementary charge

in thunderstorms, with the droplet radius $r$ in unit of μm according to observations (Takahashi, 1973). For cumuli clouds, previous studies show smaller charge amount.

Liquid stratified clouds do not have such strong charge generation as in the thunderstorms. But charging of droplets can indeed occur at the upper and lower cloud boundaries as the fair weather current passes through the clouds (Harrison et al. 2015, Baumgaertner et al. 2014). The global fair weather current and the electric field are in the downward direction. Given the electric potential of 250 kV for the ionosphere, the exact value of fair weather current density over a location depends on the electric resistance of the atmospheric column, but its typical value is about $2 \times 10^{-12}$ A m$^{-2}$ (Baumgaertner et al. 2014). The

fair weather electric field is typically about 1 V cm$^{-1}$ in the cloud-free air, but is usually much stronger inside stratus clouds, because the cloudy air has a lower electrical conductivity than the cloud-free air. There is a conductivity transition at cloud boundaries. Therefore, the cloud top is positively charged and the cloud base is negatively charged. Based on the in situ measurements of charge density in liquid stratified cloud, and assuming that the cloud has a droplet number concentration on the order of 100 cm$^{-3}$, it is estimated that the mean charge per droplet is +5e (ranging from +1e to +8e) at cloud top, and -6e

(ranging from -1e to -16e) at cloud base (Harroson et al. 2015). According to Tsutomu Takahashi (1973) and Khain (1997), the mean absolute charge of droplets in warm clouds is around $|q| \approx 6.6\ r^{1.3}$ (e, μm). For a droplet with radii of 10 μm, it is about 131 e.

In general, charging of droplets can lead to the following effects on warm cloud microphysics. Firstly, for charged haze droplets, the charges can lower the saturation vapor pressure over the droplets and enhance the cloud droplet activation

(Harrison and Carslaw, 2003, Harrison et al. 2015). Secondly, the electrostatic induction effect between charged droplets can lead to strong attraction at very small distance (Davis, 1964) and higher collision-coalescence efficiencies (Beard et al. 2002). But Harrison et al. (2015) showed that charging is more likely to affect collision processes than activation, for small droplets.

The electrostatic induction effect can be explained by regarding the charged cloud droplets as spherical conductors. The electrostatic force between two conductors is different from the well-known Coulomb force between two point charges. When

the distance between a pair of charged droplets approaches infinity, the electrostatic force converges to Coulomb force between two point charges. But when the distance of surfaces of two droplets is small (e.g. much smaller than their radii), their interaction shows extremely strong attraction. Even when the pair of droplets carry the same sign of charges, the electrostatic force can still change from repulsion to attraction at small distance. Although there is no explicit analytical expression such as Coulomb force for the electrostatic interaction between two charged droplets, a model with high accuracy has been developed

(Davis 1964) for the interaction of charged droplets in a uniform electric field. Many different approximate methods are also proposed for the convenience of computation in cloud physics (e.g. Khain et al., 2004).

Based on this induction concept, electrostatic effects on droplet collision-coalescence process have been studied in the past decades. A few experiments show that electric charges and fields can enhance coalescence between droplets. Beard et. al. (2002) conducted experiments in cloud chambers and showed that even minimal electric charge can significantly increase the

probability of coalescence when the two droplets collide. Eow et. al., (2001) examined several different electrostatic effects in water-in-oil emulsion, indicating that electric field can enhance coalescence by several mechanisms such as film drainage.

More numerical researches indicate that charges and fields can increase droplet collision efficiencies because of the electrostatic forces. Schlamp et al. (1976) used the model of Davis (1964) to study the effect of electric charges and atmospheric electric fields on collision efficiencies. They demonstrated that the collision efficiencies between small droplets (about 1~10 μm) are enhanced by an order of magnitude in thunderstorm condition, while collision between large droplets is hardly affected. Harrison et al. (2015) investigate the electrostatic effects in weakly electrified liquid clouds rather than thunderstorms. They calculated collision efficiencies between droplets with radii less than 20 μm and charge less than 50 e, by the equations of motion in Klimin (1994). Their results indicate that electric charges at the upper and lower boundaries of warm stratified clouds are sufficient to enhance collisions, and the enhancement is especially significant for small droplets. Moreover, they proposed that solar influences may change the fair weather current and droplet collision process, a possible pathway for affecting the climate system. Tinsley (2006) and Zhou (2009) also studied the collision efficiencies between charged droplets and aerosol particles in weakly electrified clouds, by treating the particles as conducting spheres. They considered many aerosol effects such as thermophoretic forces, diffusophoretic forces and Brownian diffusion.

As for the electrostatic effect on the evolution of droplet size distributon and the cloud system, few researches have been conducted. Focusing on weather modification, Khain et al. (2004) showed that a small fraction of highly charged particles could trigger the collision process, and thus accelerate raindrop formation in warm clouds or lead to fog elimination significantly. In their study, the electrostatic force between the droplet pair is represented by an approximate formula. The charge limit is set to the air-breakdown limit. The Stokes Flow is adopted to represent the hydrodynamic interaction, for deriving the trajectories of a pair of droplets. Harrison et. al. (2015) calculated droplet collision efficiencies affected by electric charges in warm clouds. When simulating the evolution of droplet size distribution in their study, the enhanced collision efficiencies are not used. Instead, the collection cross sections are multiplied by a factor of no more than 120% to approximately represent the electric enhancement of collision efficiency. The roles of electric charges and fields on precipitation acceleration still needs to be studied.

The increased aerosol loading by anthropogenic activities can lead to an increase in cloud droplet number concentration, a reduction in droplet size, and therefore an increase in cloud albedo (Twomey 1974). This imposes a cooling effect on climate. It is further recognized that the aerosol-induced reduction in droplet size can slow down droplet collision-coalescence and cause precipitation suppression. This leads to increased cloud fraction and liquid water amount, and imposes an additional cooling effect on climate (Albrecht 1989). As the charging of cloud droplets can enhance droplet collision-coalescence, especially for small droplets, it is worth studying to what extent the electrostatic effect can mitigate the aerosol effect on the evolution of droplet size distribution and precipitation formation.

This study investigates the effect of electric charges and fields on droplet collision efficiency and the evolution of the droplet size distribution. The amount of charges is set as the condiion in warm clouds, and the electric fields are set as the early stage of thunderstorms. The more accurate method for calculating the electric forces is adopted (Davis, 1964). Correction of flow field for large Reynolds numbers are also considered. Section 2 describes the theory of droplet collision-coalescence and the stochastic collection equation. Section 3 presents the equations of motion for charged droplets in an electric field. The method

for obtaining the terminal velocities and collision efficiencies for charged droplets are also presented. Section 4 describes the model setup for solving the stochastic collection equation. Different initial droplet size distributions and different electric conditions are considered. Section 5 shows the numerical results of electrostatic effects on collision efficiency, and on the evolution of droplet size distribution. We intend to find out to what extent the electric charges and fields as in the observed atmospheric conditions can accelerate warm rain process, and how sensitive these electrostatic effects are to aerosol-induced changes of droplet sizes.

## 2 Stochastic Collection Equation

The evolution of droplet size distribution due to collision-coalescence is described by the stochastic collection equation (SCE), which was first proposed by Telford (1955), and is expressed as (Lamb and Verlinde, 2011, p.442)

$$\frac{\partial n(m,t)}{\partial t} = \int_0^{m/2} K(m_x, m - m_x) \cdot n(m_x, t) n(m - m_x, t) \mathrm{d}m_x - n(m,t) \int_0^{\infty} K(m_x, m) \cdot n(m_x, t) \mathrm{d}m_x \tag{1}$$

where $n(m,t)$ is the distribution of droplet number concentration over droplet mass at time $t$, and $K$ is the collection kernel between the two classes of droplets. For example, the collection kernel $K(m_x, m - m_x)$ describes the rate that droplets of mass $m_x$ and mass $m - m_x$ collide to form new droplets of mass $m$. The first term on the right side of Eq. (1) describes the formation of droplets of mass $m$ through collision of smaller droplets, and the second term describes the loss of droplets of mass $m$ through collision with other droplets.

The collection kernel between droplets with mass $m_1$ and mass $m_2$ can be written as

$$K(m_1, m_2) = |V_1 - V_2| \cdot \pi (r_1 + r_2)^2 \cdot E(m_1, m_2) \cdot \varepsilon(m_1, m_2) \tag{2}$$

where subscripts 1 and 2 denote droplet 1 and droplet 2, respectively, $V$ is the terminal velocity of the droplet, and $r$ is droplet radius. Terminal velocity is the steady-state velocity of the droplet relative to the flow, when no other droplets are present and therefore there is no interaction from other droplets. Suppose droplet 1 is the collector and droplet 2 is the collected droplet, the term $|V_1 - V_2| \cdot \pi (r_1 + r_2)^2$ represents the geometric volume swept by droplet 1 in unit time. Collision efficiency $E(m_1, m_2)$ and coalescence efficiency $\varepsilon(m_1, m_2)$ are introduced to the kernel because not all the droplets in this volume will have collision-coalesce with the collector.

For a pair of droplets, each of them induces a flow field that interacts with the other. As the collector falls and sweeps the air volume, the droplets in the volume tend to follow the streamlines of the flow field induced by the collector. Droplets collide with the collector only when they have enough inertia and cross the streamlines. Collision efficiency is then defined as the ratio of the actual collisions over all possible collisions in the swept volume. It can be much smaller than 1.0 when the sizes of the two droplets are significantly different. The physical meaning of collision efficiency is shown in Fig. 1 for a droplet pair. The collector droplet falls faster and induces a flow field to interact with the small droplet. The small droplet follows a grazing trajectory (as shown in the figure) when the centers of the two droplets have an initial horizontal distance $r_c$, which can be

regarded as the threshold horizontal distance. Collision occurs only when the two droplets have an initial horizontal distance smaller than $r_c$. For any droplet pair, $r_c$ depends on the sizes of the two droplets. Then the collision cross section is $S_c = \pi r_c^2$, and collision efficiency is $E = r_c^2/(r_1 + r_2)^2$. There are many previous studies on collision efficiency, by both numerical simulations and chamber experiments (Pruppacher and Klett 1997)

Two droplets may not coalesce even when they collide with each other. Observations show that the droplet pair can rebound in some cases, because of an air film temporally trapped between the two surfaces. Especially for droplets with radii both larger than 100 μm, the coalescence efficiency is remarkably less than 1.0. Beard and Ochs (1984) provides a formula of coalescence efficiency for a certain range of droplet radii. Basically, coalescence efficiency is a function of the sizes of the two droplets in their formula.

In this study, electric charges and external electric fields are taken into consideration for droplet collision-coalescence process. Droplet distribution function has two variables: droplet mass $m$ (or radius $r$) and electric charge $q$. The SCE can be expressed as

$$\frac{\partial n(m,q,t)}{\partial t} = \int_0^{m/2} [\int_{-\infty}^{+\infty} K\,(m_x, q_x; m - m_x, q - q_x) \cdot n(m_x, q_x, t) n(m - m_x, q - q_x, t)\,\mathrm{d}q_x]\mathrm{d}m_x$$

$$- n(m,q,t) \int_0^{\infty} [\int_{-\infty}^{+\infty} K\,(m_x, q_x; m, q) \cdot n(m_x, q_x, t)\mathrm{d}q_x]\,\mathrm{d}m_x$$

(3)

where $n(m,q,t)$ is the distribution of droplet number concentration over mass and charge, and $K$ is the collection kernel of the two classes of droplets. The collection kernel $K\,(m_x, q_x; m - m_x, q - q_x)$ represents the rate that droplets of mass $m_x$ and charge $q_x$ collide with droplets of mass $m - m_x$ and charge $q - q_x$ to form new droplets of mass $m$ and charge $q$.

The collection kernel for charged droplets in an external electric field has the same form as Eq. (2). However, terminal velocity, collision efficiency, and coalescence efficiency in the kernel my all be affected by the electric charge and field. We consider these as electrostatic effects. In a vertical electric field, the terminal velocity of a charged droplet may be increased or decreased, depending on the charge sign and the direction of the field. The threshold horizontal distance $r_c$, the collision cross section, and the collision efficiency of a droplet pair may be changed because the electric charge and field can make the droplets to cross the streamlines more easily under some circumstances. Therefore, terminal velocity, collision efficiency, and coalescence efficiency not only depend on the sizes of the two droplets, but may also depend on the electric charge and the external electric field.

As will be seen in this study, the electrostatic effects on collision efficiency is much stronger than on terminal velocity. Therefore, the electrostatic effect on terminal velocity is presented in Section 6 as discussion, and we focus on the electrostatic effects on collision efficiency in this paper. The method for obtaining droplet terminal velocity and collision efficiency with the electrostatic effects will be presented in section 3. The electrostatic effect on coalescence efficiency is not considered here. The coalescence efficiency used in this study is the same as that for uncharged droplets, based on the results of Beard and Ochs (1984). In their study, coalescence efficiency is a function of $r_1$ and $r_2$, and is valid for $1 < r_2 < 30$ μm

and $50 < r_1 < 500$ µm. In this study, however, the range of $r_1$ and $r_2$ is much wider, from 2 to 1024 µm. The formula of coalescence efficiency in Beard and Ochs (1984) is extrapolated for the droplet size range here. Coalescence efficiency is set to be 1 if the extrapolated value is higher than 1, and set to be 0.3 if the extrapolated value is smaller than 0.3.

## 3  Method for calculating terminal velocity and collision efficiency with electrostatic effects

### 3.1  Equations of motion for charged droplets

In order to get the terminal velocity and collision efficiency, the equations of motion need to be used. Droplet motion depends on the following three forces: gravity force, the flow drag force, and the electrostatic force due to droplet charge and the external electric field. The equations of motion for a pair of droplets are shown below,

$$\frac{d\boldsymbol{v_1}}{dt} = \boldsymbol{g} - C\frac{6\pi r_1 \eta}{m_1}(\boldsymbol{v_1} - \boldsymbol{u_2}) + \frac{\boldsymbol{F_{e1}}}{m_1} \tag{4a}$$

$$\frac{d\boldsymbol{v_2}}{dt} = \boldsymbol{g} - C\frac{6\pi r_2 \eta}{m_2}(\boldsymbol{v_2} - \boldsymbol{u_1}) + \frac{\boldsymbol{F_{e2}}}{m_2} \tag{4b}$$

where subscripts 1 and 2 denotes droplet 1 and droplet 2, respectively, $\boldsymbol{g}$ is the gravitational acceleration, $\boldsymbol{v}$ is the velocity of the droplet relative to the flow if there are no other droplets present, $\boldsymbol{u}$ is the flow velocity field induced by the droplet, $\eta$ is the viscosity of air, $C$ is the drag coefficient, which is a function of Reynolds number, $r$ is droplet radius, $m$ is droplet mass, with $m = 4\pi r^3 \rho/3$, and $\boldsymbol{F_e}$ is the electrostatic force. We set air temperature $T = 283$ K and pressure $p = 900$ hPa in this study for the calculation of air viscosity.

### 3.2  The drag force term

The flow drag force is described by the second term on the right side of Eq. (4), which assumes a simple hydrodynamic interaction of the two droplets. That is, each droplet moves in the flow field induced by the other one moving alone, and it is 190    called "superposition method" in cloud physics. This method has been successfully used in many researches of the calculation of collision efficiencies (Pruppacher and Klett, 1997). The superposition method can also ensure that the stream function satisfies the no-slip boundary condition (i.e., Wang et al. 2005) To calculate the flow drag force, the induced flow field $\boldsymbol{u}$ is required. The method for obtaining the induced flow field $\boldsymbol{u}$ is discussed below.

Considering a rigid sphere moving in a viscous fluid with a velocity $U$ relative to the flow, the steam function depends on 195    Reynolds number, $N_{Re} = \frac{2rv\rho}{\mu}$, where $\rho$ is the density of the air, and $\mu$ is the dynamic viscosity of the air. It is known that when Reynolds number is small, the flow is considered as Stokes flow and the stream function can be expressed as

$$\psi_s = U\left(\frac{1}{4\tilde{R}} - \frac{3\tilde{R}}{4}\right)\sin^2\theta_0 \tag{5}$$

where $\tilde{R} = R/r$ is the normalized distance ($R$ is the distance from the sphere centre, and $r$ is the droplet radius), $\theta_0$ is the angle between the droplet velocity and vector $\boldsymbol{R}$ pointing from the sphere centre. $U$ is droplet velocity relative to the flow, i.e., $U_1 = |\boldsymbol{v_1} - \boldsymbol{u_2}|$ for droplet 1, and $U_2 = |\boldsymbol{v_2} - \boldsymbol{u_1}|$ for droplet 2. However, this stream function for Stokes flow does not apply to the system with a large Reynolds number. Hamielec and Johnson (1962, 1963) gave the stream function $\psi_h$ induced by a moving rigid sphere, which can be used for flows with large Reynolds numbers:

$$\psi_h = U\left(\frac{A_1}{\tilde{R}} + \frac{A_3}{\tilde{R}^2} + \frac{A_3}{\tilde{R}^3} + \frac{A_4}{\tilde{R}^4}\right)\sin^2\theta_0 - U\left(\frac{B_1}{\tilde{R}} + \frac{B_3}{\tilde{R}^2} + \frac{B_3}{\tilde{R}^3} + \frac{B_4}{\tilde{R}^4}\right)\sin^2\theta_0\cos\theta_0 \tag{6}$$

where $A_1, \ldots, B_4$ are functions only of Reynolds number $N_{Re}$ for each droplet. The method is valid for $N_{Re} < 5000$. But the solution deviates from the Stokes flow solution when $N_{Re} \to 0$ for small droplets. Therefore, it is needed to construct a stream function that applies to a wide range of $N_{Re}$. This work adopts a stream function that is a linear combination of $\psi_h$ and Stokes stream function $\psi_s$ (Pinsky and Khain, 2000)

$$\psi = \frac{N_{Re}\psi_h + N_{Re}^{-1}\psi_s}{N_{Re} + N_{Re}^{-1}} \tag{7}$$

which converges to stokes flow when $N_{Re} \to 0$. Then the induced flow field $\boldsymbol{u}$ is derived,

$$\boldsymbol{u} = -\frac{1}{\tilde{R}^2\sin\theta_0}\frac{\partial\psi}{\partial\theta_0}\hat{\boldsymbol{e}}_{\boldsymbol{R}} + \frac{1}{\tilde{R}\sin\theta_0}\frac{\partial\psi}{\partial\tilde{R}}\hat{\boldsymbol{e}}_{\boldsymbol{\theta}} = u_R\hat{\boldsymbol{e}}_{\boldsymbol{R}} + u_\theta\hat{\boldsymbol{e}}_{\boldsymbol{\theta}} \tag{8}$$

where $\hat{\boldsymbol{e}}_{\boldsymbol{R}}$ and $\hat{\boldsymbol{e}}_{\boldsymbol{\theta}}$ are unit vectors in the polar coordinate $(R, \theta_0)$. It can also be expressed in the Cartesian coordinate (x, z)

$$\boldsymbol{u} = (u_R\cos\varphi - u_\theta\sin\varphi)\hat{\boldsymbol{e}}_{\boldsymbol{z}} + (u_R\sin\varphi + u_\theta\cos\varphi)\hat{\boldsymbol{e}}_{\boldsymbol{x}} \tag{9}$$

where the direction of $\hat{\boldsymbol{e}}_{\boldsymbol{z}}$ is vertically down, the same as gravitation. $\varphi$ is the angle between $\hat{\boldsymbol{e}}_{\boldsymbol{z}}$ and the droplet velocity $\boldsymbol{v}$.

Both Stokes and Hamielec stream functions satisfy the no-slip boundary condition, i.e., the fluid velocity on the surface of the droplet is equal to the velocity of the droplet. Hamielec stream function is no-slip because those functions $A_1, \ldots, B_4$ in Eq. (6) satisfy $A_1 + 2A_2 + 3A_3 + 4A_4 = 1$ and $B_1 + 2B_2 + 3B_3 + 4B_4 = 0$, as long as the droplet is considered as a rigid sphere (Hamielec, 1963). These relations ensure that $u_\theta = -U\sin\theta_0$ at the surface of the droplet, which means the no-slip boundary condition. (Note that $u_\theta$ is the velocity of the fluid at droplet surface, and $U\sin\theta_0$ is the tangential velocity of the droplet surface.)

The drag coefficient $C$ in Eq. (4) is a function of $N_{Re}$,

$$C = 1 + \exp(a_0 + a_1X + a_2X^2) \tag{10}$$

where $X = \ln(N_{Re})$, and fitting constants $a_0, a_1, a_2$ are from table 2 of Beard (1976). The drag coefficient increases with Reynolds number. For example, the terminal velocity of a droplet of 16 μm in radius is 3.12 cm s$^{-1}$, with $N_{Re}$ =0.47 and $C$ =1.07; the terminal velocity of a droplet of 1024 μm in radius is 7.15m s$^{-1}$, with $N_{Re}$ =777 and $C$ =21.3. For $N_{Re} \to 0$, the drag coefficient C is 1.

For droplets with $r < 10 \ \mu m$, the assumption of no-slip boundary condition is no longer valid. The flow slips on the droplet surface. Therefore, the drag coefficient should multiply another coefficient (Lamb and Verlinde 2011, p386)

$$C' = C \cdot \left(1 + 1.26\frac{\lambda}{r}\right)^{-1} \tag{11}$$

where $\lambda$ is the free path of air molecules, and $r$ is the droplet radius.

### 3.3 The electric force term

The electric force is described by the third term on the right side of Eq. (4). The electric force includes the interactive force between the two charged droplets, and also an external electric force if there is an external electric field. It is well

known that the interaction between two point charges can be expressed as

$$\boldsymbol{F}_e = -\frac{1}{4\pi\varepsilon_0}\frac{q_1 q_2}{R^2}\hat{\boldsymbol{e}}_R \tag{12}$$

where $F_e$ is the interactive force between point charges $q_1$ and $q_2$, and $R$ is the distance between the two point charges. However, this inverse-square law does not apply to uneven charge distribution, such as the case of charged cloud droplets.

The interaction between charged conductors is a complex mathematical problem in physics. Davis (1964) demonstrated an

appropriate computational method for electric force between two spherical conductors in a uniform external field. The electric force depends on droplet radius $(r_1, r_2)$, charge $(q_1, q_2)$, centre distance $R$, electric field $E_0$, and the angle $\theta$ between the electric field and the line connecting the centres of two droplets (note that $\theta = \theta_0 + \varphi$). The resultant electric force acting on droplet 2 is expressed as

$$\boldsymbol{F}_{e2} = E_0 q_2 \cos\theta\hat{\boldsymbol{e}}_R + E_0 q_2 \sin\theta\hat{\boldsymbol{e}}_\theta +$$

$$\{r_2{}^2 E_0^2 (F_1 \cos^2\theta + F_2 \sin^2\theta) + E_0\cos\theta(F_3 q_1 + F_4 q_2) + \frac{1}{r_2^2}(F_5 q_1{}^2 + F_6 q_1 q_2 + F_7 q_2{}^2)\}\hat{\boldsymbol{e}}_R$$

$$+ \{r_2{}^2 E_0^2 F_8\sin2\theta + E_0\sin\theta(F_9 q_1 + F_{10}q_2)\}\hat{\boldsymbol{e}}_\theta$$

$$\tag{13}$$

where $\hat{\boldsymbol{e}}_R$ is the radial unit vector, and $\hat{\boldsymbol{e}}_\theta$ is tangential unit vector, $\boldsymbol{E}_0$ is the eternal electric field, $q_1$ and $q_2$ are the charges of droplet 1 and droplet 2 respectively, and parameters $F_1$ to $F_{10}$ are a series of complicated functions of geometric parameters

$(r_1, r_2, R;$ Davis 1964).

The electric force directly from the external field is shown as the two terms in the first line of Eq. (13), and can be simply written as $\boldsymbol{E}_0 q_2$ if combining the two terms. Line 2 and line 3 in Eq. (13) represent the interactive force from droplet 1 in the radial direction and tangential direction, respectively. Note that the third term in line 2 represent the interactive force from droplet 1 if there is no external electric field. Except for this term, all the other terms in lines 2 and 3 are the interactive forces

from droplet 1 due the induction from the external field.

Similarly, the resultant electric force $\boldsymbol{F}_{e1}$ acting on droplet 1 includes both the force directly from the external field and the interactive force from droplet 2. The sum of the electric forces on the two droplets, $\boldsymbol{F}_{e1} + \boldsymbol{F}_{e2}$, must equal to the external electric force acting on the system, which can be expressed as $\boldsymbol{E_0}(q_1 + q_2)$, because the two droplets can be considered as a system. Then, the electric force acting on droplet 1 could be derived immediately as

$$\boldsymbol{F}_{e1} = \boldsymbol{E_0}(q_1 + q_2) - \boldsymbol{F}_{e2} \tag{14}$$

Figure 2 is a schematic diagram showing the forces acting on each droplet in a pair. Also shown in Fig. 2 are the velocity of each droplet relative to the flow if there is no other droplets present ($\boldsymbol{v}$), and the flow velocity induced by the other droplet ($\boldsymbol{u}$). Droplet velocity relative to the flow is $\boldsymbol{v} - \boldsymbol{u}$. The electric field $\boldsymbol{E_0}$ is in the downward direction, the same as gravity. Droplet 1 has positive charge and droplet 2 has negative charge in this example. The forces acting on each droplet include gravity, flow drag force, and the electrostatic force, as seen on the right side of Eq. (4). For droplet 1, the electric force directly from the external field is in the downward direction, and is shown as $\boldsymbol{E_0}q_1$ in the figure. The interactive electric force from droplet 2, shown as $\boldsymbol{F}_{\text{inter}}$ in the figure, has a radial component and a tangential component, so that it is in a direction that does not necessarily align with the line connecting the two droplets. Because of the interactive electric force from droplet 2, the velocity $\boldsymbol{v}$ of droplet 1 is not in the vertical direction. The electrostatic force between charged droplets tend to make the droplets attract each other. This force is particularly strong when droplets are close to each other, thus to enhance collisions. The flow drag force on droplet 1 is in the opposite direction with $\boldsymbol{v} - \boldsymbol{u}$.

If there is no external electric field but only with charge effect, Eq. (13) is reduced to

$$\boldsymbol{F}_{e2} = \frac{1}{r_2^2}(F_5 q_1{}^2 + F_6 q_1 q_2 + F_7 q_2{}^2)\hat{\boldsymbol{e}}_R \tag{15}$$

To illustrate it, the comparison between the electrostatic forces derived by the inverse-square law and conductor model without electric field (i.e., Eq. 15) are shown in Fig. 3, where the electric force between droplets with opposite-sign charges (dashed lines) and with same-sign charges (solid lines) varies with distance. When $R \gg r_1, r_2$, we have $F_5, F_7 \to 0$, $F_6 \to r_2^2/R^2$, and it is also shown that two models are basically identical in remote distance. But when the spheres approach closely, the conductor interaction (blue lines) changes to strong attraction, because of electrostatic induction. The interaction is always attraction at small distance, regardless of the sign of charges. If there is only inverse-square law without electrostatic induction, it is obvious that same-sign charges must decrease collision efficiency. However, after taking electrostatic induction into account, the effects of same-sign and opposite-sign charges need to be reconsidered.

### 3.4 Terminal velocity and collision efficiency

The equations of motion (Eq. 4), along with the other equations in this section, are used to calculate the terminal velocities of charged droplets first. Note that the terminal velocity refers to the steady state velocity of a droplet relative to the flow when there is no other droplets present, as we mentioned earlier. Therefore, by setting the induced flow $\boldsymbol{u}$ to be 0, Eq. (4) can be integrated to obtain the terminal velocity of the droplets with electric charge and field.

Eq. (4), along with other equations, is also integrated to get the trajectories for the two droplets in any possible droplet pair $(r_1, q_1$ and $r_2, q_2)$ in various strengths of downward electric fields (0, 200 and 400 V cm$^{-1}$). The 2-order Runge-Kutta method is used for the integration. The initial settings of droplet positions and velocities, and the flow velocities are required. For convenience of computation, initial vertical distance is set to be $30(r_1+r_2)$, as an approximation of infinity. Initial flow velocity field $\boldsymbol{u_1}$ and $\boldsymbol{u_2}$ are set to be zero. Initial velocities of the two droplets are set to be the terminal velocities $\boldsymbol{V_1}$ and $\boldsymbol{V_2}$. Following their trajectories, the two droplets can either collide or not depending on the initial horizontal distance. We vary the initial horizontal distance between the two droplets using the bisection method, until we find a threshold distance $r_c$ that makes the two droplets follow the grazing trajectories and just exactly collide. The threshold distance is found with a precision of 0.1%. The collision cross section $S_c = \pi r_c^2$ and collision efficiency E are than calculated,

After computing the collision efficiencies $E$ for droplet pair with $(r_1, q_1)$ and $(r_2, q_2)$, the collection kernel $K(r_1, q_1, r_2, q_2)$ is derived then. With the collection kernel $K(r_1, q_1, r_2, q_2)$, the effect of electric charges and fields on droplet collision is determined by solving the SCE.

## 4  Model setup for solving the stochastic collection equation

### 4.1 Setting of the bins for droplet radius and charge

To solve the stochastic collection equation (Eq. 3) numerically, droplet radius and charge are both divided into discrete bins that are logarithmically equidistant. Droplet radius ranging from 2 to 1024 μm is divided into 37 bins, with the radius increased by a factor of $2^{1/4}$ from one bin to the next. Droplets with radii larger than 1024 μm are assumed to precipitate out and not included in the size distribution.

In each radius bin, droplets may have different amount and different sign of charges. For the bin of radius r, droplet charge ranges from $-32r^2$ to $+32r^2$ (in unit of elementary charge, and r in μm). This means that smaller droplets have a smaller range of charge. The setting here is based on the observations that the charge amount is proportional to the square of droplet radius, as discussed in Introduction. The upper limit charge bin of $32r^2$, is close to the thunderstorm condition of $42r^2$. The charge range is then divided into 15 bins, with the center bin having zero charge, 7 bins to the right having positive charges, and 7 bins to the left having negative charges. For the positive charge bins, the one next to the center bin has charge of $+0.5r^2$. The charge amount is increased by a factor of 2 from this bin to the next, until the upper limit of $32r^2$. The setting for the negative charge bins is completely symmetric to the positive charge bins. For the size bins and charge bins described above, a large matrix of kernel $K(r_1, q_1, r_2, q_2)$ is computed in advance as a lookup table for use in solving the SCE.

### 4.2 Redistribution of droplets into radius and charge bins after collision-coalescence

Droplet size and charge after collision-coalescence usually do not fall in any existing bins. A simple method is to linearly redistribute the droplets to the two neighbouring bins (Khain et al, 2004). We first redistribute droplets to the size bins. The ratio of redistribution is based on total-mass conservation and droplet-number conservation simultaneously. For example, to redistribute droplets with mass $m$ ($m_i < m < m_{i+1}$) and number $\Delta n$, a proportion of $\Delta n_i = \frac{m_{i+1}-m}{m_{i+1}-m_i}\Delta n$ is added to the $i$th bin,

and $\Delta n_{i+1} = \frac{m-m_i}{m_{i+1}-m_i}\Delta n$ is added to the $(i+1)$th bin. These droplets are then redistributed to the charge bins within each size bin, satisfying total-charge conservation and droplet-number conservation. For example, to redistribute droplets with charge $q$ ($q_{i,j} < q < q_{i,j+1}$) within the $i$th size bin, a proportion of $\Delta n_{i,j} = \frac{q_{i,j+1}-q}{q_{i,j+1}-q_{i,j}}\Delta n_i$ is added to the bin of $(i, j)$, and a proportion of $\Delta n_{i,j+1} = \frac{q-q_{i,j}}{q_{i,j+1}-q_{i,j}}\Delta n_i$ is added to the bin of $(i, j+1)$.

As shown in Fig. 4, the collision-coalescence between bin $(r_1, q_1)$ and bin $(r_2, q_2)$, shown with black dots, generates droplets

shown with the red dot. These newly generated droplets are then redistributed into 2 size bins, and further redistributed into 2 charge bins within each of the size bins, shown with blue dots. Note that the numbers close to each of the blue dots in Fig. 4 are the percentages of droplets that are redistributed into that bin. In fact, this method only reaches the first-order accuracy. Although Bott (1998) compared several methods to redistribute droplets with high-order correction, the two-parameter distribution is too complicated to do the high-order correction in this study.

### 4.3 The initial droplet size and charge distributions

The initial droplet size distribution used in this study is derived based on an exponential function in Bott (1998),

$$n(m) = \frac{L}{\bar{m}^2}\exp\left(-\frac{m}{\bar{m}}\right) \tag{16}$$

where $n(m)$ is the distribution of droplet number concentration over droplet mass, $L$ is the liquid water content, and $\bar{m}$ is the mean mass of droplets. This function is used to derive $n(lnr)$, which is the distribution of droplet number concentration over droplet radius. With the definitions of $n(m)$ and $n(lnr)$, and $m = 4\pi r^3\rho/3$, where $\rho$ is droplet density, we can derive $n(lnr)$ as

$$n(\ln r) = \frac{dN}{d \ln r} = r\frac{dN}{dr} = r\frac{dN}{dm}4\pi\rho r^2 = 4\pi\rho r^3 n(m) \tag{17}$$

By substituting Eq. (16) into Eq. (17), and assuming that $\bar{m} = 4\pi\bar{r}^3\rho/3$, where $\bar{r}$ is the mean radius, we have

$$n(\ln r) = L\frac{9r^3}{4\pi\bar{r}^6}exp\left(-\frac{r^3}{\bar{r}^3}\right) \tag{18}$$

Eq. (18) is used as the initial droplet size distribution for the calculations of collision-coalescence in this study. It has two parameters, $L$ and $\bar{r}$, and can be considered as a gamma distribution. Using parameters $L$ and $\bar{r}$ in the initial size distribution has an advantage in representing the aerosol effect. The parameter $L$ can be set as a constant. Using different

mean radius can represent different aerosol condition and different number concentration of cloud droplets.

12 cases with different initial conditions are considered to study the evolution of droplet distribution. The mean droplet radius $\bar{r}$ is set with three different sizes: 15 µm, 9 µm and 6.5 µm, where $\bar{r} = 15$ µm case represents clean conditions, and 6.5 µm represents polluted conditions. The liquid water content in our study is set to be $L$=1 g m$^{-3}$, which is a typical value in warm clouds according to observations (Warner, 1955, Miles et al. 2000). With the fixed liquid water content, a smaller mean radius corresponds to a larger number concentration. As shown in table 1, $\bar{r} = 15$ , 9, and 6.5 µm give an initial droplet number concentration of 71, 325, and 851 cm$^{-3}$, respectively.

For each $\bar{r}$, comparisons are made among four different electric conditions: (a) droplets are uncharged; (b) droplets are charged but with no external electric field, (c) droplets are charged and also with an external downward electric field of 200 V cm$^{-1}$, (d) droplets are charged and also with an external downward electric field of 400 V cm$^{-1}$. For the uncharged cloud, the initial distribution is shown in Fig. 5a, where all droplets are put in the bins with no charge. For the charged clouds, an initial charge distribution shown in Fig. 5b is made as follows.

To simulate an early stage of the warm-cloud precipitation, we need to distribute the droplets in each size bin to different charge bins, so that these droplets have different charges. Since there is little data on this, we assume a Gaussian distribution,

$$N(q) = \frac{N_0}{\sqrt{2\pi}\sigma} \exp\left(-\frac{q^2}{2\sigma^2}\right) \tag{19}$$

where $N_0$ is the number concentration in the size bin, and $\sigma$ is the standard deviation of the Gaussian distribution in that size bin. $N(q)$ represents the number concentration of droplets with charge $q$. This distribution satisfies electric neutrality $\bar{q} = 0$ . For different size bin, droplet number concentration $N_0$ is different. We purposely set the standard deviation $\sigma$ to be different for different size bins. For a larger size, the charge amount is larger, based on $\overline{|q|} = 1.31$ r$^2$ (q in unit of elementary charge and r in µm) as stated in the Introduction. Therefore, we set larger standard deviation $\sigma$ for the larger size bins. With this setting of droplet charge, the total amount of charge in each case is shown in Table 1. The $\bar{r} = 15$, 9, and 6.5 µm cases have an initial charge concentration of 9438, 15638, and 21634 e cm$^{-3}$, respectively, for both positive charge and negative charge.

The initial electric charges, and electric field strength are set according to the conditions in warm clouds or the early stage of thunderstorms. In fact, in some extreme thunderstorm cases, both the electric charge and field could be one order of magnitude larger (Takahashi, 1973) than the values used in this study. Furthermore, in natural clouds, the electric charge on a droplet leaks away gradually. In this study, the charge leakage is assumed as a process of exponential decay (Pruppacher and Klett, 1997), and the relaxation time is set to $\tau$ =120 min. Namely, all the bins lose $\frac{\Delta t}{\tau}$ of electric charge in each step of time $\Delta t = 1s$.

## 5 Results

### 5.1 Collision efficiency

Here we present collision efficiencies for typical droplet pairs to illustrate the electrostatic effects. During the evolution of droplet size distribution, the radius and charge amount of colliding droplets have large variability. In addition, the charge

sign of the colliding droplets may be the same or the opposite. Therefore, only some examples are shown.

The collision efficiencies for droplet pairs with no electric charge and field are presented in Fig. 6 as a reference. Collector droplets with radii larger than 30 μm are shown here to represent the precipitating droplets. The calculated collision efficiencies from this study are also compared with the measurements from previous studies. It is seen that results from this study are generally consistent with the measurements. Collision efficiencies increase as $r_2$ changes from 2 to 14 μm, and also

increase as $r_1$ changes from 30 to 305 μm. For two droplets that are both large enough, collision efficiency could be close to 1.

Figure 7 shows the collision efficiencies for droplet pairs with electric charge and field. The detailed characteristics of the droplet pairs are shown in Table 1. Basically, droplet pairs that have no charge, with same-sign charges, and with opposite-sign charges are selected here, and under the 0 and 400 V m$^{-1}$ electric fields. Results for the collector droplet with a radius of

395 30 μm (Fig. 7a) and 40 μm (Fig. 7b) are shown. When comparing Fig. 7a and 7b, it can be seen that electrostatic effects are less significant for a larger collector. The electrostatic effects are even weaker for collector radius larger than 40 μm (figures not shown). Therefore, we use the 30 μm collector as an example to explain the electrostatic effects on collision efficiencies below.

For the collector droplet with a radius of 30 μm (Fig. 7a), noticeable, and sometimes significant electrostatic effect can be

seen. Compared to the droplet pair with no charge (line 1), the positively-charged pair under no electric field (line 2) has a slightly smaller collision efficiency, due to the repulsive force. As can be seen in Fig. 3, when the charged droplets move together, they first experience repulsive force, then attractive force at small distance. The integrated effect is that the droplets have smaller collision efficiency. The results for negatively-charged pair under no electric field are identical to line 2 and therefore are not shown. When a downward electric field of 400 V m$^{-1}$ is added, the positively-charged pair (line 3) has a

collision efficiency very close to the pair with no charge. This implies that the enhancement of collision efficiency by the electric field offsets the repulsive force effect. For a negatively-charged pair in a downward electric field (line 4), the collision efficiency with small $r_2$ is significantly enhanced. This could be easily explained by electrostatic induction: the strong downward electric field induces positive charge on the lower part of the collector droplet (even though it is overall negatively-charged), so the negative-charged collected droplet below experiences attractive force.

As for a pair with opposite-sign charges, line 5 in Fig. 7a shows that the collision efficiency is enhanced by the electrostatic effect even when there is no electric field. The collision efficiency is nearly an order of magnitude higher with $r_2 < 5$ μm. Line 6 in Fig. 7a shows that, with an electric field of 400 V cm$^{-1}$, the electrostatic effect for the pairs with opposite-sign charges is even stronger. There is also an interesting feature in Fig. 7a: as the collector and collected droplets have similar sizes, collision efficiency is high for the pairs with opposite-sign charges. This is quite different from the other

four lines, where collision efficiencies are very low for droplet pairs with similar sizes.

Figure 8 shows the collision efficiencies for droplet pairs with charge and field, with smaller collectors. The collector droplet has a radius of 10 μm (Fig. 8a) and 20 μm (Fig. 8b) here, and can be used to represent cloud droplets. Collision efficiencies for these smaller collectors are much smaller than 1 when there is no charge (line 1 in Figs. 8a and 8b), which is already well known in cloud physics community. However, the electrostatic effects are so strong that the collision efficiencies could be significantly changed for these collectors. For the collector droplet with a radius of 10 μm (Fig. 8a), the positively-charged pair has a very small collision efficiency that is out of the scale in the figure, due to the dominating effect of the repulsive force as discussed above. For the positively-charged pair under a downward electric field, the collision efficiencies is on the similar order of magnitude as the pair with no charge. For the negatively-charged pair under the downward electric field, and for the pairs with opposite-sign charges, the electrostatic effects is very strong. The negatively-charged pair even has the collision efficiency increased by two orders of magnitude. Similarly, for the collector droplet with a radius of 20 μm (Fig. 8b), the electrostatic effect can lead to an order of magnitude increase in collision efficiencies.

It is evident that droplet charge and field can significantly affect collision efficiency, especially for smaller collectors. This means that the electrostatic effects depend on the radius of collector droplets, and mainly affects small droplets. The section below provides a detailed description on how these electrostatic effects can influence droplet size distributions.

## 5.2. Evolution of droplet size distribution

This part shows the electrostatic effects on the evolution of different droplet size distribution. As discussed in Section 4, this study uses three initial size distributions, where $\bar{r}$ = 15 μm, 9 μm and 6.5 μm, respectively. For each initial size distribution, comparisons are made among four different electric conditions, namely uncharged droplets, charged droplets without electric field, charged droplets with a 200 V cm$^{-1}$ electric field, and charged droplets with a 400 V cm$^{-1}$ electric field. Note that "charged droplets" here refers to the initial charge distribution shown in Fig. 5. We also compare the results of the uncharged clouds with $\bar{r}$ = 15 μm, 9 μm and 6.5 μm, which represents the aerosol effects, and then investigate whether the electrostatic effects can mitigate the aerosol effects during the collision-coalescence process.

Figure 9 shows the evolution of droplet size distribution with initial $\bar{r}$ = 15 μm, which has an initial droplet number concentration of 71 cm$^{-3}$. The 4 rows show different times (t = 7.5, 15, 22.5, and 30 min) during the simulated evolution. The left column shows the size distribution of droplet mass concentration $M(\ln r)$, and the right column shows the size distribution of droplet number concentration $n(\ln r)$. They are related as $M(\ln r) = 4\pi r^3 \rho/3 \cdot n(\ln r)$. A second mode in size distribution gradually form as droplets undergo the collision-coalescence process from t = 7.5 to 30 min. Although the second mode can be clearly seen in the plots of $n(\ln r)$, we show $M(\ln r)$ here so that the second mode can be seen as a peak. In each panel, the dotted line denotes the initial size distribution (t = 0 min) for reference. It is seen that droplet size distributions under 4 electric conditions have similar behavior for initial $\bar{r}$ = 15 μm: they all evolve to a double-peak form, regardless of electric charge or field. At 30 min, the 4 cases all have a modal radius of about 200 μm (Fig. 9d). The

electrostatic effect is not notable for large droplets in the $\bar{r} = 15$ μm cases, because the initial radius is large enough to start

gravitational collision-coalescence quickly.

    The evolution of droplet total number concentration and total positive charge concentration (also equal to the total negative charge concentration) is shown in Fig. 10. It is evident that droplet total number concentration decreases from 71 cm$^{-3}$ to less than 5 cm$^{-3}$ in 30 minutes, and is nearly not affected by the 4 different electric conditions. Both of the positive charge and negative charge concentration decrease from 9384 to about 1000 e cm$^{-3}$, as droplets with opposite-sign charges

go through collision-coalescence and charge neutrality occurs.

    Figure 11 shows the evolution of droplet size distribution with initial $\bar{r} = 9$ μm. For the uncharged cloud, it takes 60 min to have the second peak grow to about 200 μm. Therefore, the 4 panels of Fig. 11 show the simulated evolution for t = 15, 30, 45, and 60 min. The charges and the electric fields have more significant effect in the $\bar{r} = 9$ μm case than in the $\bar{r} = 15$ μm case. It is seen that, at 15 and 30 min, the clouds with different electric conditions evidently differ from each other, but the

second mode is not obvious. At 45 min, the electrostatic effects on the second peak is evident. The charged cloud (red line) evolves more quickly than the uncharged cloud, as can been from the lower first peak and the growing second peak. Moreover, the downward electric fields further boost the collision-coalescence process of charged droplets (green and purple lines). At 60 min, the modal radius of the second peak is about 200 μm for the uncharged cloud, 300 μm for the charged cloud but without an electric field, 500 μm for the charged cloud with a field of 200 V cm$^{-1}$, and 700 μm for the charged

cloud with a field of 400 V cm$^{-1}$, respectively.

    As for the evolution of droplet total number concentration and charge concentration, Fig. 12 shows that they are distinctly affected by the 4 different electric conditions. The charged cloud with a field of 400 V cm$^{-1}$ has very low droplet number concentration and charge concentration at 60 min. the electrostatic effects play an important role in converting smaller droplets to larger droplets. The 2-dimensional distribution of droplet mass concentration for $\bar{r} = 9$ μm at 60 min is shown in

Fig. 13. Figure 13a is for the uncharged situation. Figs. 13b, 13c, and 13d are for the situations with charges and with electric fields of 0, 200, 400 V cm$^{-1}$, respectively. After 60 min of evolution, the distribution of mass over the charge bins is still symmetric. It is also shown that both mass and charges are transported from smaller droplets to larger droplets during collision-coalescence. Note that the integration of this 2-dimensional distribution along the charge bins gives the 1-dimentional distribution over droplet size at 60 min as shown in Fig. 11d.

Figure 14 shows the evolution of droplet size distribution with initial $\bar{r} = 6.5$ μm. For the uncharged cloud, it takes 120 min to have the second peak grow to about 200 μm. Therefore, the 4 panels of Fig. 14 show the simulated evolution for t = 30, 60, 90 and 120 min. The enhancement by the electric field on collision-coalescence process is much more obvious than $\bar{r}$ = 9 μm. After 90 min of evolution, the uncharged cloud (blue line) and charged cloud without field (red line) are almost the same as the initial distribution. This is because the droplets are too small to initiate gravitational collision. At 120 min, a

second peak has formed for the situations with no charge and with charge but no field. In contrast, under the external electric field of 200 and 400 V cm$^{-1}$ (green and purple lines), the cloud droplets grow much more quickly than the no-field situations. Some droplets even have evolved to larger than 1024 μm, which are supposed to precipitate out from the clouds. The

evolution of droplet total number concentration and charge concentration is shown in Fig. 15, which indicates that droplet total number concentrations and charge concentration are strongly affected by the electrostatic effects. These results show that, the electric field would remarkably trigger the collision-coalescence process for the small droplets.

As for the initial mean droplet radius $\bar{r} < 6$ μm (figure not shown), similar to Fig. 14, the droplet size distribution of uncharged and charged cloud without electric field would nearly have no difference, while the effect of electric fields is much stronger. This means that charge effect is relatively small compared to electric fields when the initial droplet radius of the cloud is small enough.

Now we compare the electrostatic effects shown above with the aerosol effects. Let us take the cases with $\bar{r} = 15$ μm and $\bar{r} = 9$ μm as examples. When there is no electrostatic effects, the case with $\bar{r} = 15$ μm can develop a significant second peak in the size distribution in less than 30 min, while it takes about 60 min for the $\bar{r} = 9$ μm case to develop a similar second peak, as can be seen in Figs. 9 and 11. This can be regarded as an aerosol effect. When considering the electrostatic effects, it only takes about 45 min for the $\bar{r} = 9$ μm case to develop a similar second peak, as can be seen in Fig. 9. Therefore, the aerosol-induced precipitation suppression effect is mitigated by the electrostatic effects.

## 6 Discussion

According to Eq. (2), collection kernel $K$ is composed of the collision efficiency $E$, relative terminal velocity, and coalescence efficiency $\varepsilon$. It is found that the total electrostatic effect on $K$ is mainly contributed by $E$. The relative terminal velocity term also contributes to the collection kernel $K$. As mentioned in Section 3.4, terminal velocities $V_1$ or $V_2$ are derived by simulating just single one charged droplet in air with a certain electric field, and letting it fall until its velocity converges to the terminal velocity. Therefore, the electric field can affect terminal velocities of charged droplets, thus to affect the collection kernels. Terminal velocities of droplets in an external electric field is illustrated in Fig. 16. In a downward electric field of 400 V cm$^{-1}$, the terminal velocity of a large droplet is hardly affected. The difference of velocity caused by the electric field for $r = 1000$ μm does not exceed 1%, and the one for 100 μm does not exceed 5%. On the contrary, electric fields strongly affect the terminal velocities of charged small droplets. For $r < 5$ μm, the terminal velocity of a negatively-charged droplet even turns "upwards". Electric fields mainly affect terminal velocities of small charged droplets because droplet mass $m \propto r^3$, while droplet charge $q \propto r^2$ according to observation. Therefore, $q \propto m^{2/3}$ means that the acceleration contributed by the electric force decreases with increasing droplet mass.

This study still neglects some possible electrostatic effects in collision-coalescence process. Electrostatic effect on coalescence efficiency $\varepsilon$ is neglected. Rebound (collide but not coalesce) happens because of an air film temporally trapped between the two surfaces, which is a barrier to coalescence. This barrier may be overcome by strong electric attraction occurring at small distance. Many experiments show that electric charges and fields would enhance coalescence efficiency, such as Jayaratne and Mason (1964) and Beard et. al. (2002). The latter experiment indicates that even minimal electric charge

incapable of enhancing collision can significantly increase $\varepsilon$, while the marginal utility of larger electric charges on $\varepsilon$ is very small. However, there is no proper numerical model to evaluate the effect. Therefore, this study may underestimate the electrostatic effect on droplet collision-coalescence process.

Induced charge redistribution is also neglected when rebound happens. For instance, let us consider a rebound event in a positive (downward) electric field. The larger droplet is often above the smaller droplet, and the smaller one will carry positive charge instantaneously according to electrostatic induction, then move apart. The rebound would cause charge redistribution between the pair. This may lead to some change in the evolution of clouds.

## 7 Conclusion

The effect of electric charges and atmospheric electric fields on cloud droplet collision-coalescence and on the evolution of cloud droplet size distribution is studied numerically. The equations of motion for cloud droplets are solved to get the trajectories of droplet pair of any radii (2 to 1024 μm) and charges (-32 to $+32r^2$, in unit of elementary charge, droplet radius $r$ in unit of μm) in different strength of downward electric fields (0, 200 and 400 V cm$^{-1}$). Based on trajectories, we determine whether a droplet pair collide or not. Thus, collision efficiencies for the droplet pairs are derived. It is seen that collision efficiency is increased by electric charges and fields, especially when the droplet pair are oppositely charged or both negatively charged in a downward electric field. We consider these effects as the electrostatic effects. The increase of collision efficiency is particularly significant for a pair of small droplets.

With collision efficiencies derived in this study, the SCE is solved to simulate the evolution of cloud droplet size distribution under the influence of electrostatic effects. The initial droplet size distributions include $\bar{r}$ = 15 μm, 9 μm, and 6.5 μm, and the initial electric conditions include uncharged and charged droplets (with charge amount proportional to droplet surface area) in different strength of electric fields (0, 200 and 400 V cm$^{-1}$). The magnitudes of electric charges and fields used in this study represent the the observed atmospheric conditions. In the natural precipitation process, the charge amount, the strength of electric fields, and the time scale of the evolution are similar to those in this study. It is seen that the electrostatic effects are not notable for clouds with initial $\bar{r}$ =15μm, since the initial radius is large enough to start gravitational collision quickly. For clouds with initial $\bar{r}$ = 9 μm, electric charges could enhance droplet collision evidently compared to the uncharged condition when there is no electric field, and the existence of electric fields further accelerates collision-coalescence and the formation of large drops. For clouds with initial $\bar{r}$ = 6.5 μm, it is difficult for gravitational collision to occur. The enhancement of droplet collision merely by electric charge without field is still not significant, but electric fields could remarkably enhance the collision process. These results indicate that clouds with droplet sizes smaller than 10 μm are more sensitive to electrostatic effects, which can significantly enhance the collision-coalescence process and trigger the raindrop formation.

It is known that the increase of aerosol number and therefore the decrease of cloud droplet size lead to suppressed precipitation and longer cloud lifetime. But with the electrostatic effect, the aerosol effect can be mitigated to a certain extent. The three initial droplet size distributions used in this study, with $\bar{r}$ = 15, 9, and 6.5 μm, have an initial droplet number

concentration of 71, 325, and 851 cm$^{-3}$, respectively. The three cases can represent different aerosol conditions. Smaller droplets size and higher droplet number concentration represents a more polluted condition. It is seen that collision-coalescence process is significantly slowed down as $\bar{r}$ changes from 15 μm to 9 μm, and to 6.5 μm. It takes about 30 min, 60 min, and 120 min, respectively, for the three cases to form a mode of 200 μm in droplet size distribution. We consider this as an aerosol
effect. When the electrostatic effect is considered, the case with $\bar{r} = 9$ μm now only takes about 45 min to form the mode of 200 μm. Therefore, the enhancement of raindrop formation due to electrostatic effects can mitigate the suppression of rain due to aerosols.

**Code and data availability**

Data and programs are available from Shian Guo ([guoshian@pku.edu.cn](mailto:guoshian@pku.edu.cn)) upon request.

**Author contribution**

Shian Guo developed the model, wrote the codes of program, and performed the simulation. Huiwen Xue advised on the case settings of the numerical simulation. Huiwen Xue and Shian Guo worked together to prepare the manuscript.


**Competing interests**

The authors declare that they have no conflict of interest.

**Acknowledgements**

This study is supported by the National Innovative Training Program and the Chines NSF grant 41675134. We are grateful to Jost Heintzenberg and Shizuo Fu for constructive discussions on this study.

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

**Table 1.** Meaning of the 6 different curves in Fig. 7.

| $r_1 = 30$ μm curve settings | $q_1$ (e) | $q_2$ (e) | electric field $E_0$ (V cm$^{-1}$) |
|---|---|---|---|
| (1) | 0 | 0 | 0 |
| (2) | $+32\ r_1^2$ | $+32\ r_2^2$ | 0 |
| (3) | $+32\ r_1^2$ | $+32\ r_2^2$ | +400 |
| (4) | $-32\ r_1^2$ | $-32\ r_2^2$ | +400 |
| (5) | $+32\ r_1^2$ | $-32\ r_2^2$ | 0 |
| (6) | $+32\ r_1^2$ | $-32\ r_2^2$ | +400 |

**Table** 2. Total number concentration and charge content for all initial droplet distributions

| mean radius $\bar{r}$ (μm) | total number concentration (cm$^{-3}$) | total positive charge concentration (e cm$^{-3}$) | total negative charge concentration (e cm$^{-3}$) |
|---|---|---|---|
| 15 | 70.6 | +9384 | -9384 |
| 9 | 324.8 | +15638 | -15638 |
| 6.5 | 850.5 | +21634 | -21634 |

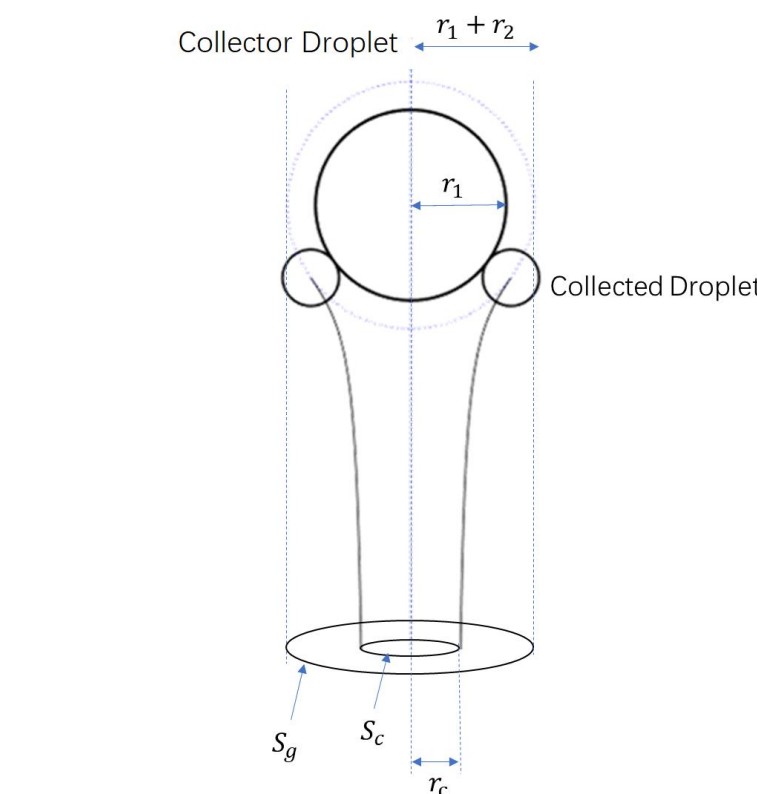


**FIG. 1.** A geometry sketch map of collision. The initial vertical distance between the center of the two droplets is set to be $30(r_1 + r_2)$, which can approximately represent that the droplets initially has a distance of infinity. To calculate the collection cross section $S_c = \pi r_c^2$, the initial horizontal distance needs to be changed with the bisection method, until it converges to $r_c$. Collision happens only when the initial horizontal distance is smaller than $r_c$.


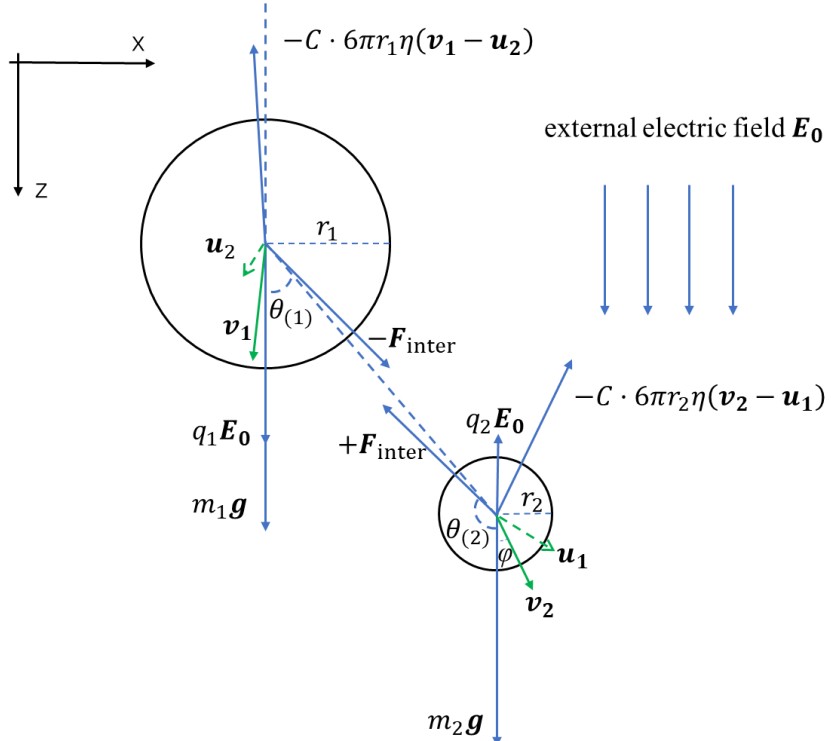

**FIG. 2.** A schematic diagram of all the forces acting on the two droplets, as well as droplet velocities and the induced flow velocities. The electric field $E_0$ is vertically downward, and electric charges $q_1 > 0$, $q_2 < 0$. Note that the electrostatic force $F_{e1}$, $F_{e2}$ include two parts: the electric force from the other droplet ($F_{inter}$, in the figure), and the force purely from the external electric field ($q_1 E_0$, $q_2 E_0$ in the figure).

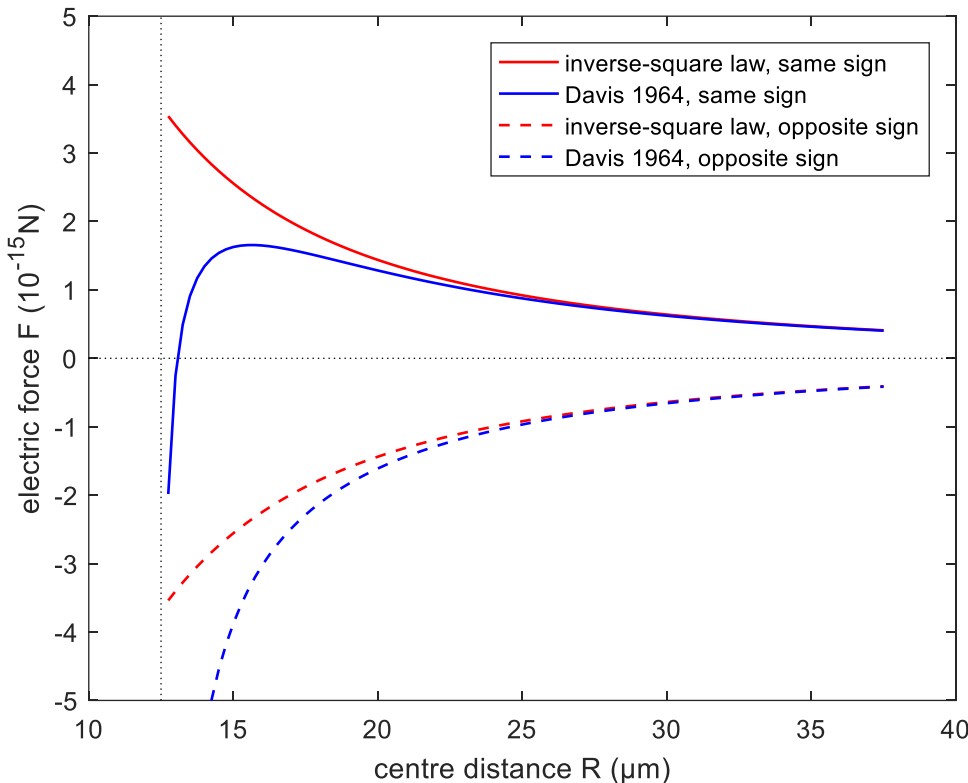


**FIG. 3.** Comparison of the electric force from the conductor model (Davis 1964, Eq. 15 in this study) and the inverse-square law (Eq. 12 in this study). Positive force represents repulsion and negative force represents attraction. Radius of the pair is set to $r_1 = 10$ μm and $r_2 = 2.5$ μm respectively. Solid lines are for the droplet pair with the same sign of electric charges, with $q_1 = +100$ e, and $q_2 = +25$ e. Dashed lines are for droplets with the opposite sign of electric charges, with $q_1 = +100$ e, and

$q_2 = -25$ e.

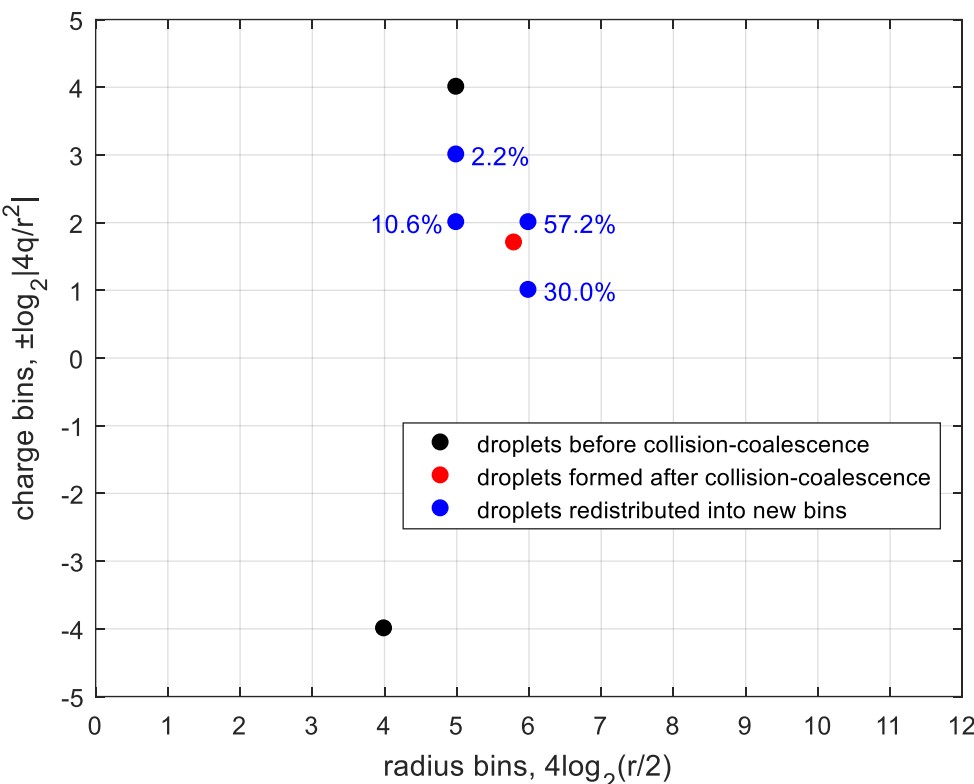

**FIG. 4.** An example of droplet redistribution to the new size and charge bins after collision-coalescence. Black dots denote the two bins of droplets before collision-coalescence. The red dot denotes the droplets after collision-coalescence but not on the bin grids. Blue dots denote the droplets that are redistributed to the new bins. Numbers close to the blue dots are the percentage of droplets that are redistributed into that bin. The redistribution method is constrained by particle number conservation, mass conservation, and charge conservation.

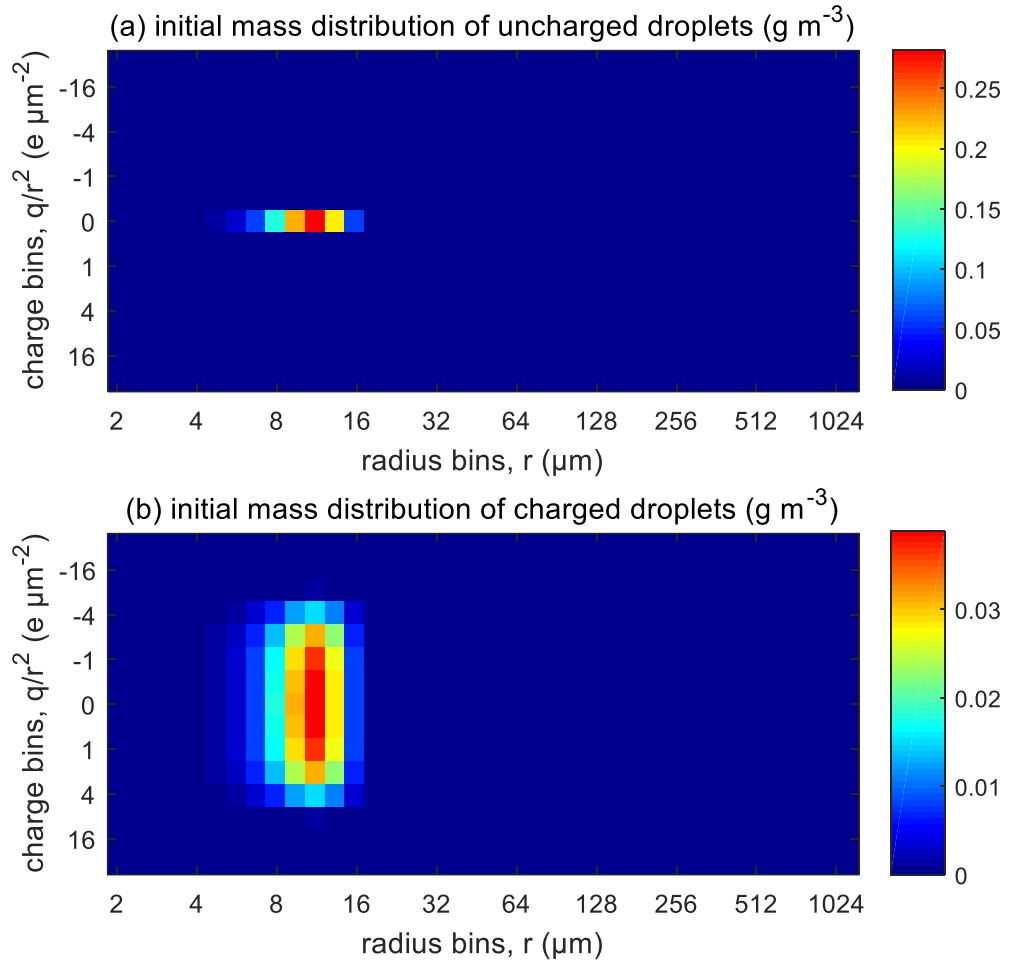

**FIG. 5.** The initial droplet mass distributed over the size and charge bins. Colours stand for water mass content in the bins (in unit of g m$^{-3}$). (a) Uncharged droplets (b) charged droplets.

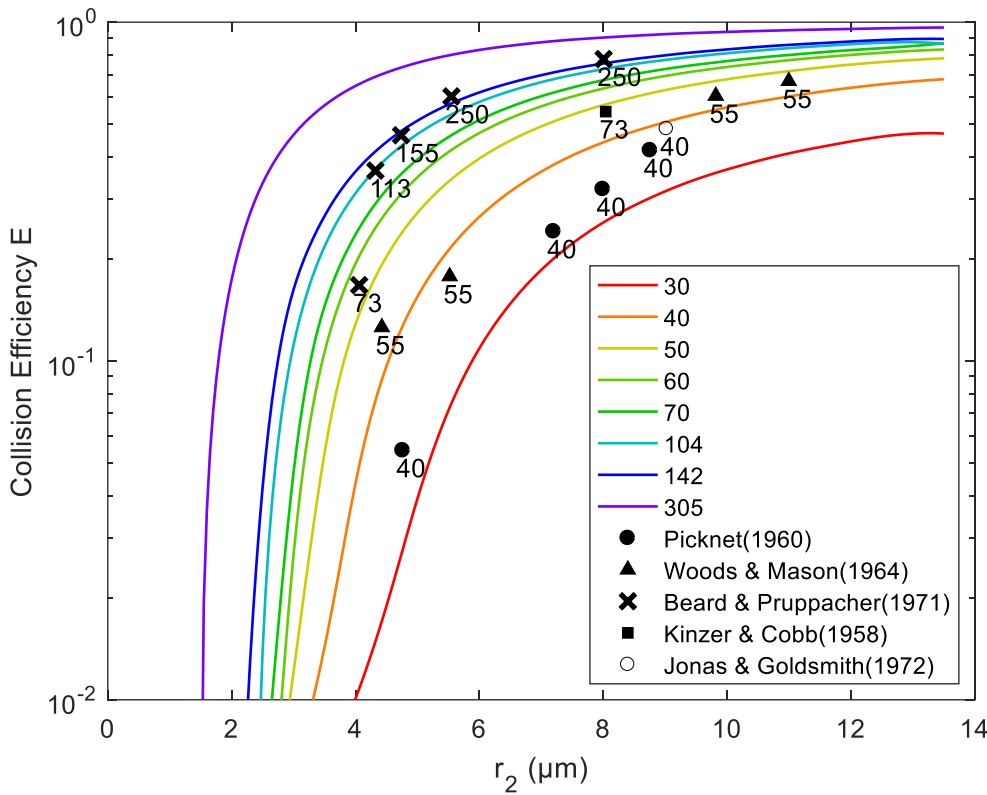


**FIG. 6.** Collision efficiency for droplets with no electric charge or field. Lines are results computed in this study. Different lines represent different collector radius $r_1$, from 30 to 305 μm. X-axis denotes the collected droplet radius $r_2$ . Scatter points are collision efficiencies from previous experimental studies.


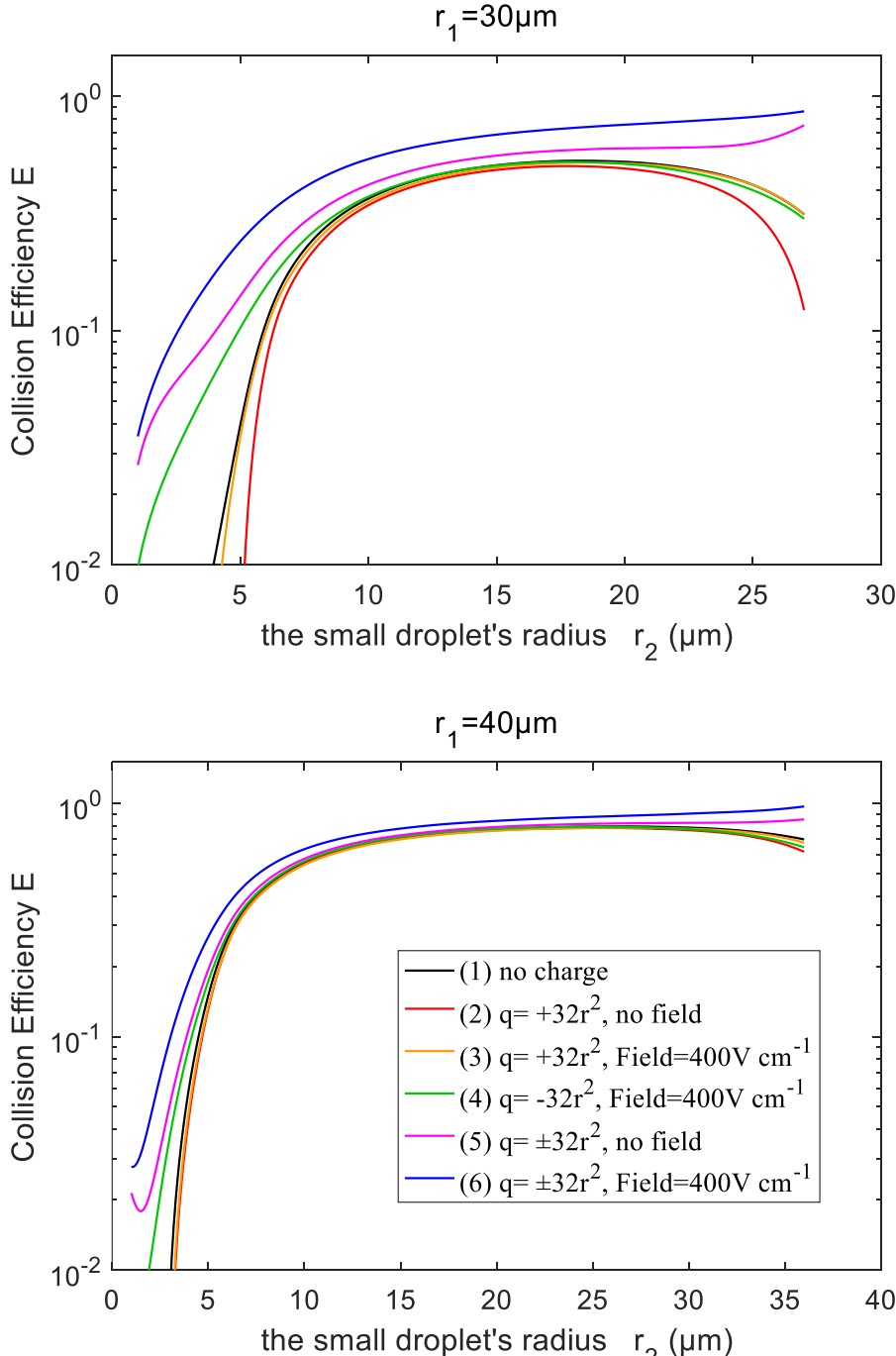

**FIG. 7.** Collision efficiency for droplets with electric charge and field. The radius of the collector droplet $r_1$ is: (a) 30.0 μm, (b) 40.0 μm. X-axis denotes the collected droplet radius $r_2$. The two droplets carry electric charges proportional to $r^2$. The lines for droplet pairs with no charge (line 1 in Fig. 7a and 7b) are the same as the 30 μm and 40 μm lines in Fig. 6.


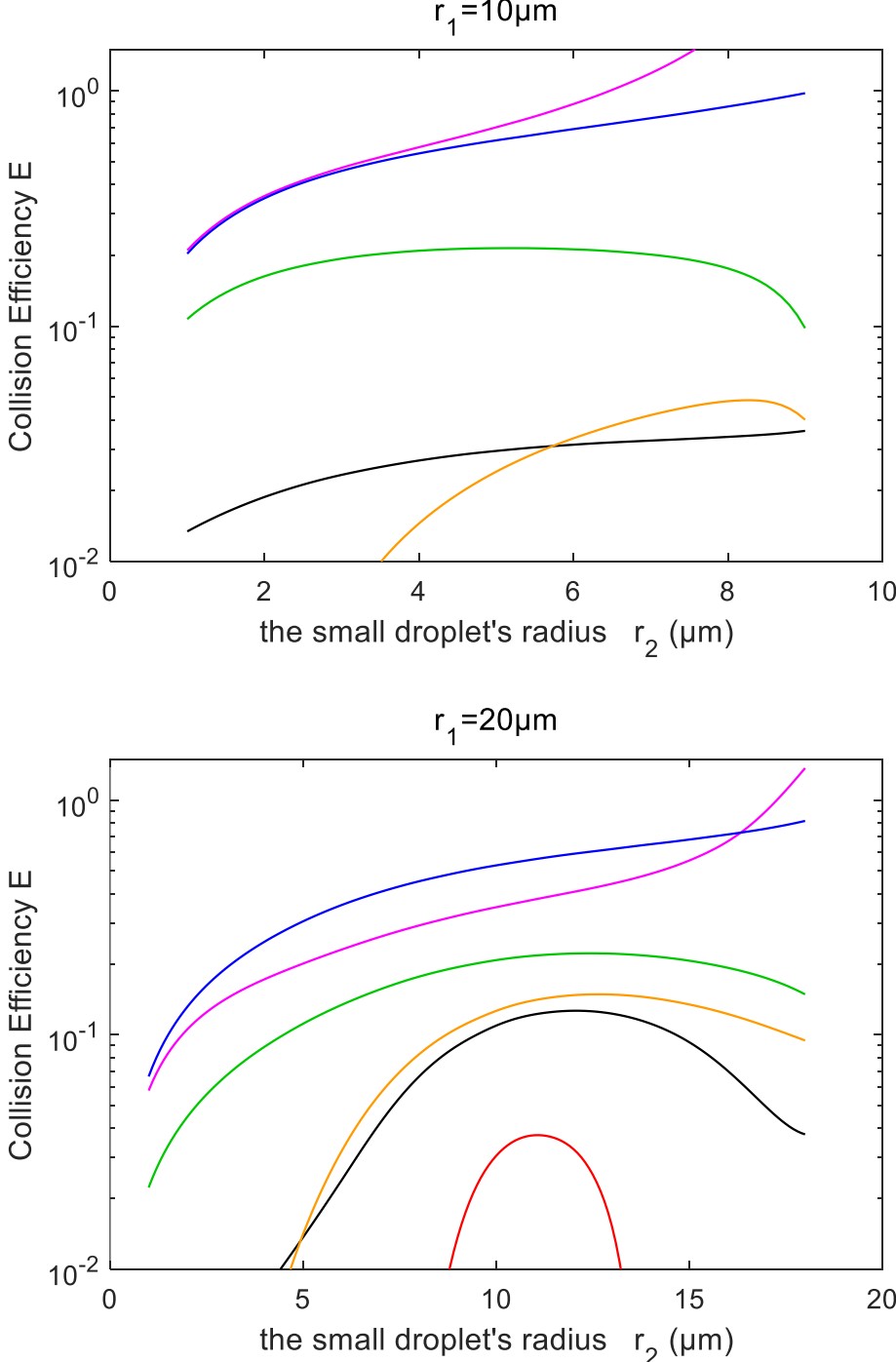

**FIG. 8.** Collision efficiency for droplets with electric charge and field. The radius of the collector droplet $r_1$ is: (a) 10.0 μm, (b) 20.0 μm. The other characteristics of the droplet pairs are similar to those in Fig. 7.

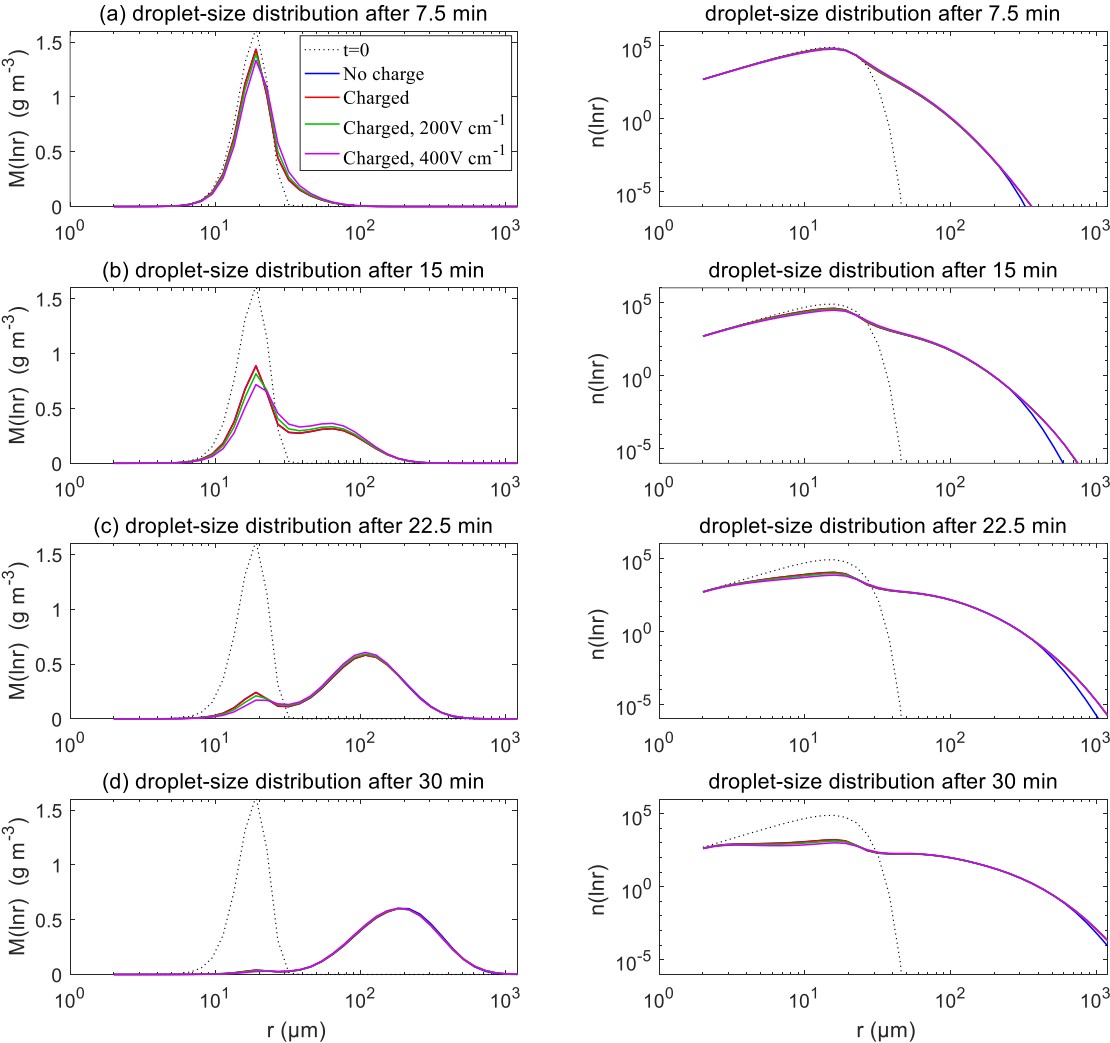

**FIG. 9.** The evolution of droplet size distribution with initial $\bar{r} = 15$ μm. These panels show different stages of the evolution from top to bottom. The left column shows the size distribution of droplet mass concentration, and the right column shows the size distribution of droplet number concentration, on logarithmic scales. In each panel, comparisons are made for 4 different electric conditions. Blue lines denote the uncharged cloud. Red lines denote charged cloud without electric field. Green and purple lines denote charged cloud with a field of 200 V cm$^{-1}$ and 400 V cm$^{-1}$, respectively. Dotted lines show the initial size distribution.

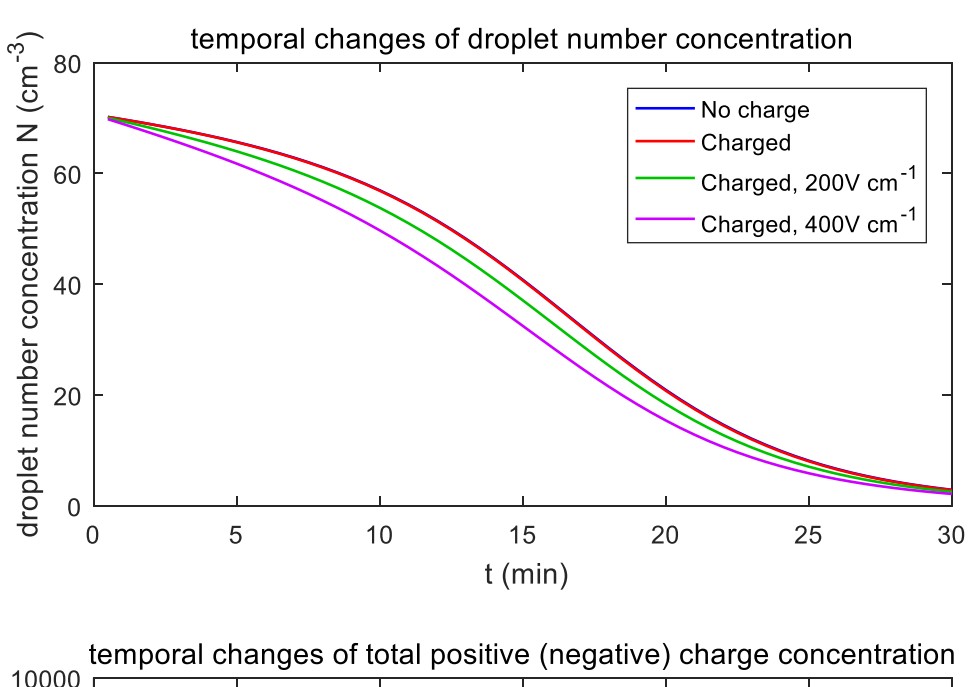

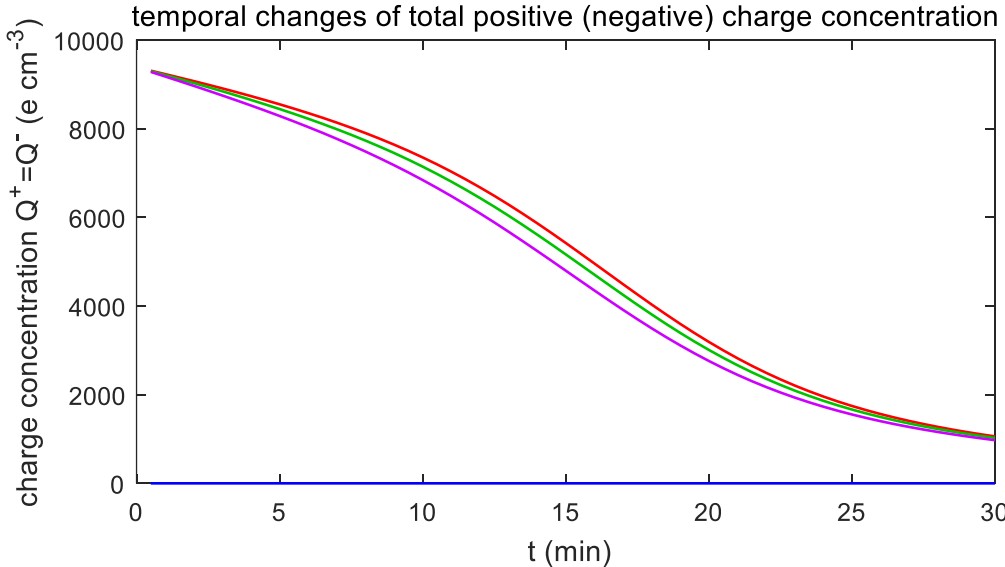

**FIG. 10.** Temporal changes of droplet total number concentration and total charge content for $\bar{r} = 15$ μm

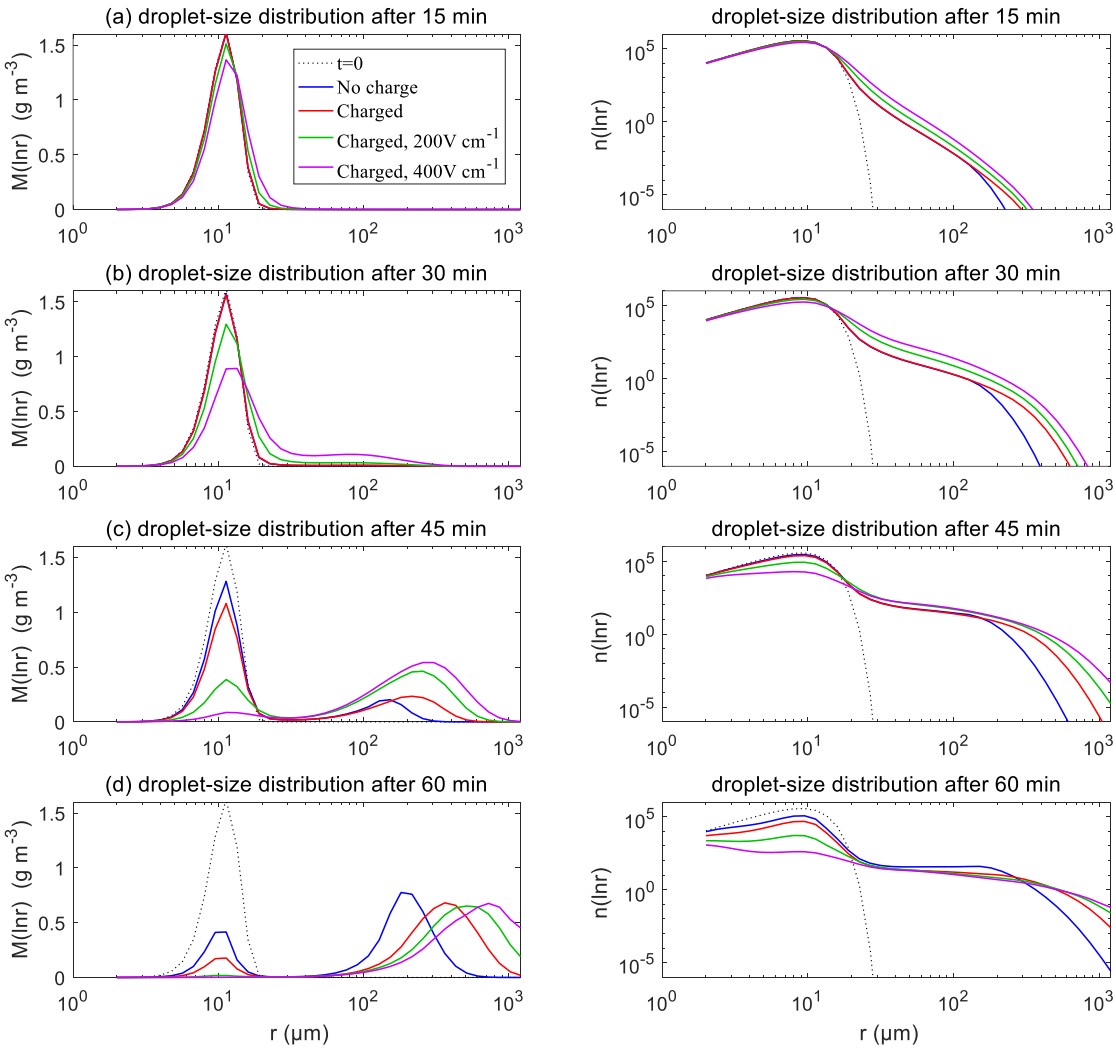

**FIG. 11.** The evolution of the droplet size distribution with initial $\bar{r} = 9$ μm.

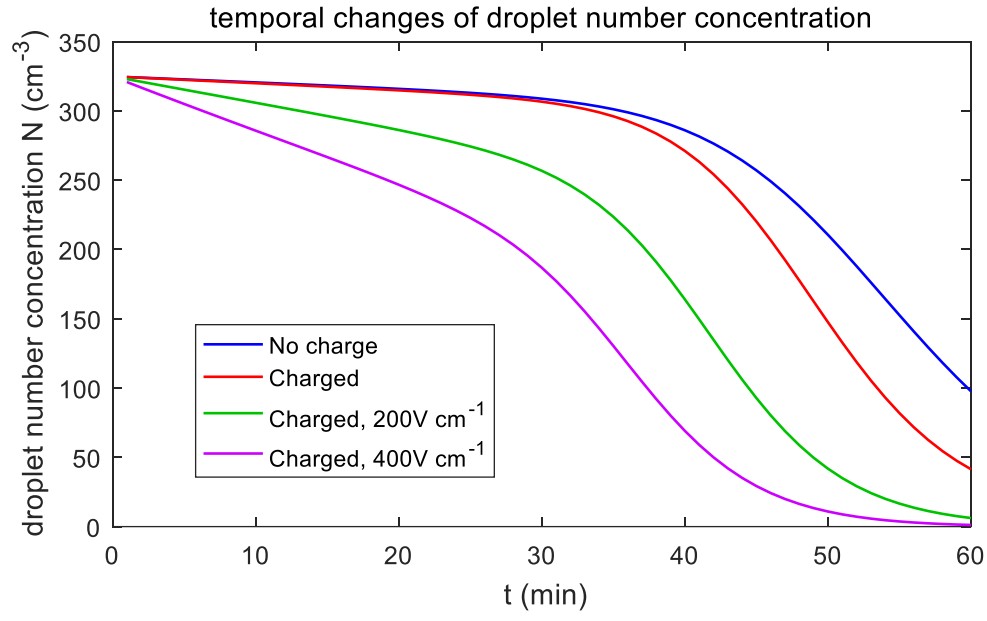

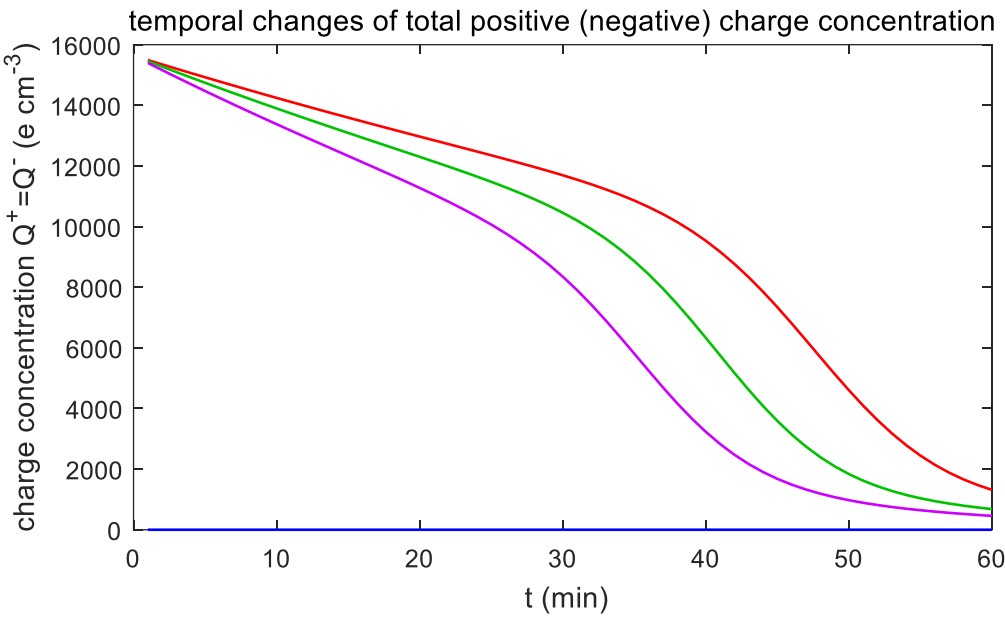

**FIG. 12.** Temporal changes of droplet total number concentration and total charge content for $\bar{r} = 9$ μm

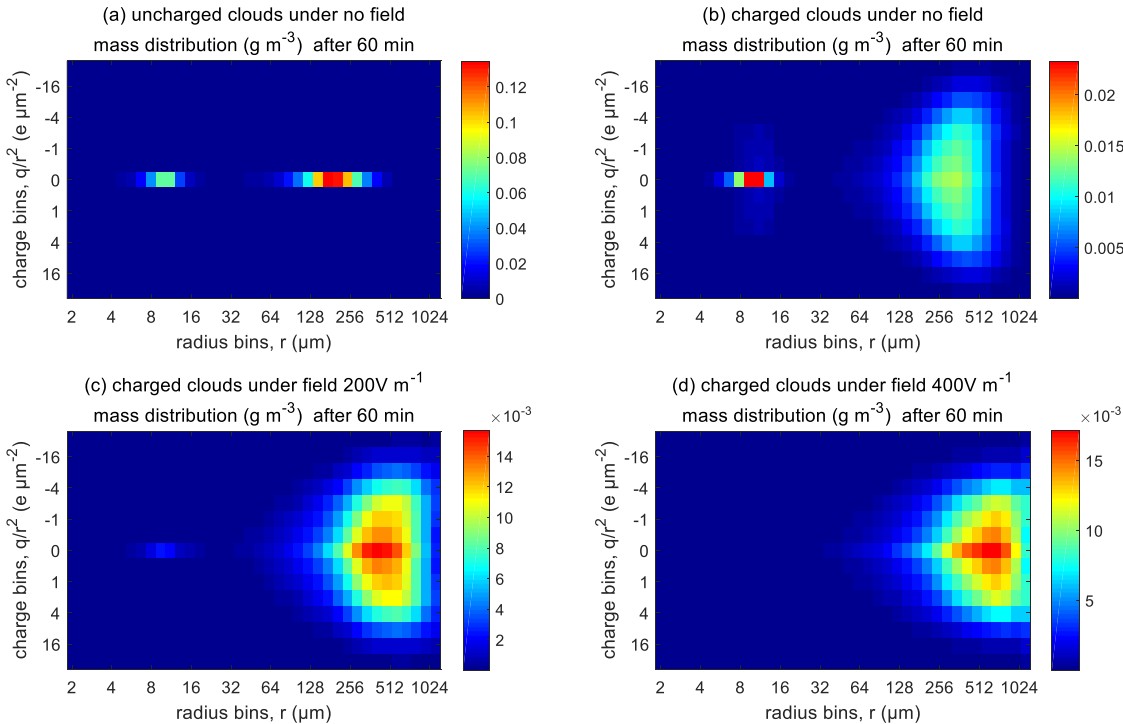

**FIG. 13.** Comparison of evolutions of 2-dimensional distribution of droplet mass concentration with different electric conditions at 60 min (initial $\bar{r} = 9$ μm).

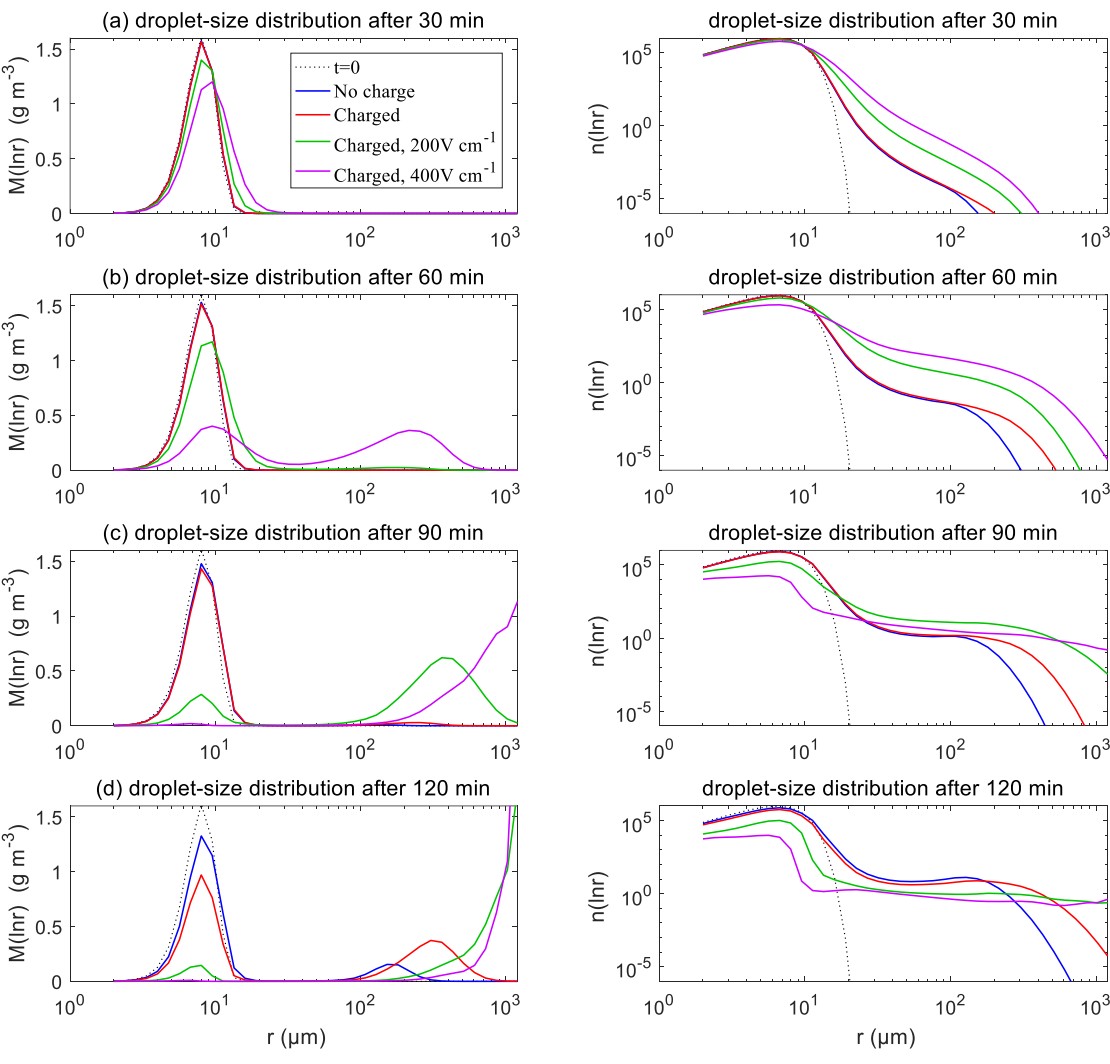

**FIG. 14.** The evolution of the droplet size distribution with initial $\bar{r} = 6.5$ μm.

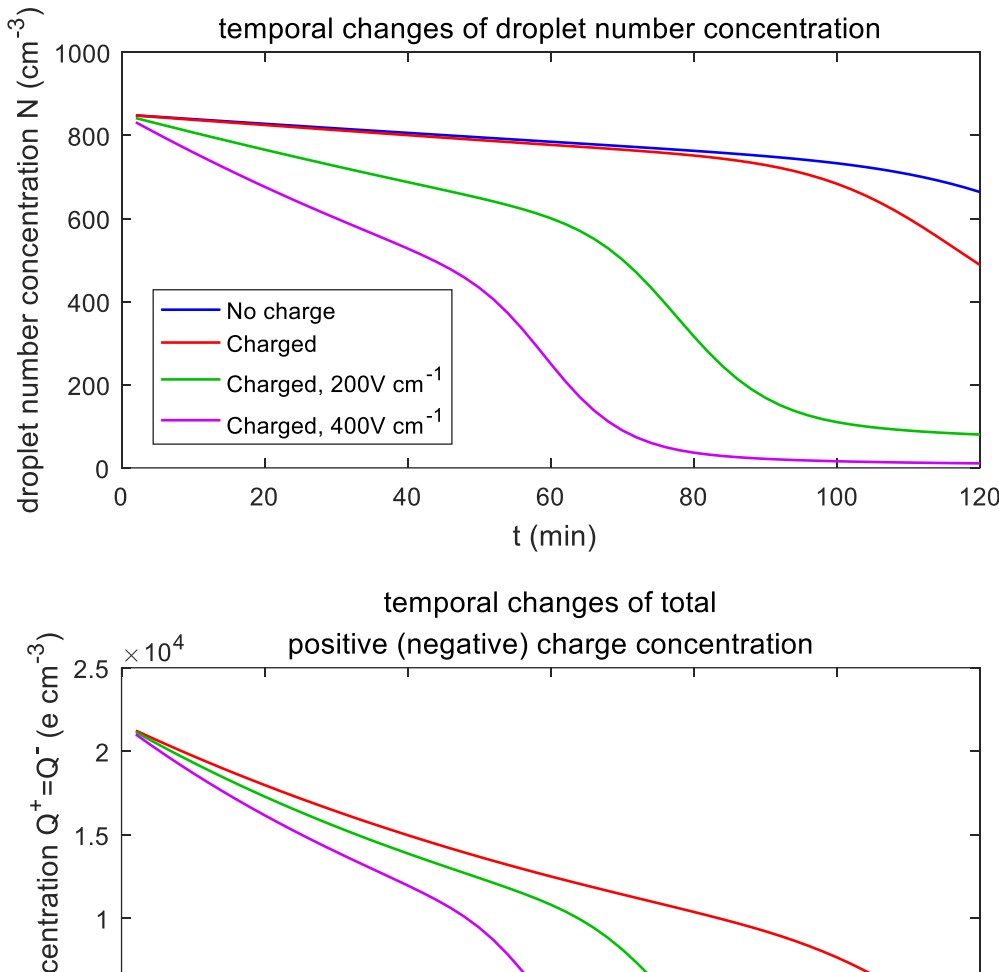

720

**FIG. 15.** Temporal changes of droplet total number concentration and total charge content for $\bar{r} = 6.5$ μm.

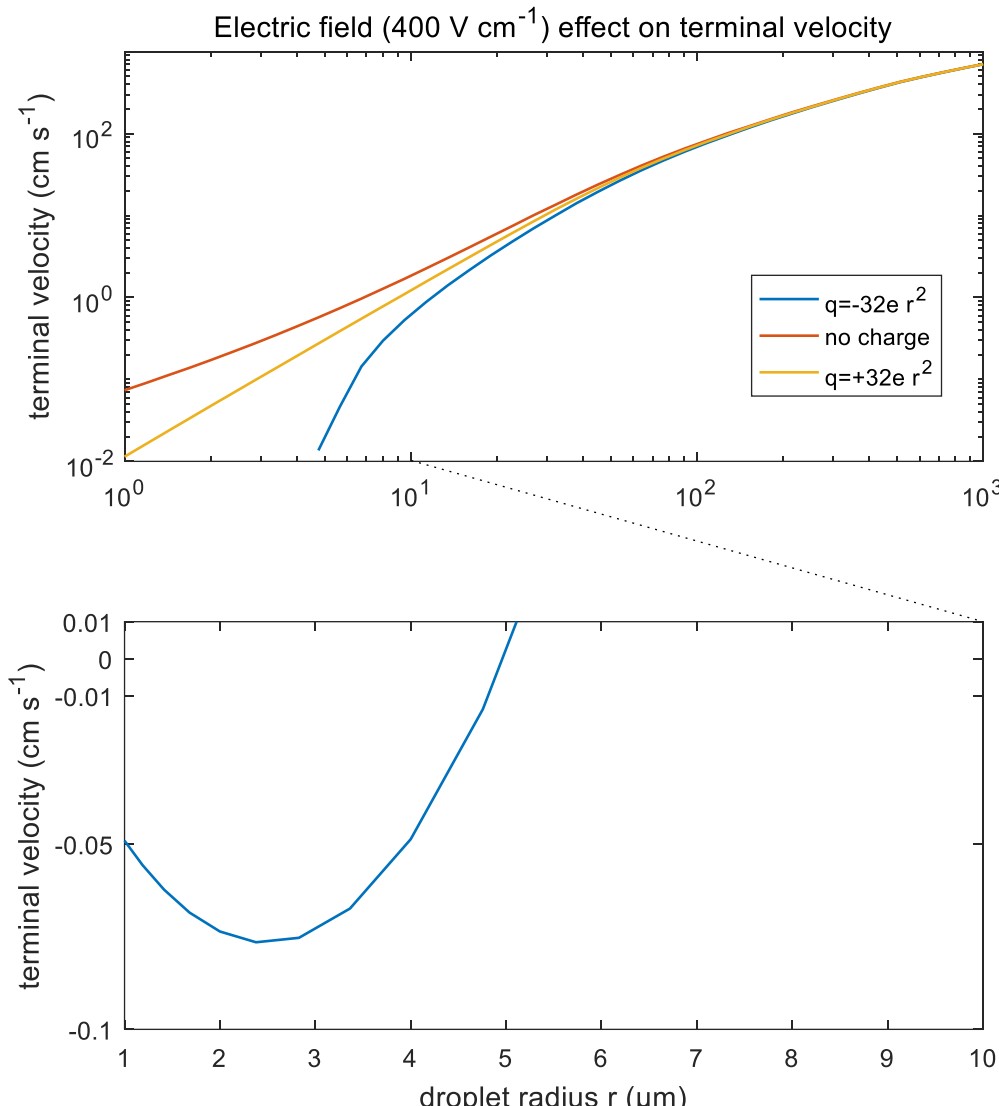

**FIG. 16.** Terminal velocities of droplets in an external electric field 400 V cm$^{-1}$. Different lines denote different droplet charge conditions. It is significant that terminal velocity of negatively charged droplets smaller than 5 μm would turn upwards, which leads to the discontinuity of the lower curve in the figure.