# Peer review of "The enhancement of droplet collision by electric charges and atmospheric electric fields"

_Atmospheric Chemistry and Physics, 2019_

## Referee Comment (RC1) · Anonymous Referee #1 · 4 Apr 2020

Comments to the manuscript with ID "acp-2019-1140"

General comments: This manuscript investigated the effect of electric charges and atmospheric electric fields on the size distribution of cloud droplets numerically. The authors concluded that electric charges and fields enhance the collision efficiency of small droplets. My main concern of the manuscript is the novelty. As far as I understand, the manuscript does not specify clearly how different the study is from the one of Khain et al, 2004. The novelty should be stated clearly in the abstract and conclusion as well as in the introduction. Especially, the introduction needs to be improved substantially. This manuscript can be improved if the authors can summarize the open

questions in previous studies and address them in their study. By such a treatment, the authors can place their contribution in a more general context. Overall, this manuscript does not satisfy the novelty requirement of the ACP journal. Major revision is needed before it can be considered for publication.

Main comments, 1. The authors concluded that electric charges and fields enhance the collision efficiency of small droplets. Is this new in the cloud physics field? If so, how different this study is compared with the one of Khain et al, 2004? Which open question does this manuscript address? The third paragraph (starting from Line 35) of the introduction part summarized the work of Khain et al, 2004, but didn't bring up the open question in Khain et al, 2004. 2. The main conclusion of the manuscript is that electric charges and electric fields enhance the collision efficiency of small droplets pairs. The evolution of droplet size distribution with different initial radius is shown in Fig.7, 8, 10. To compare the evolution for different initial radii, I would suggest the authors to plot the size distributions in one plot at a single snapshot, i.e., plot r/r0 at x-axis and n(x,t=15 min) of different r0 at y-axis in one plot. This can help clearly demonstrate the conclusion. 3. The authors mentioned the Navier-Stokes (N-S) equation just above Eq.5. if you consider the backreaction from droplets to the flow, you can add the backreaction term to the NS equation. I don't see immediately why solving the N-S equation numerically with a low Reynolds number is difficult in this study. 4. How can I see from the terminal velocity curve in Fig.11 that the 5-um size droplet turns upwards?

Specific comments: I would suggest the authors improve the English writing of this manuscript carefully across the paper. One way to improve the readability is to read the manuscript more carefully before submit it. 1. Could it be an idea to use "droplet size distribution" instead of "droplet spectrum/spectra" so that readers from a different background (physics, astrophysics) can understand it? As you don't do any Fourier transform, right? What does the spectrum/spectra mean here? 2. L10: a pairà pairs 3. L12: the cloud à clouds. Please read through the paper and check if the same

revision is needed. 4. L22: in unit of um. 5. L30: "this method" is unclear. 6. L36: used Stokes flow to represent. 7. L43: SoàTherefore. 8. L56: meansà represents. 9. L69: you already defined "\epsilon" just below Eq.2. So, the first sentence is a repetition and is misleading. You may also consider merge the two paragraphs, where E and \epsilon are discussed. Also, could you provide the expression of \epsilon? 10. L73: usedàadopted 11. L85: What about "Momentum equation droplets"? Could you go through the paper and check "motion equation"? In physics, it is "the equation of motion". 12. L88: the flow drag 13. L92: velocity vectorà velocity. You may remove "relative to the earth". 14. L95: What does "The fluid property is treated as air" mean? 15. L100: I don't understand this paragraph. Do you mean that there are no droplet-droplet interactions? In English, it is very are to put two nouns together in a sentence. You may read through the paper and try to rewrite those, which can help improve the readability of the manuscript. 16. L105: The nomenclature of the Reynolds number is unique here. It is "Re". How do you define your Reynolds number here? I know in some atmospheric books, "N_Re" was invented. 17. L115: a function. 18. L131: a complex mathematical problem in physics 19. L146: the sign. 20. L147: it is obvious that. 21. L169: are not included. 22. L171: In thunderstorm conditions. 23. L173: approachesà is close to 24. L176: to the certain mass binsà to mass bins 25. L239: by a factor of about 26. L249: evolution of the droplet size distribution 27. L291: nearly notàhardly 28. L291: and differenceà and the one 29. L294: to the observation. Can you add the reference as well? 30. L326: Do you mean "the observed atmospheric conditions"? What does "real" mean here?

---

## Referee Comment (RC2) · Anonymous Referee #2 · 13 Apr 2020

Comments to the manuscript with ID "acp-2019-1140" General comments: This manuscript studied the effect of electric charges and atmospheric electric fields on collision efficiency and the size distribution of cloud droplets numerically. The author concluded that electric charges and fields could accelerate large-drop formation in natural conditions, particularly for clouds with small droplet size. In my opinion, the manuscript is not acceptable for publication in its present form. Some major corrections should be done to make sure that the results can be more appropriate. Main points: 1) There are some errors in Eq. (3). The second term of the right hands of Eq. (3) should be the loss of droplets of mass m, however, the collection kernel is about droplets of mass mx and mass m-mx. 2) Equation (7) describe the induced flow

field u, however, Eq. (7) dose not satisfy no-slip boundary for two interacting droplets. Specifically, in the superimposed induced flow field according Eq. (7), the fluid velocity on the surface of the droplet is not equal to the velocity of the droplet. The detailed description paper of the theory was published in Journal of the Atmospheric Sciences in 2005 (https://journals.ametsoc.org/doi/full/10.1175/JAS3397.1). 3) Fig. 4 gives the initial spectrum mass distribution in 2D grids of bin. For charged clouds, the initial charge is distributed symmetrically, as shown in Fig. 4b: 14% with charge +1r2, 14% with charge -1r2, 22% with charge +0.5r2, 22% with charge -0.5r2, and 28% with no charge. What is the principle determining the abovementioned charge ratio? Is there any observation data to prove the charge ratio?

---

## Referee Comment (RC3) · Anonymous Referee #3 · 16 Apr 2020

A review of the manuscript acp-2019-1140 "The enhancement of droplet collision by electric charges and atmospheric electric fields" authored by Shian Guo and Huiwen Xue.

The authors numerically investigate collisions of charged cloud droplets and rain drops, accounting for influence of atmospheric electric field. For this purpose, they first calculate collision efficiency table considering gravity force, drag forces and electrical forces acting on drops in course of their interaction. Corresponding drop motion equations are formulated using superposition method and are integrated using second order Runge-Kutta method. Then authors solve stochastic collection equation (SCE) for 2D drop

size distribution (DSD), where the first independent variable is drop mass and the second the drop charge. SCE is solved for various initial DSDs, charges and electric field strengths. Authors conclude that "electric field could significantly enhance the collision process" in the case when the initial DSD is given in the range of small cloud droplets. I would like to note that theory and methods used by the authors in their research are not new (all the needed references are given in the study). Nevertheless, the results obtained in the study are of interest so I recommend the manuscript for publication in ACP after major revision.

1. The English language of the manuscript is of a very low quality. Please find a way to enhance it in order to render the text more readable and comprehensible. 2. It is worth explaining in the introduction how charges appear in cloudy drops 3. Line 123: Suddenly, the concept of "no-slip boundary conditions" appear. To explain. 4. To illustrate Eqs. (11) and (12) by a figure. To show directions of all the forces acting on drops and the velocities of the drops. 5. Is it right that appear in the Eq. (12)? 6. Line 153: actually, you integrate the system of 12 (or 8) equations 7. Line 187: Eq. (13) is the exponential distribution and not the gamma distribution. 8. How did you obtain Eq. (15) from Eq. (14)? 9. The authors should check the correctness of equations (13-15). 10. Lines 195-200. To add information about number concentration, liquid water content and charge content for all initial drop spectra. 11. Line 216: Please add the figure showing collision efficiency between cloud droplets (1-20 ïA■m in radii), the same as in fig. 5. It is all the more important because you obtained the maximal effect for cloud droplets. 12. Figure 6: Please, add illustrations for different collectors (say 15 ïA■m, and 10 ïA■m in radii) and comment them. 13. Section 5.2: Please show temporal changes of drop concentration and charge content and comment on them. How fast the charges of opposite signs compensate each other? 14. Line 288: "The relative terminal velocity term also contributes to the collection kernel, and the electric field can affect terminal velocity of small charged droplets significantly." – Please, cover this issue in more detail in the article.

---

## Author Comment (AC2) · 11 Jun 2020

**Response to Referee #2**

**General comments:**

**This manuscript studied the effect of electric charges and atmospheric electric fields on collision efficiency and the size distribution of cloud droplets numerically. The author concluded that electric charges and fields could accelerate large-drop formation in natural conditions, particularly for clouds with small droplet size. In my opinion, the manuscript is not acceptable for publication in its present form. Some major corrections should be done to make sure that the results can be more appropriate.**

**Main points:**

**1. There are some errors in Eq. (3). The second term of the right hands of Eq. (3) should be the loss of droplets of mass m, however, the collection kernel is about droplets of mass mx and mass m-mx.**

Response: Thanks for the reviewer's careful reading. We have changed the following expression in the second term on the right hand of Eq. (3):

$$K(m_x, q_x; m-m_x, q-q_x)$$

To

$$K(m_x, q_x; m, q)$$

**2. Equation (7) describes the induced flow field u, however, Eq. (7) does not satisfy no-slip boundary for two interacting droplets. Specifically, in the superimposed induced flow field according Eq. (7), the fluid velocity on the surface of the droplet is not equal to the velocity of the droplet. The detailed description paper of the theory was published in Journal of the Atmospheric Sciences in 2005 (https://journals.ametsoc.org/doi/full/10.1175/JAS3397.1).**

Response: Thanks for the reviewer's comment. We have made the corrections. The error was made when we typed the equations. Actually, the equations in our computer program are correct and satisfy no-slip boundary condition. We have made a thorough check to the equations. Lines 103 to 114 in the original manuscript now reads as:

The formulas to compute the flow velocity $\boldsymbol{u}$ are discussed in this section. We consider a rigid sphere moving with a velocity $U$ relative to the viscous fluid. It is known that when the Reynolds number is small, the Stokes flow gives a concise solution of the stream function

$$\psi_s = U\left(\frac{1}{4\tilde{R}} - \frac{3\tilde{R}}{4}\right)\sin^2\theta \tag{5a}$$

where $\tilde{R} = R/r$ is the normalized distance ($R$ is the distance from the sphere centre, $r$ is the droplet radius), $\theta$ is the angle between the droplet's velocity and vector $\boldsymbol{R}$ pointing from the sphere centre. $U$ is the droplet velocity relative to the fluid, i.e., $U_1 = |\boldsymbol{v_1} - \boldsymbol{u_2}|$ for droplet 1, and $U_2 = |\boldsymbol{v_2} - \boldsymbol{u_1}|$ for droplet 2. However, this stream function of the Stokes flow does not apply to the system with a large Reynolds number. It is needed to approximatively construct a stream function which depends on Reynolds number $N_{Re} = \frac{2r v \rho}{\mu}$, where $\rho$ is

the density of the air, and $\mu$ is the dynamic viscosity of the air. Hamielec and Johnson (1962, 1963) gave the stream function $\psi_h$ induced by a moving rigid sphere, for large Reynolds numbers:

$$\psi_h = U\left(\frac{A_1}{\tilde{R}} + \frac{A_3}{\tilde{R}^2} + \frac{A_3}{\tilde{R}^3} + \frac{A_4}{\tilde{R}^4}\right)\sin^2\theta - U\left(\frac{B_1}{\tilde{R}} + \frac{B_3}{\tilde{R}^2} + \frac{B_3}{\tilde{R}^3} + \frac{B_4}{\tilde{R}^4}\right)\sin^2\theta\cos\theta \qquad (5b)$$

where $A_1, \ldots, B_4$ are functions only of Reynolds number $N_{Re}$ for each droplet. The method is valid for $N_{Re} < 5000$. But the solution deviates from the Stokes flow solution when $N_{Re} \to 0$ for small droplets. Therefore, this work adopts a smooth combination of $\psi_h$ and Stokes stream function $\psi_s$ (Pinsky and Khain, 2000)

$$\psi = \frac{N_{Re}\psi_h + N_{Re}^{-1}\psi_s}{N_{Re} + N_{Re}^{-1}} \qquad (6)$$

which converges to stokes flow when $N_{Re} \to 0$. Then the induced flow field $\boldsymbol{u}$ is derived,

$$\boldsymbol{u} = -\frac{1}{\tilde{R}^2\sin\theta}\frac{\partial\psi}{\partial\theta}\hat{\boldsymbol{e}}_R + \frac{1}{\tilde{R}\sin\theta}\frac{\partial\psi}{\partial\tilde{R}}\hat{\boldsymbol{e}}_\theta = u_R\hat{\boldsymbol{e}}_R + u_\theta\hat{\boldsymbol{e}}_\theta \qquad (7a)$$

where $\hat{\boldsymbol{e}}_R$ and $\hat{\boldsymbol{e}}_\theta$ are unit vectors in the polar coordinate $(R, \theta)$. It can also be expressed in the Cartesian coordinate (x, z) as:

$$\boldsymbol{u} = (u_R\cos\varphi - u_\theta\sin\varphi)\hat{\boldsymbol{e}}_z + (u_R\sin\varphi + u_\theta\cos\varphi)\hat{\boldsymbol{e}}_x \qquad (7b)$$

where the direction of $\hat{\boldsymbol{e}}_z$ is downward, the same as gravity. $\varphi$ is the angle between the droplet's velocity and $\hat{\boldsymbol{e}}_z$.

Both Stokes and Hamielec stream functions satisfy the no-slip boundary condition, i.e., the fluid velocity on the surface of the droplet is equal to the velocity of the droplet. Hamielec stream function is no-slip because those functions $A_1, \ldots, B_4$ in Eq. (10) satisfy $A_1 + 2A_2 + 3A_3 + 4A_4 = 1$ and $B_1 + 2B_2 + 3B_3 + 4B_4 = 0$, as long as the droplet is considered as a rigid sphere (Hamielec, 1963). These relations ensure that $u_\theta = -U\sin\theta$ at the surface of the droplet. Note that $u_\theta$ is the velocity of the fluid at the surface, and $U\sin\theta$ is the tangential velocity of the droplet surface. This equation ensures the no-slip boundary condition.

In our study, the adopted stream function is a linear combination of Stokes flow and Hamielec (1963) flow field, because the latter one works well for a wide range of Reynolds numbers up to $10^3$. Both Hamielec and Stokes stream functions satisfy the no-slip boundary condition.

In addition, the superposition method used in our study does accord with Eq. (19) and (20) in Wang and Ayala's paper in 2005 (https://journals.ametsoc.org/doi/full/10.1175/JAS3397.1) based on the Stokes flow.

Reference:
Wang, L. P., Ayala, O., Grabowski, W. W.: Improved Formulations of the Superposition Method, J. Atmos. Sci, 62(4):1255-1266, doi: 10.1175/JAS3397.1, 2005

**3. Fig. 4 gives the initial spectrum mass distribution in 2D grids of bin. For charged clouds, the initial charge is distributed symmetrically, as shown in Fig. 4b: 14% with charge $+1r^2$, 14% with charge $-1r^2$, 22% with charge $+0.5r^2$, 22% with charge $-0.5r^2$, and 28% with no charge. What is the principle determining the abovementioned charge ratio? Is there any observation data to prove the charge ratio?**

Response: We thank the reviewer for raising these questions. The ratio in the original manuscript is an approximation of 2:3:4:3:2, but it is arbitrarily chosen. The basic idea was to let the droplets distribution over charge bins mimic a normal distribution, and also to satisfy electric neutrality $\bar{q} = 0$.

In fact, there are some observations on mean charges of droplets, as can be seen in Figure 1 below (from Pruppacher and Klett, 1997). But there is no observational data for the kind of charge ratio that we used. Now we use a Gaussian distribution in the revised manuscript to describe the droplet distribution over the charge bins.

Lines 199-202 have been revised and it reads as follows in the revised manuscript:

To simulate an early stage of the warm-cloud precipitation, we need to distribute the droplets in each size bin to different charge bins, so that these droplets have different charges. Since there is little data on this, we assume a Gaussian distribution,

$$N(q) = \frac{N}{\sqrt{2\pi}\sigma} \exp\left(-\frac{q^2}{2\sigma^2}\right)$$

where $N$ is the number concentration in the size bin, and $\sigma$ is the standard deviation of the Gaussian distribution in that bin. $N(q)$ represents the number concentration of droplets with charge $q$. This distribution satisfies electric neutrality $\bar{q} = 0$. For different size bin, droplet number concentration $N$ is different. We purposely set the standard deviation $\sigma$ to be different for different size bins. For larger size, the charge amount is larger, based on $\overline{|q|}$ = 1.31 $r^2$ (q in unit of elementary charge and r in μm) as stated in the Introduction. Therefore, we set larger standard deviation $\sigma$ for the larger size bins.

[Figure]

596                                      CHAPTER 17

Fig. 17-2.   Mean absolute electric charge on cloud and raindrops. Round symbols indicate warm cloud cases, triangular symbols indicate thunderstorm cases; solid symbols indicate negative charge, open symbols indicate positive charge. PK Phillips and Kinzer (1958), S Sergieva (1959), CR Colgate and Romero (1970), TW Twomey (1956), TC Takahashi and Craig (1973), T1 Takahashi (1965), T2 Takahashi (1972), TF Takahashi and Fullerton (1972), G1 Gunn (1949), G2 Gunn (1950). (Adapted with changes from Takahashi (1973).)

**Figure 1(Appendix).** Observational data for the relationship between droplet charge and radius

(Pruppacher and Klett 1997). Our new setting $\overline{|q|}$ = 1.31 $r^2$ (q in unit of elementary charge and r in μm) approaches line (4) around $r \approx 10$μm, which is the weakly electrified warm cloud case.

New simulations using the Gaussian charge distribution have been performed. Figs. 4, 7-10 in the original manuscript are now replaced with the new simulations, and comparisons are shown below. But the changes in results are not significant. Therefore, discussions in section 5.2 basically remain unchanged.

[Figure]

(Figure 4 in the original manuscript)
**Figure 4.** The initial droplet mass distributed over the size and charge bins. Colours stand for water mass content in the bins (in unit of g m$^{-3}$). (a) Uncharged droplets (b) charged droplets.

[Figure]

(Figure 4 in the revised manuscript)

**Figure 4.** The initial droplet mass distributed over the size and charge bins. Colours stand for water mass content in the bins (in unit of g m$^{-3}$). (a) Uncharged droplets (b) charged droplets.

[Figure]

(Figure 8 in the original manuscript)

**FIG. 8.** The evolution of the droplet size distribution with initial $\bar{r} = 9$ μm.

[Figure]

(Figure 8 in the new manuscript)

**FIG. 8 (it is Figure 10 now).** The evolution of the droplet size distribution with initial $\bar{r} = 9$ μm.

---

## Author Comment (AC3) · 11 Jun 2020

**Response to Referee #3**

**The authors numerically investigate collisions of charged cloud droplets and rain drops, accounting for influence of atmospheric electric field. For this purpose, they first calculate collision efficiency table considering gravity force, drag forces and electrical forces acting on drops in course of their interaction. Corresponding drop motion equations are formulated using superposition method and are integrated using second order Runge-Kutta method. Then authors solve stochastic collection equation (SCE) for 2D drop size distribution (DSD), where the first independent variable is drop mass and the second the drop charge. SCE is solved for various initial DSDs, charges and electric field strengths. Authors conclude that "electric field could significantly enhance the collision process" in the case when the initial DSD is given in the range of small cloud droplets. I would like to note that theory and methods used by the authors in their research are not new (all the needed references are given in the study). Nevertheless, the results obtained in the study are of interest so I recommend the manuscript for publication in ACP after major revision.**

**1. The English language of the manuscript is of a very low quality. Please find a way to enhance it in order to render the text more readable and comprehensible.**

Response: Thanks for the reviewer's comment on the writing of the paper. We have substantially revised the whole manuscript. The Introduction of the manuscript is completely rewritten. Most part of the Results, Abstract, and Conclusion are rewritten. Description of the methods are now improved in writing. The whole manuscript is now more organized and more readable. We have also checked the grammar throughout the manuscript. Grammar errors and unclear sentences are all changed. In the response to reviewer's 2nd comment, we show part of the rewritten Introduction to summarize the electrification process in clouds. In the response to reviewer's 12th comment, we show the rewritten section 5.1, where the electrostatic effects on collision efficiency is discussed.

**2. It is worth explaining in the introduction how charges appear in cloudy drops.**

Response: Yes, it is necessary to explain the charging process in clouds. At the beginning of the rewritten Introduction, we use two paragraphs to explain the electrification in thunderstorms and in warm clouds. These paragraphs now read as:

Clouds are usually electrified (Pruppacher and Klett 1997). For thunderstorms, several theories of electrification have been proposed in the past decades. The proposed theories assume that the electrification involves the collision of graupel or hailstones with ice crystals or supercooled cloud droplets, based on radar observational result that the onset of strong electrification follows the formation of graupel or hailstones within the cloud (Wallace and Hobbs, 2006). However, the exact conditions and mechanisms are still under debate. One charging process could be due to the thermoelectric effect between the rimed and relatively warm graupel or hailstones with the relatively cold ice crystals or supercooled cloud droplets. Another charging process could be due to the polarization of particles by the downward atmospheric electric field. The thunderstorm electrification can increase the electric fields to several thousand V cm$^{-1}$, while the magnitude of electric fields in fair

weather air is only about 1 V cm$^{-1}$ (Pruppacher and Klett 1997). Droplet charges can reach $|q| \approx 42r^2$ in unit of elementary charge in thunderstorms, with the droplet radius $r$ in unit of µm according to observations (Takahashi, 1973).

Liquid stratified clouds do not have such strong charge generation as in the thunderstorms. But charging of droplets can indeed occur at the upper and lower cloud boundaries as the fair weather current passes through the clouds (Harrison et al. 2015, Baumgaertner et al. 2014). The global fair weather current and the electric field are in the downward direction. Given the electric potential of 250 kV for the ionosphere, the exact value of fair weather current density over a location depends on the electric resistance of the atmospheric column, but its typical value is about $2 \times 10^{-12}$ A m$^{-2}$ (Baumgaertner et al. 2014). The fair weather electric field is typically about 1 V cm$^{-1}$ in the cloud-free air, but is usually much stronger inside stratus clouds, because the cloudy air has a lower electrical conductivity than the cloud-free air. There is a conductivity transition at cloud boundaries. Therefore, the cloud top is positively charged and the cloud base is negatively charged. Based on the in situ measurements of charge density in liquid stratified cloud, and assuming that the cloud has a droplet number concentration on the order of 100 cm$^{-3}$, it is estimated that the mean charge per droplet is +5e (ranging from +1e to +8e) at cloud top, and -6e (ranging from -1e to -16e) at cloud base. Other studies found different amount of charges in clouds. According to Tsutomu Takahashi (1973) and Khain (1997), the mean absolute charge of droplets in warm clouds is around $|q| \approx$ 6.6 r$^{1.3}$ (with units of e and µm for q and r, respectively). For a droplet with radii of 10 µm, it is about 131 e.

New references:

Tsutomu Takahashi: Measurement of electric charge of cloud droplets, drizzle, and raindrops, Reviews of Geophysics and Space Physics,11, 903-924, 1973
Harrison, R. G., Nicoll, K. A., Ambaum, M. H. P.: On the microphysical effects of observed cloud edge charging, Q. J. R. Meteorol. Soc., 141, 2690–2699, doi:10.1002/qj.2554, 2015
Baumgaertner, A. J. G., Lucas, G. M., Thayer, J. P., Mallios, S. A.: On the role of clouds in the fair weather part of the global electric circuit, Atmos. Chem. Phys., 14, 8599–8610, doi:10.5194/acp-14-8599-2014, 2014
Wallace, J. M., Hobbs, P. V.: Atmospheric Science, Second Edition, Academic Press, 2006

**3. Line 123: Suddenly, the concept of "no-slip boundary conditions" appear. To explain.**

Response: Thanks for the reviewer to point this out. An explanation is added to the manuscript. Before line 115, the following paragraph is added. (In the revised manuscript, $U$ is the droplet velocity relative to the fluid.)

Both Stokes and Hamielec stream functions satisfy the no-slip boundary condition, which means that the fluid at the droplet boundary has zero velocity relative to the droplet. Hamielec stream function is no-slip because those functions $A_1, ..., B_4$ in Eq. (5b) satisfy $A_1 + 2A_2 + 3A_3 + 4A_4 = 1$ and $B_1 + 2B_2 + 3B_3 + 4B_4 = 0$, as long as the droplet is considered as a rigid sphere (Hamielec, 1963). These relations ensure that $u_\theta = -U \sin \theta$ at the surface of the droplet. Note that $u_\theta$ is the velocity of fluid at the surface, and $U \sin \theta$ is the tangential velocity of the droplet surface. The two velocities are equal, which ensures the no-slip boundary condition.

**4. To illustrate Eqs. (11) and (12) by a figure. To show directions of all the forces acting on drops and the velocities of the drops.**

Response: Thanks for the reviewer's comment. Equation (11) represents the force acting on droplet 2, due to the charge of droplet 1 and the external electric field. We now changed the order of the last two terms in the first line of equation (11), so that the first three terms in the first line represent the force in the radial direction due to the external electric field, and the fourth term in the first line represents the force in the radial direction due to the charge of droplet 1. The second line of Equation (11) represents the force in the tangential direction due to the electric field.
The equation now is:

$$\boldsymbol{F}_{e2} = \{r_2{}^2 E_0^2 (F_1 \cos^2 \theta + F_2 \sin^2 \theta) + E_0 \cos\theta (F_3 q_1 + F_4 q_2) + E_0 q_2 \cos\theta + \frac{1}{r_2^2}(F_5 q_1{}^2 + F_6 q_1 q_2 + F_7 q_2{}^2)\}\hat{\boldsymbol{e}}_R$$

$$+ \{r_2{}^2 E_0^2 F_8 \sin 2\theta + E_0 \sin\theta (F_9 q_1 + F_{10} q_2) + E_0 q_2 \sin\theta\}\hat{\boldsymbol{e}}_\theta$$

When the electric field $E_0$ is zero, the equation is reduced to

$$\boldsymbol{F}_{e2} = \frac{1}{r_2^2}(F_5 q_1{}^2 + F_6 q_1 q_2 + F_7 q_2{}^2)\hat{\boldsymbol{e}}_R$$

This equation describes the force due to the charge in droplet 1. We have added the above new equation to the manuscript. In Figure 2 of the original manuscript, the force from the conductor model is indeed based on the equation above. In Figure 2, this conductor model is compared with the inverse-square law as described by Equation (10).

The figure below shows the forces acting on droplet 1 and droplet 2, and the velocities of the droplets. It has been added to the revised manuscript as Figure 3. These forces are terms on the right hand side of Eq. 4, including gravity force, flow drag force, and electrostatic force. Note that the electrostatic force $\boldsymbol{F}_{e1}$, $\boldsymbol{F}_{e2}$ include two parts: the electric force from the other droplet ($F_{inter}$ in the figure), and the force from the external electric field ($E_0 q_1$ in the figure). When considering the two droplets as a system, the electric force from the other droplet can be considered as internal force. The velocity of droplet 2 is usually not straightly downward, because it tends to follow the streamlines when approaching with droplet 1.

[Figure]

Flow drag on drop 1

external electric field
$\boldsymbol{E_0}$

$\boldsymbol{v_1}$

$+\boldsymbol{F}_{\text{inter}}$

$\boldsymbol{E_0}\,\boldsymbol{q_1}$

Flow drag on drop 2

$-\boldsymbol{F}_{\text{inter}}$

$\boldsymbol{G_1} = m_1\boldsymbol{g}$

$\boldsymbol{v_2}$

$\boldsymbol{E_0}\,\boldsymbol{q_2}$

$\boldsymbol{G_2} = m_2\boldsymbol{g}$

The resultant electric force acting on each droplet:
$\boldsymbol{F}_{e1} = \boldsymbol{E_0}\,\boldsymbol{q_1} + \boldsymbol{F}_{\text{inter}}$
$\boldsymbol{F}_{e2} = \boldsymbol{E_0}\,\boldsymbol{q_2} - \boldsymbol{F}_{\text{inter}}$

**5. Is it right that appear in the Eq. (12)?**

Response: Yes it is right. We should have emphasized that $\boldsymbol{F}_{e1}$ and $\boldsymbol{F}_{e2}$ consist not only the electric force from the other droplet, but also the force from the external electric field. As mentioned in the response to reviewer's 4th comment, the order of two terms in Equation (11) is changed, so that it is easier to identify the force due to the external electric field and the force due to the charge in the other droplet. Because the two droplets can be considered as a system, the sum of the forces they experience independently ($\boldsymbol{F}_{e1} + \boldsymbol{F}_{e2}$) must be equal to the external electric force acting on the system $E_0(q_1 + q_2)\hat{\boldsymbol{e}}_z$. This relation is expressed in equation (12). If we have already known $F_{e2}$, then $F_{e1}$ is derived immediately from Eq. (12). In line 140 of the original manuscript, we have made some changes in the writing of the manuscript to explain this.

**6. Line 153: actually, you integrate the system of 12 (or 8) equations**

Response: Thanks for the reviewer's careful reading. We have added this information to the manuscript based on reviewer's comment.

**7. Line 187: Eq. (13) is the exponential distribution and not the gamma distribution.**

Response: Yes Eq. (13) is the exponential distribution. Thanks to the reviewer for pointing this out. We have made the correction in the manuscript. We should mention that Eq. (14) and (15) are gamma distributions. Please refer to the response to the 8th comment for detailed discussions on how to obtain Eq. (15).

**8. How did you obtain Eq. (15) from Eq. (14)?**

Response:

Thanks for this question. We now have added more information to this part, so that it is easier to understand the equations for size distribution. Basically, definitions of the size distribution is used for the derivation. Recall that $n(m)$ is the droplet number concentration per unit mass interval, and $M(m)$ is the mass concentration per unit mass interval.

The distribution of droplet number concentration $n(m)$ can also be written as $n(r)$, or $n(\ln r)$. We know that the definition of $n(m)$ is: $n(m) = dN/dm$, where $dm$ is the mass interval, and $dN$ is the droplet number concentration in that mass interval. $n(r) = dN/dr$ represents the droplet number concentration per unit size interval. $n(\ln r) = dN/d \ln r$ represents droplet number concentration per unit interval of logarithmic size. Similarly, the distribution of droplet mass concentration $M(m)$ can be written as $M(r)$, and $M(\ln r)$. These functions are related together.
$M(\ln r)$ and $M(r)$ are related through:
$$M(\ln r) = dM/d \ln r = r \cdot dM/dr = r \cdot M(r)$$
While $M(r)$ can be related with $M(m)$ through:
$$M(m) = \frac{dM}{dm} = \frac{1}{4\pi r^2} \frac{dM}{dr} = \frac{M(r)}{4\pi r^2}$$
With $m = 4\pi r^3 \rho/3$, and assuming that $\bar{m} = 4\pi \bar{r}^3 \rho/3$, where $\bar{r}$ is the mean radius, we can obtain $M(\ln r)$ from $M(m)$,

$$M(\ln r) = 3L \frac{r^6}{\bar{r}^6} exp\left(-\frac{r^3}{\bar{r}^3}\right)$$

In the revised manuscript, we added a new equation for $n(\ln r)$, because $n(\ln r)$ is also plotted and discussed in the Results section.

$$n(\ln r) = L \frac{9r^3}{4\pi \bar{r}^6} exp\left(-\frac{r^3}{\bar{r}^3}\right)$$

**9. The authors should check the correctness of equations (13-15).**

Response: Thanks for the suggestion. We have checked the equations to make sure they are correct.

**10. Lines 195-200. To add information about number concentration, liquid water content and charge content for all initial drop spectra.**

Response: The initial liquid water content is set to be $L=1$ g m$^{-3}$, for all simulations. This is a typical value in warm clouds. The initial averaged droplet radius $\bar{r}$ is set to be 15 μm, 9 μm and 6.5 μm, where $\bar{r} =$ 15 μm case represents the clean conditions (less aerosol), and 6.5 μm represents polluted conditions (more aerosol). These settings give an initial droplet number concentration of 70, 325, and 850 cm$^{-3}$, respectively. The charge content is set as in the following table. The number concentration and charge content for all initial drop size distribution are shown in table 2 in the revised manuscript.

**Table** 2. Total number concentration, charge content for the initial droplet size distribution

| mean radius $\bar{r}$ (μm) | number concentration N (cm$^{-3}$) | total positive charge Q$^+$ (e cm$^{-3}$) | total negative charge Q$^-$ (e cm$^{-3}$) |
|---|---|---|---|
| 15 | 70.6 | +9384 | -9384 |
| 9 | 324.8 | +15638 | -15638 |
| 6.5 | 850.5 | +21634 | -21634 |

Note that the initial droplet number concentration is distributed into different size bins and different charge bins. The size distribution is based on functions described in Equations (13)-(15). The charge distribution is now based on a Gaussian distribution in the revised manuscript, instead of the method described in lines 200-202 in the original manuscript. Ratios shown in lines 200-202 in the original manuscript is an approximation of 2:3:4:3:2, but it is arbitrarily chosen to mimic a normal distribution, and also to satisfy electric neutrality $\bar{q} = 0$. In the revised manuscript, we use a Gaussian distribution to describe droplet distribution over the charge bins.

Lines 199-202 have been revised and it reads as follows in the revised manuscript:

To simulate an early stage of the warm-cloud precipitation, we need to distribute the droplets in each size bin to different charge bins, so that these droplets have different charges. Since there is little data on this, we assume a Gaussian distribution,

$$N(q) = \frac{N}{\sqrt{2\pi}\sigma} \exp\left(-\frac{q^2}{2\sigma^2}\right)$$

where $N$ is the number concentration in the size bin, and $\sigma$ is the standard deviation of the Gaussian distribution in that bin. $N(q)$ represents the number concentration of droplets with charge $q$. This distribution satisfies electric neutrality $\bar{q} = 0$. For different size bin, droplet number concentration $N$ is different. We purposely set the standard deviation $\sigma$ to be different for different size bins. For larger size, the charge amount is larger, based on $\overline{|q|} = 1.31 \, r^2$ (q in unit of elementary charge and r in μm) as stated in the Introduction. Therefore, we set larger standard deviation $\sigma$ for the larger size bins.

**11. Line 216: Please add the figure showing collision efficiency between cloud droplets (1-20 μm in radii), the same as in fig. 5. It is all the more important because you obtained the maximal effect for cloud droplets.**

Response: As suggested, we plot a new figure below, to show the collision efficiencies for smaller collectors ($r_1$ = 5, 10, 15, 20 and 25 μm) when the droplet pairs have no charge. X-axis denotes the ratio of radius r$_2$/r$_1$. As will be seen in the response to reviewer's 12$^{th}$ comment, the 10 μm and 20 μm lines will be shown together with the results for charged droplets (new Fig. 6). Therefore, we think this figure is not necessary in the manuscript.

[Figure]

**12. Figure 6: Please, add illustrations for different collectors (say 15 ïA¿ m, and 10 ïA¿ m in radii) and comment them.**

Response: Thank you very much for the suggestion. As suggested, we have shown the different collectors $r_1$= 10, 20, 30 and 40 μm in the new figure below. Fig. 6 now describes the collision efficiency for the 30 and 40 μm collectors (precipitating droples). Fig. 7 now describes the collision efficiency for the 10 and 20 μm collectors (cloud droplets). Therefore, section 5.1 has been substantially revised. Most part of it has been rewritten. It is clear that electrostatic effects are significant for small droplets. We show the rewritten section 5.1 here:

**5.1 Collision efficiency**

[revised manuscript text omitted]

**13. Section 5.2: Please show temporal changes of drop concentration and charge content and comment on them. How fast the charges of opposite signs compensate each other?**

Response: Thanks for the reviewer's suggestion. The evolution of droplet concentration and charge

content are shown in the below. These figures are also added to the manuscript as new Fig. 9, 11, and 14.

From Fig. 9 ($\bar{r}$ =15 μm), it is evident that droplet concentrations in the 4 different electric conditions decrease from about 70 cm$^{-3}$ to less than 5 cm$^{-3}$, and the evolution is nearly not affected by the electric conditions. The electrostatic effect is therefore negligible in this case.

From Fig. 11 ($\bar{r}$ =9 μm), we can see the evolution is distinctly affected by the 4 different electric conditions. Electric charges and fields play an important role in converting smaller droplets to larger droplets, and decreasing the droplet number concentration.

From Figure 14 ($\bar{r}$ =6.5 μm), droplet concentration is strongly affected by the 4 different electric conditions. Results show that the electric field would remarkably trigger the collision-coalescence process for the small droplets.

Comparing the upper and lower panels of each figure, it is evident that the charges of opposite signs compensate each other as fast as the decrease of number concentration (except for the uncharged case). The phases of charge neutralizations are the same as changes of drop concentration. In all the three figures, more than 90% charges of opposite signs are neutralized during the evolution.

[Figure]

[Figure]

**FIG. 9.** Temporal changes of droplet total number concentration and total charge content for $\bar{r} = 15$ μm.

[Figure]

[Figure]

**FIG. 11.** Temporal changes of droplet total number concentration and total charge content for $\bar{r} = 9$ μm

[Figure]

**FIG. 14.** Temporal changes of droplet total number concentration and total charge content for $\bar{r} = 6.5$ μm.

**14. Line 288: "The relative terminal velocity term also contributes to the collection kernel, and the electric field can affect terminal velocity of small charged droplets significantly." – Please, cover this issue in more detail in the article.**

Response: Thanks for the reviewer's suggestion. We have improved the writing of this part
Lines 287-295 "The electric enhancement of…" have been revised to:
However, the relative terminal velocity term also contributes to the collection kernel, which is shown in Eq. (2). As mentioned above, terminal velocities $V_1$ or $V_2$ are derived by simulating just single one charged droplet in air with certain electric field, and letting it fall until its velocity converges to the terminal velocity. Therefore, the electric field can affect terminal velocities of charged droplets, thus to affect the collection kernels. Terminal velocities of droplets in an external electric field is illustrated in Fig. 15. In downwards electric field 400 V cm$^{-1}$, terminal velocity of a large droplet is

hardly affected. The difference of velocity caused by electric field at $r = 1000$ μm does not exceed 1%, and the one at 100 μm does not exceed 5%. On the contrary, electric fields strongly affect the terminal velocities of charged small droplets. For r < 5 μm, the terminal velocities of negative-charged droplets even turn "upwards", namely the electric field lifts them up in the air. Electric fields mainly affect terminal velocities of small charged droplets, because droplet mass $m \propto r^3$, while droplet charge $q \propto r^2$ according to observation. So, $q \propto m^{2/3}$ means that acceleration contributed by electric force decreases with increasing droplet mass, which explain that the terminal velocity of small charged droplets is more sensitive to the electric field.

In Fig.11 of the original manuscript, y-axis is in logarithmic scale and stands for the absolute value of terminal velocity, which is ambiguous. In the revised manuscript, we plot the negative terminal velocity in a separate panel, as shown below. (The whole manuscript has been revised substantially, so it becomes Fig. 15 now)

[Figure]

Figure 15. Terminal velocities of droplets in an external electric field 400 V cm$^{-1}$. Different lines denote different droplet charge conditions. It is seen that the terminal velocity of negatively-charged droplets

smaller than 5 μm would turn upwards, which leads to the discontinuity of the lower curve in the figure.

---

## Author Response (AR1)

Dear editor,

Thank you very much for the editorial handling of this manuscript. According to the comments of the three referees, we have made a substantial revision to the manuscript. Some of the paragraphs have been rewritten. The important changes are:

1. The Introduction has been completely rewritten. Cloud electrification is now introduced in more detail, as suggested by referee #3. The physical mechanism and the effects of electrostatic induction are now written in a more organized way. The novelty of this manuscript is also emphasized, as suggested by referee #1.

2. In Section 2, the kernel in the stochastic collection equation for uncharged droplets and charged droplets are now explained in more detail.

3. In section 3.2, the formulas of stream functions and flow fields have been corrected, as suggested by referee #2. The explanations of them have been rewritten. In the previous version, same symbols in section 3.2 and 3.3 are used to represent different variables. This is now corrected. We now introduce $\tilde{R} = R/r$, $\theta_0 = \theta - \varphi$ to make sure different variables are represented by different symbols in Section 3.2 and 3.3.

4. In Section 3.3, a new figure (Fig. 2) is added to show the forces acting on each droplet, the velocity of each droplet, and the induced flow velocity, as suggested by referee #3. Detailed discussion is also added to the manuscript.

5. Section 4 has been divided into 3 subsections. The explanations of model settings and initial droplet size distributions have been revised, and they are much more understandable now. The initial setting of droplet charge distribution now uses a more appropriate model, as response to referee #2's comment.

6. In section 5.1, new figures (Fig. 7b and Fig. 8) are added to show collision efficiencies for various collector sizes, including collectors with sizes typical of cloud droplets, as suggested by referee #3.

7. In Section 5.2, we have added new figures (Figs. 10, 12, and 15) to show the temporal changes of droplet total number concentration and total charge concentration during the collision-coalescence process, as suggested by referee #3.

8. In Section 6, the axis of the Fig. 16 has been changed to explicitly show the change of the terminal velocity, as response to referee #1's comment.

Furthermore, the grammatical errors and inappropriate expressions have been corrected. We have checked the languages of the revised manuscript. Attached are the responses to the comments of the three referees, and a marked-up version of the revised manuscript. In the responses to referees' comments, the sentences in blue are responses to the referee, and the sentences in red are revised texts of the manuscript. In the marked-up version of the revised manuscript, words and sentences in blue are contents added in the revision of our manuscript, and those in red with strikethrough are contents which have been deleted.

Thank you very much and we look forward to your reply.

Sincerely,
Shian Guo, Huiwen Xue

**Response to Referee #1**

**General comments:**

**This manuscript investigated the effect of electric charges and atmospheric electric fields on the size distribution of cloud droplets numerically. The authors concluded that electric charges and fields enhance the collision efficiency of small droplets. My main concern of the manuscript is the novelty. As far as I understand, the manuscript does not specify clearly how different the study is from the one of Khain et al, 2004. The novelty should be stated clearly in the abstract and conclusion as well as in the introduction. Especially, the introduction needs to be improved substantially. This manuscript can be improved if the authors can summarize the open questions in previous studies and address them in their study. By such a treatment, the authors can place their contribution in a more general context. Overall, this manuscript does not satisfy the novelty requirement of the ACP journal. Major revision is needed before it can be considered for publication.**

Response:

We thank the reviewer for pointing out that the novelty of this study should be more addressed in the manuscript. The Introduction of the manuscript is now completely rewritten. Now the Introduction summarizes the previous work on cloud electrification, the physical mechanism of the electrostatic induction, the effect of electrostatic induction on droplet collision efficiency, and the subsequent effect on precipitation formation. Now the rewritten Introduction is shown below in red fonts. Abstract and Conclusion have also been substantially revised, but not shown here. The novelty has been emphasized in all these sections.

This study is motivated as the aerosol-cloud interaction study regarding climate change has been widely carried out. It has been confirmed by both observational studies and modeling studies that increased aerosols can result in more numerous but smaller droplets, hence slower collision-coalescence process, and suppressed warm-rain precipitation process. Since cloud electrification has been found for both thunderstorms and warm clouds, and electrification can increase the possibility of collision-coalescence, as described in the revised Introduction of this manuscript, it is worthy of investigating whether the electrostatic effects can mitigate the aerosol effects. This kind of study has not been performed. Previous studies of electrostatic effect such as Khain et al. (2004) focuses on weather modification, including rain enhancement and fog elimination. Here we are interested in finding out to what extent the electrostatic effect can mitigate the aerosol effect.

To investigate the electrostatic effect vs. aerosol effect on droplet collision-coalescence, we purposely choose an initial droplet size distribution function based on Bott (1998), i.e., Equation 13 in the original manuscript. This distribution function has two parameters: liquid water content and averaged size of droplets. We set the liquid water content as constant ($1 \text{ g m}^{-3}$) and vary the averaged size of droplets in the initial size distribution ($\bar{r}$ = 15, 9, and 6.5 μm) to represent the effect of aerosols on cloud microphysics. These settings give an initial droplet number concentration of 70, 325, and 850 $\text{cm}^{-3}$, respectively. As suggested by Reviewer #3, description of droplet number concentration is added to the manuscript. The electrostatic effect is then investigated for the three cases.

Here is a simple example to compare the electrostatic effect vs. the aerosol effect: When there is no

electric charge and field, the case with initial $\bar{r}$ = 15 μm can develop a significant second peak in the size distribution through collision-coalescence in less than 30 min, while it takes about 60 min for the $\bar{r}$ = 9 μm case to develop a similar second peak. This represents an aerosol effect. When considering the electric charge and field effects, it only takes about 45 min for the $\bar{r}$ = 9 μm case to develop a similar second peak (as can be seen in Figs. 7 and 8 in the original manuscript). The aerosol-induced precipitation suppression effect is mitigated by the electrostatic effects. We emphasize on this issue in various places in the revised manuscript.

The Introduction now reads as:

[revised manuscript text omitted]

**Main Comments**

**1. The authors concluded that electric charges and fields enhance the collision efficiency of small droplets. Is this new in the cloud physics field? If so, how different this study is compared with the one of Khain et al, 2004? Which open question does this manuscript address? The third paragraph (starting from Line 35) of the introduction part summarized the work of Khain et al, 2004, but didn't bring up the open question in Khain et al, 2004.**

Response:

Thanks to the reviewer for asking these questions. It is not new that the electric charges and fields enhance collision efficiency of small droplets. Studies of Khain et al. (2004) and Harrison et al. (2015) already had this finding. In our study, we intend to compare the precipitation suppression effect due to increased aerosols and the electrostatic enhancement effect. We have revised the manuscript to emphasize on this issue.

Regarding the difference between our study and Khain et al. (2004), the two studies are different in many aspects. Firstly, Khain et al. (2004) focuses on justifying cloudy seeding via artificial charging process, for use in weather modification, while our study investigates to what extent the electrostatic effect mitigates the aerosol effect on the evolution of droplet size distribution. Secondly, the amount of electric charges on cloud droplets are extremely large in their study, and natural clouds probably do not meet that condition. In our study, however, the amount of electric charges and fields used in our study represent conditions in natural clouds such as warm clouds or the early stage of thunderstorms. Thirdly, simplified models are used for the electric force between charged droplets and for describing droplet motion in Khain et al. (2004). Our study uses more accurate models for electric force and droplet motions. Our study finds that electric charges and fields can accelerate precipitation under conditions in the real atmosphere and that the aerosol-induced precipitation suppression can be mitigated.

Reference:

Harrison, R. G., Carslaw, K. S.: Ion-Aerosol-Cloud Processes in the Lower Atmosphere, Rev. Geophys., 41(3), doi:10.1029/2002RG000114, 2003

**2. The main conclusion of the manuscript is that electric charges and electric fields enhance the collision efficiency of small droplets pairs. The evolution of droplet size distribution with different initial radius is shown in Fig.7, 8, 10. To compare the evolution for different initial radii, I would suggest the authors to plot the size distributions in one plot at a single snapshot, i.e., plot $r/r_0$ at x-axis and $n(x, t=15 \text{ min})$ of different $r_0$ at y-axis in one plot. This can help clearly**

**demonstrate the conclusion.**

Response:

We tried to plot droplet size distributions as suggested by the reviewer. The figures are shown below. The main problem is that collision-coalescence is significantly slowed down in the smaller $r_0$ cases. Therefore the time ($t_0$) required for a second peak to form in the size distribution is quite different for different $r_0$. For the three cases in this study, the time $t_0$ is about 30, 60 and 120 min, respectively. We use a normalized time, namely $t/t_0$, for 5 snapshots. Because both the radius and the time are normalized, information shown in the figures are not very straightforward. Therefore we prefer that Figures 7, 8 and 10 remain unchanged. (Now they are Figs. 9, 11, and 14 in the revised manuscript).

[Figure]

Figure 1. The evolution of normalized droplet size distributions. X-axis denotes the normalized droplet size $r/r_0$, where $r_0$=15, 9 and 6.5 μm separately. Different panels show different snapshots, i.e., at different normalized time $t/t_0$, where $t_0$=30, 60 and 120 min separately. Comparisons are made between uncharged droplets and charged droplets without electric fields.

[Figure]

Figure 2. The evolution of normalized droplet size distributions. Comparisons are made between uncharged droplets and charged droplets with an electric field of 200 V cm⁻¹.

[Figure]

Figure 3. The evolution of normalized droplet size distributions. Comparisons are made between uncharged droplets and charged droplets with an electric field of 400 V cm$^{-1}$.

**3. The authors mentioned the Navier-Stokes (NS) equation just above Eq.5. if you consider the backreaction from droplets to the flow, you can add the backreaction term to the NS equation. I don't see immediately why solving the N-S equation numerically with a low Reynolds number is difficult in this study.**

Response:

We now realized that the sentence where N-S equation are mentioned is very misleading. In the revised manuscript, we have deleted this sentence in line 103 "Considering a sphere moving in a viscous fluid, the exact solution of the induced flow velocity field is to solve the Navier-Stokes equations. But the computation is too complicated in this study."

Solving the N-S equation is not difficult. However, the computation burden for the problem in this study would be heavy. With 37 size bins and 15 charge bins, the number of collision efficiency is on the order of 37×37×15×15. For each collision efficiency, about 10$^4$ steps are needed, including using the bisection method. It takes several days of computer time to derive all the collision efficiencies using the current method. Solving the N-S equation would be a much heavier computation burden.

**4. How can I see from the terminal velocity curve in Fig.11 that the 5-um size droplet turns**

**upwards?**

Response:

Thanks for pointing this out. In Fig.11 of the original manuscript, y-axis is in logarithmic scale and stands for the absolute value of terminal velocity. Negative terminal velocity means upward motion. However, minus is not compatible with the logarithmic coordinate. We therefore plotted the absolute value of terminal velocity in the figure (new Fig.16).

In the revised manuscript, we plot the negative terminal velocity in a separate panel, as shown below.

[Figure]

Figure 16. Terminal velocities of droplets in an external electric field 400 V cm$^{-1}$. Different lines denote different droplet charge conditions. It is seen that the terminal velocity of negatively-charged droplets smaller than 5 μm would turn upwards, which leads to the discontinuity of the lower curve in the figure.

**Specific comments:**

**I would suggest the authors improve the English writing of this manuscript carefully across the paper. One way to improve the readability is to read the manuscript more carefully before submit it.**

Response: Thank you very much for pointing this out. We have made substantial changes to the manuscript. The Introduction is completely rewritten. Most parts of Results and some descriptions of Methods are also rewritten. The writing of the paper is much more organized now.

**1. Could it be an idea to use "droplet size distribution" instead of "droplet spectrum/spectra" so that readers from a different background (physics, astrophysics) can understand it? As you don't do any Fourier transform, right? What does the spectrum/spectra mean here?**

Response: As the reviewer suggested, we have changed all the "spectrum/spectra" to "size distribution" in the manuscript, including text in figures. And it is true that we do not do any Fourier transform.

**2. L10: a pair -> pairs.** Changed.

**3. L12: the cloud -> clouds. Please read through the paper and check if the same revision is needed.**
Changed and checked.

**4. L22: in unit of um.** We have corrected all of them.

**5. L30: "this method" is unclear.**
The sentence has been changed to "Schlamp et al. (1976) used the model of Davis (1964) to study the effect of …".

**6. L36: used Stokes flow to represent.** Changed.

**7. L43: So -> Therefore.** Changed.

**8. L56: means -> represents.** Changed.

**9. L69: you already defined "/epsilon" just below Eq.2. So, the first sentence is a repetition and is misleading. You may also consider merge the two paragraphs, where E and /epsilon are discussed. Also, could you provide the expression of /epsilon?**
Response:
Thanks for raising this question. Both E and $\varepsilon$ are discussed in details now. We have revised line 59 "…and $\varepsilon$ is the coalescence efficiency" to "Collision efficiency $E(m_1, m_2)$ and coalescence efficiency $\varepsilon(m_1, m_2)$ are introduced to the kernel because not all the droplets in this volume will have collision-coalesce with the collector." The first sentence of line 69 has been deleted, and the whole paragraph of lines 69-73 has been changed to:
Two droplets may not coalesce even when they collide with each other. Observations show that the droplet pair can rebound in some cases, because of an air film temporally trapped between the two surfaces. Especially for droplets with radii both larger than 100 μm, the coalescence efficiency is remarkably less than 1.0. Beard and Ochs (1984) provides a formula of coalescence efficiency for a certain range of droplet radii. Basically coalescence efficiency is a function of the sizes of the two droplets in their formula.

As for the expression of $\varepsilon$, it is just an empirical law (Beard and Ochs, 1984)

$$\varepsilon = (a - b)^{\frac{1}{3}} - (a + b)^{\frac{1}{3}} + 0.459$$

$$a = (b^2 + 0.00441)^{\frac{1}{2}}$$

$$b = 0.0946\beta - 0.319$$
$$\beta = \ln(r_2/\mu m) + 0.44\ln(r_1/200\mu m)$$

**10. L73: used -> adopted.** Corrected.

**11. L85: What about "Momentum equation droplets"? Could you go through the paper and check "motion equation"? In physics, it is "the equation of motion".**

Response: Thanks for pointing this out. We have gone through the paper and correct the following sentences.
Line 85: "Droplet motion equation" is changed to "Equations of motion for droplets"
Line 87: "In order to get the collision efficiency, the motion equation of droplets is integrated to get the trajectories of droplets" is changed to "In order to get the collision efficiency between a pair of droplets, the equations of motion are integrated to get the trajectories of the two droplets."
Line 89: "The motion equations for a pair of droplets…" is changed to "The equations of motion for a pair of droplets…"
Line 310: "The motion equation of droplets in the atmosphere is solved…" is changed to "The equations of motion for droplets in the atmosphere are solved…"

**12. L88: the flow drag.** Changed.

**13. L92: velocity vector -> velocity. You may remove "relative to the earth".** Changed.

**14. L95: What does "The fluid property is treated as air" mean?**
Response: We have changed "The fluid property is treated as air with temperature…" to
We set air temperature $T = 283$ K and pressure $p = 900$ hPa in this study for the calculation of air viscosity.

**15. L100: I don't understand this paragraph. Do you mean that there are no droplet-droplet interactions? In English, it is very rare to put two nouns together in a sentence. You may read through the paper and try to rewrite those, which can help improve the readability of the manuscript.**

Response: Thank the reviewer for raising these concerns. Actually, the "superposition method" is a term in many papers of cloud physics, including our references. We should make a detailed explanation. This paragraph is revised and is more comprehensible:

The flow drag force is described by the second term on the right side of Eq. (4), which assumes a simple hydrodynamic interaction of the two droplets. That is, each droplet moves in the flow field induced by the other one moving alone, and it is called "superposition method" in cloud physics. This method has been successfully used in many researches of the calculation of collision efficiencies (Pruppacher and Klett, 1997). To calculate the flow drag force, the induced flow field $\boldsymbol{u}$ is required. The method for obtaining the induced flow field $\boldsymbol{u}$ is discussed below.

**16. L105: The nomenclature of the Reynolds number is unique here. It is "Re". How do you define your Reynolds number here? I know in some atmospheric books, "N_Re" was invented.**

Response: Actually $N_{Re}$ is widely used. We chose to use this instead of $Re$ because $Re$ can be misleading when it appears in an equation, especially in an equation like Eq. 6 in the manuscript. The two letters in $Re$ can be mistakenly thought as distance $R$ and elementary charge $e$.
The Reynolds number $N_{Re}$ is defined as

$$N_{Re} = \frac{2rv\rho}{\mu}$$

in line 193 of the revised manuscript, when it appears for the first time.

**17. L115: a function.** Changed.
**18. L131: a complex mathematical problem in physics.** Changed.
**19. L146: the sign.** Changed.
**20. L147: it is obvious that.** Changed.
**21. L169: are not included.** Changed.
**22. L171: In thunderstorm conditions.** Changed.
**23. L173: approaches -> is close to.** Changed.
**24. L176: to the certain mass bins -> to mass bins.** Changed.
**25. L239: by a factor of about.** Changed.
**26. L249: evolution of the droplet size distribution.** Changed.
**27. L291: nearly not -> hardly.** Changed.
**28. L291: and difference -> and the one.** Changed.

**29. L294: to the observation. Can you add the reference as well?**

Response: As suggested, we have added "according to Tsutomu Takahashi (1973) and Pruppacher and Klett (1997)" after line 294 "…to the observation". The results of observation in several previous researches are shown in Chapter 17.4.2.1 of Pruppacher and Klett (1997) .

Reference:
Tsutomu Takahashi: Measurement of electric charge of cloud droplets, drizzle, and raindrops, Reviews of Geophysics and Space Physics,11, 903-924, 1973

**30. L326: Do you mean "the observed atmospheric conditions"? What does "real" mean here?**

Response: Yes. As suggested, we have changed line 326 "…represent the real conditions in the atmosphere" to "…represent the observed atmospheric conditions."

**Response to Referee #2**

**General comments:**

**This manuscript studied the effect of electric charges and atmospheric electric fields on collision efficiency and the size distribution of cloud droplets numerically. The author concluded that electric charges and fields could accelerate large-drop formation in natural conditions, particularly for clouds with small droplet size. In my opinion, the manuscript is not acceptable for publication in its present form. Some major corrections should be done to make sure that the results can be more appropriate.**

**Main points:**

**1. There are some errors in Eq. (3). The second term of the right hands of Eq. (3) should be the loss of droplets of mass m, however, the collection kernel is about droplets of mass mx and mass m-mx.**

Response: Thanks for the reviewer's careful reading. We have changed the following expression in the second term on the right hand of Eq. (3):

$$K(m_x, q_x; m-m_x, q-q_x)$$

To

$$K(m_x, q_x; m, q)$$

**2. Equation (7) describes the induced flow field u, however, Eq. (7) does not satisfy no-slip boundary for two interacting droplets. Specifically, in the superimposed induced flow field according Eq. (7), the fluid velocity on the surface of the droplet is not equal to the velocity of the droplet. The detailed description paper of the theory was published in Journal of the Atmospheric Sciences in 2005**
 **(https://journals.ametsoc.org/doi/full/10.1175/JAS3397.1).**

Response: Thanks for the reviewer's comment. We have made the corrections. The error was made when we typed the equations.
Actually, the equations in our computer program are correct and satisfy no-slip boundary condition. We have made a thorough check for Section 3.2. We found that the dimensions of these stream functions are wrong in the original manuscript. We also used same symbols for different variables in Section 3.2 and 3.3 in the original manuscript. Now in the revised manuscript, we introduce $\tilde{R} = R/r$, $\theta_0 = \theta - \varphi$ to make sure different variables are represented by different symbols in Section 3.2 and 3.3. $\theta$ is replaced by $\theta_0$ in section 3.2 because $\theta$ also appears in section 3.3 and represents a different variable. Their relation is $\theta_0 = \theta - \varphi$, where $\theta$ is the angle between the downward vector $\hat{e}_z$ and the line connecting the centres of two droplets, and $\varphi$ is the angle between $\hat{e}_z$ and the droplet's velocity $v$. The coefficient of Stokes flow has also been corrected in section 3.2.
Lines 103 to 114 in the original manuscript now reads as:

Considering a rigid sphere moving in a viscous fluid with a velocity $U$ relative to the flow, the steam function depends on Reynolds number, $N_{Re} = \frac{2rv\rho}{\mu}$, where $\rho$ is the density of the air, and $\mu$ is the dynamic viscosity of the air. It is known that when Reynolds number is small, the flow is considered as Stokes flow and the stream

function can be expressed as

$$\psi_s = U\left(\frac{1}{4\tilde{R}} - \frac{3\tilde{R}}{4}\right)\sin^2\theta_0 \tag{5}$$

where $\tilde{R} = R/r$ is the normalized distance ($R$ is the distance from the sphere centre, and $r$ is the droplet radius), $\theta_0$ is the angle between the droplet velocity and vector $\boldsymbol{R}$ pointing from the sphere centre. $U$ is droplet velocity relative to the flow, i.e., $U_1 = |\boldsymbol{v_1} - \boldsymbol{u_2}|$ for droplet 1, and $U_2 = |\boldsymbol{v_2} - \boldsymbol{u_1}|$ for droplet 2. However, this stream function for Stokes flow does not apply to the system with a large Reynolds number. Hamielec and Johnson (1962, 1963) gave the stream function $\psi_h$ induced by a moving rigid sphere, which can be used for flows with large Reynolds numbers:

$$\psi_h = U\left(\frac{A_1}{\tilde{R}} + \frac{A_3}{\tilde{R}^2} + \frac{A_3}{\tilde{R}^3} + \frac{A_4}{\tilde{R}^4}\right)\sin^2\theta_0 - U\left(\frac{B_1}{\tilde{R}} + \frac{B_3}{\tilde{R}^2} + \frac{B_3}{\tilde{R}^3} + \frac{B_4}{\tilde{R}^4}\right)\sin^2\theta_0\cos\theta_0 \tag{6}$$

where $A_1, \ldots, B_4$ are functions only of Reynolds number $N_{Re}$ for each droplet. The method is valid for $N_{Re} < 5000$. But the solution deviates from the Stokes flow solution when $N_{Re} \to 0$ for small droplets. Therefore, it is needed to construct a stream function that applies to a wide range of $N_{Re}$. This work adopts a stream function that is a linear combination of $\psi_h$ and Stokes stream function $\psi_s$ (Pinsky and Khain, 2000)

$$\psi = \frac{N_{Re}\psi_h + N_{Re}^{-1}\psi_s}{N_{Re} + N_{Re}^{-1}} \tag{7}$$

which converges to stokes flow when $N_{Re} \to 0$. Then the induced flow field $\boldsymbol{u}$ is derived,

$$\boldsymbol{u} = -\frac{1}{\tilde{R}^2\sin\theta_0}\frac{\partial\psi}{\partial\theta_0}\hat{\boldsymbol{e}}_R + \frac{1}{\tilde{R}\sin\theta_0}\frac{\partial\psi}{\partial\tilde{R}}\hat{\boldsymbol{e}}_\theta = u_R\hat{\boldsymbol{e}}_R + u_\theta\hat{\boldsymbol{e}}_\theta \tag{8}$$

where $\hat{\boldsymbol{e}}_R$ and $\hat{\boldsymbol{e}}_\theta$ are unit vectors in the polar coordinate $(R, \theta_0)$. It can also be expressed in the Cartesian coordinate (x, z)

$$\boldsymbol{u} = (u_R\cos\varphi - u_\theta\sin\varphi)\hat{\boldsymbol{e}}_z + (u_R\sin\varphi + u_\theta\cos\varphi)\hat{\boldsymbol{e}}_x \tag{9}$$

where the direction of $\hat{\boldsymbol{e}}_z$ is vertically down, the same as gravitation. $\varphi$ is the angle between $\hat{\boldsymbol{e}}_z$ and the droplet velocity $\boldsymbol{v}$.

Both Stokes and Hamielec stream functions satisfy the no-slip boundary condition, i.e., the fluid velocity on the surface of the droplet is equal to the velocity of the droplet. Hamielec stream function is no-slip because those functions $A_1, \ldots, B_4$ in Eq. (6) satisfy $A_1 + 2A_2 + 3A_3 + 4A_4 = 1$ and $B_1 + 2B_2 + 3B_3 + 4B_4 = 0$, as long as the droplet is considered as a rigid sphere (Hamielec, 1963). These relations ensure that $u_\theta = -U\sin\theta_0$ at the surface of the droplet, which means the no-slip boundary condition. (Note that $u_\theta$ is the velocity of the fluid at droplet surface, and $U\sin\theta_0$ is the tangential velocity of the droplet surface.)

In our study, the adopted stream function is a linear combination of Stokes flow and Hamielec (1963) flow field, because the latter one works well for a wide range of Reynolds numbers up to $10^3$. Both Hamielec and Stokes stream functions satisfy the no-slip boundary condition.

In addition, the superposition method used in our study does accord with Eq. (19) and (20) in Wang and Ayala's paper in 2005 (https://journals.ametsoc.org/doi/full/10.1175/JAS3397.1) based on the Stokes flow.

To simulate an early stage of the warm-cloud precipitation, we need to distribute the droplets in each size bin to different charge bins, so that these droplets have different charges. Since there is little data on this, we assume a Gaussian distribution,

$$N(q) = \frac{N_0}{\sqrt{2\pi}\sigma} \exp\left(-\frac{q^2}{2\sigma^2}\right) \tag{19}$$

where $N_0$ is the number concentration in the size bin, and $\sigma$ is the standard deviation of the Gaussian distribution in that size bin. $N(q)$ represents the number concentration of droplets with charge $q$. This distribution satisfies electric neutrality $\bar{q} = 0$ . For different size bin, droplet number concentration $N_0$ is different. We purposely set the standard deviation $\sigma$ to be different for different size bins. For a larger size, the charge amount is larger, based on $\overline{|q|}$ = 1.31 r² (q in unit of elementary charge and r in μm) as stated in the Introduction. Therefore, we set larger standard deviation $\sigma$ for the larger size bins. With this setting of droplet charge, the total amount of charge in each case is shown in Table 1. The $\bar{r} =$ 15, 9, and 6.5 μm cases have an initial charge concentration of 9438, 15638, and 21634 e cm⁻³, respectively, for both positive charge and negative charge.

[Figure]

Fig. 17-2.   Mean absolute electric charge on cloud and raindrops. Round symbols indicate warm cloud cases, triangular symbols indicate thunderstorm cases; solid symbols indicate negative charge, open symbols indicate positive charge. PK Phillips and Kinzer (1958), S Sergieva (1959), CR Colgate and Romero (1970), TW Twomey (1956), TC Takahashi and Craig (1973), T1 Takahashi (1965), T2 Takahashi (1972), TF Takahashi and Fullerton (1972), G1 Gunn (1949), G2 Gunn (1950). (Adapted with changes from Takahashi (1973).)

**Appendix 1.** Observational data for the relationship between droplet charge and radius (Pruppacher and Klett 1997). Our new setting $\overline{|q|}$ = 1.31 $r^2$ (q in unit of elementary charge and r in μm) approaches line (4) around $r \approx 10$μm, which is the weakly electrified warm cloud case.

New simulations using the Gaussian charge distribution have been performed. Figs. 4, 7-10 in the original manuscript are now replaced with the new simulations, and comparisons are shown below. But the changes in results are not significant.

[Figure]

(Figure 4 in the original manuscript)

**Figure 4.** The initial droplet mass distributed over the size and charge bins. Colours stand for water mass content in the bins (in unit of g m$^{-3}$). (a) Uncharged droplets (b) charged droplets.

[Figure]

(Figure 4 in the revised manuscript)

**Figure 4 (it is Figure 5 now).** The initial droplet mass distributed over the size and charge bins. Colours stand for water mass content in the bins (in unit of g m$^{-3}$). (a) Uncharged droplets (b) charged droplets.

[Figure]

(Figure 8 in the original manuscript)

**FIG. 8.** (it is Figure 9 now.) The evolution of the droplet size distribution with initial $\bar{r}$ = 9 μm.

[Figure]

(Figure 8 in the new manuscript)

**FIG. 8 (it is Figure 11 now)**. The evolution of the droplet size distribution with initial $\bar{r} = 9$ µm.

**Response to Referee #3**

**The authors numerically investigate collisions of charged cloud droplets and rain drops, accounting for influence of atmospheric electric field. For this purpose, they first calculate collision efficiency table considering gravity force, drag forces and electrical forces acting on drops in course of their interaction. Corresponding drop motion equations are formulated using superposition method and are integrated using second order Runge-Kutta method. Then authors solve stochastic collection equation (SCE) for 2D drop size distribution (DSD), where the first independent variable is drop mass and the second the drop charge. SCE is solved for various initial DSDs, charges and electric field strengths. Authors conclude that "electric field could significantly enhance the collision process" in the case when the initial DSD is given in the range of small cloud droplets. I would like to note that theory and methods used by the authors in their research are not new (all the needed references are given in the study). Nevertheless, the results obtained in the study are of interest so I recommend the manuscript for publication in ACP after major revision.**

**1. The English language of the manuscript is of a very low quality. Please find a way to enhance it in order to render the text more readable and comprehensible.**

Response: Thanks for the reviewer's comment on the writing of the paper. We have substantially revised the whole manuscript. The Introduction of the manuscript is completely rewritten. Most part of the Results, Abstract, and Conclusion are rewritten. Description of the methods are now improved in writing. The whole manuscript is now more organized and more readable. We have also checked the grammar throughout the manuscript. Grammar errors and unclear sentences are all changed. In the response to reviewer's 2$^{nd}$ comment, we show part of the rewritten Introduction to summarize the electrification process in clouds. In the response to reviewer's 12$^{th}$ comment, we show the rewritten section 5.1, where the electrostatic effects on collision efficiency is discussed.

**2. It is worth explaining in the introduction how charges appear in cloudy drops.**

Response: Yes, it is necessary to explain the charging process in clouds. At the beginning of the rewritten Introduction, we use two paragraphs to explain the electrification in thunderstorms and in warm clouds. These paragraphs now read as:

Clouds are usually electrified (Pruppacher and Klett 1997). For thunderstorms, several theories of electrification have been proposed in the past decades. The proposed theories assume that the electrification involves the collision of graupel or hailstones with ice crystals or supercooled cloud droplets, based on radar observational result that the onset of strong electrification follows the formation of graupel or hailstones within the cloud (Wallace and Hobbs, 2006). However, the exact conditions and mechanisms are still under debate. One charging process could be due to the thermoelectric effect between the relatively warm, rimed graupel or hailstones and the relatively cold ice crystals or supercooled cloud droplets. Another charging process could be due to the polarization of particles by the downward atmospheric electric field. The thunderstorm electrification can increase the electric fields to several thousand V cm$^{-1}$, while the magnitude of electric fields in fair weather air is only about 1 V cm$^{-1}$ (Pruppacher and Klett 1997). Droplet charges can reach $|q| \approx 42r^2$ in unit

of elementary charge in thunderstorms, with the droplet radius $r$ in unit of μm according to observations (Takahashi, 1973). For cumuli clouds, previous studies show smaller charge amount.

Liquid stratified clouds do not have such strong charge generation as in the thunderstorms. But charging of droplets can indeed occur at the upper and lower cloud boundaries as the fair weather current passes through the clouds (Harrison et al. 2015, Baumgaertner et al. 2014). The global fair weather current and the electric field are in the downward direction. Given the electric potential of 250 kV for the ionosphere, the exact value of fair weather current density over a location depends on the electric resistance of the atmospheric column, but its typical value is about $2 \times 10^{-12}$ A m$^{-2}$ (Baumgaertner et al. 2014). The fair weather electric field is typically about 1 V cm$^{-1}$ in the cloud-free air, but is usually much stronger inside stratus clouds, because the cloudy air has a lower electrical conductivity than the cloud-free air. There is a conductivity transition at cloud boundaries. Therefore, the cloud top is positively charged and the cloud base is negatively charged. Based on the in situ measurements of charge density in liquid stratified cloud, and assuming that the cloud has a droplet number concentration on the order of 100 cm$^{-3}$, it is estimated that the mean charge per droplet is +5e (ranging from +1e to +8e) at cloud top, and -6e (ranging from -1e to -16e) at cloud base (Harroson et al. 2015). According to Tsutomu Takahashi (1973) and Khain (1997), the mean absolute charge of droplets in warm clouds is around $|q| \approx 6.6 \, r^{1.3}$ (e, μm). For a droplet with radii of 10 μm, it is about 131 e.

New references:

Tsutomu Takahashi: Measurement of electric charge of cloud droplets, drizzle, and raindrops, Reviews of Geophysics and Space Physics,11, 903-924, 1973.

Harrison, R. G., Nicoll, K. A., Ambaum, M. H. P.: On the microphysical effects of observed cloud edge charging, Q. J. R. Meteorol. Soc., 141, 2690–2699, doi:10.1002/qj.2554, 2015.

Baumgaertner, A. J. G., Lucas, G. M., Thayer, J. P., Mallios, S. A.: On the role of clouds in the fair weather part of the global electric circuit, Atmos. Chem. Phys., 14, 8599–8610, doi:10.5194/acp-14-8599-2014, 2014.

Wallace, J. M., Hobbs, P. V.: Atmospheric Science, Second Edition, Academic Press, 2006.

**3. Line 123: Suddenly, the concept of "no-slip boundary conditions" appear. To explain.**

Response: Thanks for the reviewer to point this out. An explanation is added to the manuscript. Before line 115, the following paragraph is added. (In the revised manuscript, $U$ is the droplet velocity relative to the fluid.)

Both Stokes and Hamielec stream functions satisfy the no-slip boundary condition, i.e., the fluid velocity on the surface of the droplet is equal to the velocity of the droplet. Hamielec stream function is no-slip because those functions $A_1, \ldots, B_4$ in Eq. (6) satisfy $A_1 + 2A_2 + 3A_3 + 4A_4 = 1$ and $B_1 + 2B_2 + 3B_3 + 4B_4 = 0$, as long as the droplet is considered as a rigid sphere (Hamielec, 1963). These relations ensure that $u_\theta = -U \sin \theta_0$ at the surface of the droplet, which means the no-slip boundary condition. (Note that $u_\theta$ is the velocity of the fluid at droplet surface, and $U \sin \theta_0$ is the tangential velocity of the droplet surface.)

**4. To illustrate Eqs. (11) and (12) by a figure. To show directions of all the forces acting on drops and the velocities of the drops.**

Response: Thanks for the reviewer's comment. Equation (11) (now it is Eq.13) represents the electrostatic force acting on droplet 2, due to the charge of droplet 1 and the external electric field. The new figure below shows all the forces acting on droplet 1 and droplet 2, and the velocities of the droplets. It has been added to the revised manuscript as Figure 2. These forces are terms on the right hand side of Eq. 4, including gravity force, flow drag force, and electrostatic force. We changed the presentation of Equation (13) in the revised manuscript so that the electric forces acting on droplet 2 can be understood more easily.

Lines 135-146 have been revised to:

$$\boldsymbol{F}_{e2} = E_0 q_2 \cos\theta \hat{\boldsymbol{e}}_{\boldsymbol{R}} + E_0 q_2 \sin\theta \hat{\boldsymbol{e}}_{\boldsymbol{\theta}} +$$

$$\{r_2{}^2 E_0^2 (F_1 \cos^2\theta + F_2 \sin^2\theta) + E_0 \cos\theta (F_3 q_1 + F_4 q_2) + \frac{1}{r_2^2}(F_5 q_1{}^2 + F_6 q_1 q_2 + F_7 q_2{}^2)\}\hat{\boldsymbol{e}}_{\boldsymbol{R}}$$

$$+ \{r_2{}^2 E_0^2 F_8 \sin 2\theta + E_0 \sin\theta (F_9 q_1 + F_{10} q_2)\}\hat{\boldsymbol{e}}_{\boldsymbol{\theta}}$$

$$(13)$$

where $\hat{\boldsymbol{e}}_{\boldsymbol{R}}$ is the radial unit vector, and $\hat{\boldsymbol{e}}_{\boldsymbol{\theta}}$ is tangential unit vector, $\boldsymbol{E_0}$ is the eternal electric field, $q_1$ and $q_2$ are the charges of droplet 1 and droplet 2 respectively, and parameters $F_1$ to $F_{10}$ are a series of complicated functions of geometric parameters $(r_1, r_2, R$; Davis 1964).

The electric force directly from the external field is shown as the two terms in the first line of Eq. (13), and can be simply written as $\boldsymbol{E_0} q_2$ if combining the two terms. Line 2 and line 3 in Eq. (13) represent the interactive force from droplet 1 in the radial direction and tangential direction, respectively. Note that the third term in line 2 represent the interactive force from droplet 1 if there is no external electric field. Except for this term, all the other terms in lines 2 and 3 are the interactive forces from droplet 1 due the induction from the external field.

Similarly, the resultant electric force $\boldsymbol{F}_{e1}$ acting on droplet 1 includes both the force directly from the external field and the interactive force from droplet 2. The sum of the electric forces on the two droplets, $\boldsymbol{F}_{e1} + \boldsymbol{F}_{e2}$, must equal to the external electric force acting on the system, which can be expressed as $\boldsymbol{E_0}(q_1 + q_2)$, because the two droplets can be considered as a system. Then, the electric force acting on droplet 1 could be derived immediately as

$$\boldsymbol{F}_{e1} = \boldsymbol{E_0}(q_1 + q_2) - \boldsymbol{F}_{e2} \qquad (14)$$

Figure 2 is a schematic diagram showing the forces acting on each droplet in a pair. Also shown in Fig. 2 are the velocity of each droplet relative to the flow if there is no other droplets present ($\boldsymbol{v}$), and the flow velocity induced by the other droplet ($\boldsymbol{u}$). Droplet velocity relative to the flow is $\boldsymbol{v} - \boldsymbol{u}$. The electric field $\boldsymbol{E_0}$ is in the downward direction, the same as gravity. Droplet 1 has positive charge and droplet 2 has negative charge in this example. The forces acting on each droplet include gravity, flow drag force, and the electrostatic force, as seen on the right side of Eq. (4). For droplet 1, the electric force directly from the external field is in the downward direction, and is shown as $\boldsymbol{E_0} q_1$ in the figure. The interactive electric force from droplet 2, shown as $\boldsymbol{F}_{\text{inter}}$ in the figure, has a radial component and a tangential component, so that it is in a direction that does not necessarily align with the line connecting the two droplets. Because of the interactive electric force from droplet 2, the velocity $\boldsymbol{v}$ of droplet 1 is not in the vertical direction. The electrostatic force between charged droplets tend to make the droplets attract each other. This force is particularly strong when droplets are close to each other, thus to enhance collisions. The flow drag force on droplet 1 is in the opposite

direction with $\boldsymbol{v} - \boldsymbol{u}$.

If there is no external electric field but only with charge effect, Eq. (13) is reduced to

$$\boldsymbol{F}_{e2} = \frac{1}{r_2^2}(F_5 q_1{}^2 + F_6 q_1 q_2 + F_7 q_2{}^2)\hat{\boldsymbol{e}}_R \qquad (15)$$

To illustrate it, the comparison between the electrostatic forces derived by the inverse-square law and conductor model without electric field (i.e., Eq. 15) are shown in Fig. 3, where the electric force between droplets with opposite-sign charges (dashed lines) and with same-sign charges (solid lines) varies with distance. When $R \gg r_1, r_2$, we have $F_5, F_7 \to 0$, $F_6 \to r_2^2/R^2$, and it is also shown that two models are basically identical in remote distance. But when the spheres approach closely, the conductor interaction (blue lines) changes to strong attraction, because of electrostatic induction. The interaction is always attraction at small distance, regardless of the sign of charges.

[Figure]

**FIG. 2.** A schematic diagram of all the forces acting on the two droplets, as well as the velocities of the droplets. The electric field $\boldsymbol{E_0}$ is vertically downward, and electric charges $q_1 > 0$, $q_2 < 0$. Note that the electrostatic force $\boldsymbol{F}_{e1}$, $\boldsymbol{F}_{e2}$ include two parts: the electric force from the other droplet ($\boldsymbol{F}_{inter}$ in the figure), and the force from the external electric field (shown as $q_1\boldsymbol{E_0}$ and $q_2\boldsymbol{E_0}$ In the figure).

**5. Is it right that appear in the Eq. (12)?**

Response: Yes it is right. We should have emphasized that $\boldsymbol{F}_{e1}$ and $\boldsymbol{F}_{e2}$ consist not only the electric force from the other droplet, but also the force from the external electric field. As mentioned in the response to reviewer's 4[th] comment, the order of two terms in Equation (11) is changed, so that it is easier to identify the force due to the external electric field and the force due to the charge in the other droplet. Because the two droplets can be considered as a system, the sum of the forces they experience independently ($\boldsymbol{F}_{e1} + \boldsymbol{F}_{e2}$) must be equal to the external electric force acting on the system

$E_0(q_1 + q_2)\hat{e}_z$. This relation is expressed in equation (12). If we have already known $F_{e2}$, then $F_{e1}$ is derived immediately from Eq. (12). In line 140 of the original manuscript, we have made some changes in the writing of the manuscript to explain this.

**6. Line 153: actually, you integrate the system of 12 (or 8) equations**

Response: Thanks for the reviewer's careful reading. We have added this information to the manuscript based on reviewer's comment.

**7. Line 187: Eq. (13) is the exponential distribution and not the gamma distribution.**

Response: Yes Eq. (13) (which is Eq. 16 in the revised manuscript) is the exponential distribution. Thanks to the reviewer for pointing this out. We have made the correction in the manuscript. We should mention that Eq. (14) and (15) are gamma distributions. Please refer to the response to the 8th comment for detailed discussions on how to obtain Eq. (15).

**8. How did you obtain Eq. (15) from Eq. (14)?**

Response:

Thanks for this question. We now have added more information to this part, so that it is easier to understand the equations for size distribution. Basically, definitions of the size distribution is used for the derivation. Recall that $n(m)$ is the droplet number concentration per unit mass interval, and $M(m)$ is the mass concentration per unit mass interval.

The distribution of droplet number concentration $n(m)$ can also be written as $n(r)$, or $n(\ln r)$. We know that the definition of $n(m)$ is: $n(m) = dN/dm$, where $dm$ is the mass interval, and $dN$ is the droplet number concentration in that mass interval. $n(r) = dN/dr$ represents the droplet number concentration per unit size interval. $n(\ln r) = dN/d \ln r$ represents droplet number concentration per unit interval of logarithmic size. Similarly, the distribution of droplet mass concentration $M(m)$ can be written as $M(r)$, and $M(\ln r)$. These functions are related together.
$M(\ln r)$ and $M(r)$ are related through:
$$M(\ln r) = dM/d \ln r = r \cdot dM/dr = r \cdot M(r)$$
While $M(r)$ can be related with $M(m)$ through:
$$M(m) = \frac{dM}{dm} = \frac{1}{4\pi r^2} \frac{dM}{dr} = \frac{M(r)}{4\pi r^2}$$
With $m = 4\pi r^3 \rho/3$, and assuming that $\bar{m} = 4\pi \bar{r}^3 \rho/3$, where $\bar{r}$ is the mean radius, we can obtain $M(\ln r)$ from $M(m)$ ,

$$M(\ln r) = 3L \frac{r^6}{\bar{r}^6} exp\left(-\frac{r^3}{\bar{r}^3}\right)$$

In the revised manuscript, we added a new equation for $n(\ln r)$, because $n(\ln r)$ is also plotted and discussed in the Results section.

$$n(\ln r) = L \frac{9r^3}{4\pi \bar{r}^6} exp\left(-\frac{r^3}{\bar{r}^3}\right)$$

We feel that the derivations above is not very concise and does not look very straightforward. Therefore, in the revised manuscript, we choose a different way to present the initial size distribution function. Lines 187-194 in the original manuscript are now replaced by the following paragraph:

The initial droplet size distribution used in this study is derived based on an exponential function in Bott (1998),

$$n(m) = \frac{L}{\bar{m}^2} \exp\left(-\frac{m}{\bar{m}}\right) \tag{16}$$

where $n(m)$ is the distribution of droplet number concentration over droplet mass, $L$ is the liquid water content, and $\bar{m}$ is the mean mass of droplets. This function is used to derive $n(lnr)$, which is the distribution of droplet number concentration over droplet radius. With the definitions of $n(m)$ and $n(lnr)$, and $m = 4\pi r^3 \rho/3$, where $\rho$ is droplet density, we can derive $n(lnr)$ as

$$n(\ln r) = \frac{dN}{d\ln r} = r\frac{dN}{dr} = r\frac{dN}{dm}4\pi\rho r^2 = 4\pi\rho r^3 n(m) \tag{17}$$

By substituting Eq. (16) into Eq. (17), and assuming that $\bar{m} = 4\pi\bar{r}^3\rho/3$, where $\bar{r}$ is the mean radius, we have

$$n(\ln r) = L\frac{9r^3}{4\pi\bar{r}^6}exp\left(-\frac{r^3}{\bar{r}^3}\right) \tag{18}$$

**9. The authors should check the correctness of equations (13-15).**

Response: Thanks for the suggestion. We have checked the equations to make sure they are correct. Please also refer to the response to the 8[th] comment for information on the initial size distribution.

**10. Lines 195-200. To add information about number concentration, liquid water content and charge content for all initial drop spectra.**

Response: The initial liquid water content is set to be $L$=1 g m[-3], for all simulations. This is a typical value in warm clouds. The initial averaged droplet radius $\bar{r}$ is set to be 15 μm, 9 μm and 6.5 μm, where $\bar{r} =$ 15 μm case represents the clean conditions (less aerosol), and 6.5 μm represents polluted conditions (more aerosol). These settings give an initial droplet number concentration of 70, 325, and 850 cm[-3], respectively. The charge content is set as in the following table. The number concentration and charge content for all initial drop size distribution are shown in table 2 in the revised manuscript.

**Table** 2. Total number concentration, charge content for the initial droplet size distribution

| mean radius $\bar{r}$ (μm) | total number concentration (cm[-3]) | total positive charge concentration $Q^+$ (e cm[-3]) | total negative charge concentration $Q^-$ (e cm[-3]) |
|---|---|---|---|
| 15 | 70.6 | +9384 | -9384 |
| 9 | 324.8 | +15638 | -15638 |
| 6.5 | 850.5 | +21634 | -21634 |

Note that the initial droplet number concentration is distributed into different size bins and different charge bins. The size distribution is based on functions described in Equations (13)-(15). The charge distribution is now based on a Gaussian distribution in the revised manuscript, instead of the method described in lines 200-202 in the original manuscript. Ratios shown in lines 200-202 in the original manuscript is an approximation of 2:3:4:3:2, but it is arbitrarily chosen to mimic a normal distribution, and also to satisfy electric neutrality $\bar{q} = 0$. In the revised manuscript, we use a Gaussian distribution to describe droplet distribution over the charge bins.

Lines 199-202 have been revised and it reads as follows in the revised manuscript:

To simulate an early stage of the warm-cloud precipitation, we need to distribute the droplets in each size bin to different charge bins, so that these droplets have different charges. Since there is little data on this, we assume a Gaussian distribution,

$$N(q) = \frac{N_0}{\sqrt{2\pi}\sigma} \exp\left(-\frac{q^2}{2\sigma^2}\right) \qquad (19)$$

where $N_0$ is the number concentration in the size bin, and $\sigma$ is the standard deviation of the Gaussian distribution in that size bin. $N(q)$ represents the number concentration of droplets with charge $q$. This distribution satisfies electric neutrality $\bar{q} = 0$ . For different size bin, droplet number concentration $N_0$ is different. We purposely set the standard deviation $\sigma$ to be different for different size bins. For a larger size, the charge amount is larger, based on $\overline{|q|}$ = 1.31 $r^2$ (q in unit of elementary charge and r in µm) as stated in the Introduction. Therefore, we set larger standard deviation $\sigma$ for the larger size bins. With this setting of droplet charge, the total amount of charge in each case is shown in Table 1. The $\bar{r}$ = 15, 9, and 6.5 µm cases have an initial charge concentration of 9438, 15638, and 21634 e cm$^{-3}$, respectively, for both positive charge and negative charge.

**11. Line 216: Please add the figure showing collision efficiency between cloud droplets (1-20 µm in radii), the same as in fig. 5. It is all the more important because you obtained the maximal effect for cloud droplets.**

Response: As suggested, we plot a new figure below, to show the collision efficiencies for smaller collectors ($r_1$ = 5, 10, 15, 20 and 25 µm) when the droplet pairs have no charge. X-axis denotes the ratio of radius r$_2$/r$_1$. As will be seen in the response to reviewer's 12$^{th}$ comment, the 10 µm and 20 µm lines will be shown together with the results for charged droplets (new Fig. 6). Therefore, we think this figure is not necessary in the manuscript.

[Figure]

**12. Figure 6: Please, add illustrations for different collectors (say 15 ïAₗ m, and 10 ïAₗ m in radii) and comment them.**

Response: Thank you very much for the suggestion. As suggested, we have shown the different collectors $r_1$= 10, 20, 30 and 40 μm in the new figure below. Fig. 7 now describes the collision efficiency for the 30 and 40 μm collectors (precipitating droples). Fig. 8 now describes the collision efficiency for the 10 and 20 μm collectors (cloud droplets). Therefore, section 5.1 has been substantially revised. Most part of it has been rewritten. It is clear that electrostatic effects are significant for small droplets. We show the rewritten section 5.1 here:

**5.1   Collision efficiency**

[revised manuscript text omitted]

**13. Section 5.2: Please show temporal changes of drop concentration and charge content and comment on them. How fast the charges of opposite signs compensate each other?**

Response: Thanks for the reviewer's suggestion. The evolution of droplet concentration and charge content are shown in the below. These figures are also added to the manuscript as new Fig. 10, 12, and 15.

From Fig. 10 ($\bar{r}$ =15 μm), it is evident that droplet concentrations in the 4 different electric conditions decrease from about 70 cm$^{-3}$ to less than 5 cm$^{-3}$, and the evolution is nearly not affected by the electric conditions. The electrostatic effect is therefore negligible in this case.

From Fig. 12 ($\bar{r}$ =9 μm), we can see the evolution is distinctly affected by the 4 different electric conditions. Electric charges and fields play an important role in converting smaller droplets to larger droplets, and decreasing the droplet number concentration.

From Figure 15 ($\bar{r}$ =6.5 μm), droplet concentration is strongly affected by the 4 different electric conditions. Results show that the electric field would remarkably trigger the collision-coalescence process for the small droplets.

Comparing the upper and lower panels of each figure, it is evident that the charges of opposite signs compensate each other as fast as the decrease of number concentration (except for the uncharged case). The phases of charge neutralizations are the same as changes of drop concentration. In all the three figures, more than 90% charges of opposite signs are neutralized during the evolution.

[Figure]

**FIG. 10.** Temporal changes of droplet total number concentration and total charge content for $\bar{r}$ =15 μm.

[Figure]

[Figure]

**FIG. 12.** Temporal changes of droplet total number concentration and total charge content for $\bar{r} = 9$ μm

[Figure]

**FIG. 15.** Temporal changes of droplet total number concentration and total charge content for $\bar{r} = 6.5$ μm.

**14. Line 288: "The relative terminal velocity term also contributes to the collection kernel, and the electric field can affect terminal velocity of small charged droplets significantly." – Please, cover this issue in more detail in the article.**

Response: Thanks for the reviewer's suggestion. We have improved the writing of this part
Lines 287-295 "The electric enhancement of…" have been revised to:

[revised manuscript text omitted]

As for the electric effect on droplet spectrum evolution, few researches have been conducted. Khain et al. (2004), focused on weather modification, showed that droplet electric charge could enhance precipitation. They considered interaction of droplet pair by image charge, and use Stokes Flow to calculate hydrodynamic interaction. The charge limit is set up to the air-breakdown limit. It is found that a small fraction of extremely charged particles could trigger the collision process, and thus accelerate raindrop formation or fog elimination significantly.

As for the electrostatic effect on the evolution of droplet size distributon and the cloud system, few researches have been conducted. Focusing on weather modification, Khain et al. (2004) showed that a small fraction of highly charged particles could trigger the collision process, and thus accelerate raindrop formation in warm clouds or lead to fog elimination significantly. In their study, the electrostatic force between the droplet pair is represented by an approximate formula. The charge limit is set to the air-breakdown limit. The Stokes Flow is adopted to represent the hydrodynamic interaction, for deriving the trajectories of a pair of droplets. Harrison et. al. (2015) calculated droplet collision efficiencies affected by electric charges in warm clouds. When simulating the evolution of droplet size distribution in their study, the enhanced collision efficiencies are not used. Instead, the collection cross sections are multiplied by a factor of no more than 120% to approximately represent the electric enhancement of collision efficiency. The roles of electric charges and fields on precipitation acceleration still needs to be studied.

Previous studies about Albrecht (1989) effect show that increase of aerosol number decreases cloud droplet size, and thus extending cloud lifetime and suppress precipitation. But with the existence of electric charge, the Albrecht effect might be partially weakened. As mentioned above, Schlamp et al. (1976) had already shown that smaller droplets are more sensitive to electric effect. So, the coupling of electric effect and Albrecht effect needs to be considered.

The increased aerosol loading by anthropogenic activities can lead to an increase in cloud droplet number concentration, a reduction in droplet size, and therefore an increase in cloud albedo (Twomey 1974). This imposes a cooling effect on climate. It is further recognized that the aerosol-induced reduction in droplet size can slow down droplet collision-coalescence and cause precipitation suppression. This leads to increased cloud fraction and liquid water amount, and imposes an additional cooling effect on climate (Albrecht 1989). As the charging of cloud droplets can enhance droplet collision-coalescence, especially for small droplets, it is worth studying to what extent the electrostatic effect can mitigate the aerosol effect on the evolution of droplet size distribution and precipitation formation.

This study investigates the effect of electric charges and fields on droplet collision efficiency and the evolution of droplet spectrum. the droplet size distribution. The amount of charges is set as the condition in warm clouds, and the electric fields are set as the early stage of thunderstorms. The more accurate method for calculating the electric forces is adopted (Davis, 1964). Correction of flow field for large Reynolds numbers are also considered. Section 2 describes the theory of

droplet collision-coalescence and stochastic collection equation.  ection 3 presents the equations of motion for charged droplets in an electric field. The method for obtaining the terminal velocities and collision efficiencies for charged droplets are also presented. Section 4 describes the model setup for solving the stochastic collection equation. Different initial droplet size  distributions and different electric conditions are considered. Section 5 shows the numerical results of electrostatic effects on collision efficiency, and on  the evolution of droplet size distribution. We intend to find out to what extent the electric charges and fields as in the observed atmospheric conditions can accelerate warm rain process, and how sensitive these electrostatic effects are to aerosol-induced changes of droplet sizes.

**2  Stochastic Collection Equation**

The evolution of droplet size  distribution due to collision-coalescence is described by the stochastic collection equation (SCE), which was first proposed by Telford (1955), and is shown as (Lamb and Verlinde, 2011, p.442)

$$\frac{\partial n(m,t)}{\partial t} = \int_0^{m/2} K(m_x, m - m_x) \cdot n(m_x)n(m - m_x)\mathrm{d}m_x - n(m)\int_0^{\infty} K(m_x, m) \cdot n(m_x)\mathrm{d}m_x \qquad (1)$$

where $n(m,t)$ is the  distribution of droplet number concentration over droplet mass at time t, and $K$ is the collection kernel between the two classes of droplets. For example, the collection kernel $K(m_x, m - m_x)$ describes the rate that droplets of mass $m_x$ collected by $m - m_x$ and form new droplets of mass $m$.  The first term on the right side of Eq. (1), the first term describes formation of droplets of mass $m$ through the collision of smaller droplets, and the second term means the loss of droplets of mass $m$ through collision with other droplets.

The collection kernel between droplets with mass $m_1$ and mass $m_2$ can be written as

$$K(m_1, m_2) = |V_1 - V_2| \cdot \pi(r_1 + r_2)^2 \cdot E(m_1, m_2) \cdot \varepsilon(m_1, m_2) \qquad (2)$$

~~where $V_1$ and $V_2$ are the terminal velocity of each droplet, $r_1$ and $r_2$ are droplet radius, $E$ is the collision efficiency, and $\varepsilon$ is the coalescence efficiency. Obviously, $|V_1 - V_2| \cdot \pi(r_1 + r_2)^2$ means the geometric volume swept in unit time, but not all the small droplets in this volume could collide with large droplet. Because the flow induced by the larger droplet may drive some smaller droplets to flow past it. Thus, collision efficiency $E$ is introduced as a proportion factor for true collision and it is much smaller than 1.0 when the droplet sizes are significantly different.~~

[revised manuscript text omitted]

$$\bm{F}_{e2} = E_0 q_2 \cos\theta\hat{\bm{e}}_{\bm{R}} + E_0 q_2 \sin\theta\hat{\bm{e}}_{\bm{\theta}} +$$

$$\{r_2{}^2 E_0^2 (F_1 \cos^2\theta + F_2 \sin^2\theta) + E_0\cos\theta(F_3 q_1 + F_4 q_2) + \frac{1}{r_2^2}(F_5 q_1{}^2 + F_6 q_1 q_2 + F_7 q_2{}^2)\}\hat{\bm{e}}_{\bm{R}}$$

$$+ \{r_2{}^2 E_0^2 F_8 \sin2\theta + E_0\sin\theta(F_9 q_1 + F_{10} q_2)\}\hat{\bm{e}}_{\bm{\theta}}$$

(13)

where $\hat{\bm{e}}_{\bm{R}}$ is the radial unit vector, and $\hat{\bm{e}}_{\bm{\theta}}$ is tangential unit vector, $\bm{E_0}$ is the eternal electric field, $q_1$ and $q_2$ are the charges of droplet 1 and droplet 2 respectively, . $F_1 \dots F_{10}$ are a series of complicated functions of geometric parameters $(r_1, r_2, R$; Davis 1964).

The electric force directly from the external field is shown as the two terms in the first line of Eq. (13), and can be simply written as $\bm{E_0}q_2$ if combining the two terms. Line 2 and line 3 in Eq. (13) represent the interactive force from droplet 1 in the radial direction and tangential direction, respectively. Note that the third term in line 2 represent the interactive force from droplet 1 if there is no external electric field. Except for this term, all the other terms in lines 2 and 3 are the interactive forces from droplet 1 due the induction from the external field.

Similarly, the resultant electric force $\bm{F}_{e1}$ acting on droplet 1 includes both the force directly from the external field and the interactive force from droplet 2. The sum of the electric forces on the two droplets, $\bm{F}_{e1} + \bm{F}_{e2}$, must equal to the external electric force acting on the system, which can be expressed as $\bm{E_0}(q_1 + q_2)$, because the two droplets can be considered as a system. Then, the electric force on droplet 1 could be derived immediately  as

$$\bm{F}_{e1} = E_0(q_1 + q_2)\hat{\bm{e}}_{\bm{z}} - \bm{F}_{e2} \qquad (14)$$

[revised manuscript text omitted]

~~With the collision efficiencies of droplets with different radii and charges in different strength of electric field all computed, it is found that the electric effect is sensitive to droplet radii. Results are only discussed for $r_1 = 30$ μm and shown in Fig. 6. Totally 6 combinations of electric conditions are selected to be shown here, and the details are summarized in table 1. The droplet pair is set to have no charge, same-sign charge, or opposite charge. The electric field is set to be 0 or 400 V m$^{-1}$. Compared to the no-charge pair (curve 1), the same-sign charge without electric field (curve 2) slightly decreases collision efficiency, because of the repulsive force. The results of both positively charged pair and negatively charged pair are identical, since there is no electric field. In a downward electric field, the collision efficiency of the two situations is changed. For a positively charged pair (curve 3), the collision efficiency is very close to the no-charge pair, which implies that enhancement of electric field offset the repulsive 
[revised manuscript text omitted]

[Figure]

FIG. 4 FIG. 5. Initial spectrum mass distribution shown in 2-dimensional grids of bins The initial spectrum droplet mass distributed over the size and charge bins. Different Colours stand for water mass content in the bins (in unit of g m$^{-3}$). (a) Uncharged spectrum mass distribution droplets (b) charged spectrum mass distribution droplets.

[Figure]

**FIG. 5 FIG. 6.** Collision efficiency between for droplets with no electric charge or field. Colour Lines are results computed in this study. Different lines show represent different large collector radius $r_1$, from 30 to 305 μm. X-axis denotes the smaller collected droplet radius $r_2$. Scatter points are collision efficiencies from previous experimental studies.

[Figure]

**FIG. 7.** Collision efficiency for droplets with electric charge and field. The radius of the collector droplet $r_1$ is: (a) 30.0 μm, (b) 40.0 μm. X-axis denotes the collected droplet radius $r_2$. The two droplets carry electric charges proportional to $r^2$. The lines for droplet pairs with no charge (line 1 in Fig. 7a and 7b) are the same as the 30 μm and 40 μm lines in Fig. 6.

[Figure]

**FIG. 8.** Collision efficiency for droplets with electric charge and field. The radius of the collector droplet $r_1$ is: (a) 10.0 μm, (b) 20.0 μm. The other characteristics of the droplet pairs are similar to those in Fig. 7.

[Figure]

**FIG. 7 FIG. 9.** The evolution of  droplet size distribution with initial $\bar{r}$ = 15 μm. These panels show different stages of the evolution from top to bottom. The left  column shows the size distribution of droplet mass concentration, and the right  column shows the size distribution of droplet number concentration, on logarithmic scales. In each panel, comparisons are made for 4 different electric conditions. Blue lines denote the uncharged cloud . Red lines denote charged cloud without electric field. Green and purple lines denote charged cloud with a field of 200 V cm[-1] and 400 V cm[-1], respectively. Dotted lines show the initial  size distribution.

[Figure]

[Figure]

**FIG. 10.** Temporal changes of droplet total number concentration and total charge content for $\bar{r} = 15$ μm

[Figure]

FIG. 8. **FIG. 11.** The evolution of  droplet size distribution with initial $\bar{r} = 9$ μm.

[Figure]

[Figure]

**FIG. 12.** Temporal changes of droplet total number concentration and total charge content for $\bar{r} = 9 \ \mu m$

[Figure]

FIG. 9. FIG. 13. Comparison of evolutions of 2-dimensional spectrum evolution distribution of droplet mass concentration with different electric conditions at 60 min (initial $\bar{r} = 9$ μm).

[Figure]

FIG. 10. **FIG. 14.** The evolution of  droplet size distribution with initial $\bar{r}$ = 6.5 μm.

[Figure]

**FIG. 15.** Temporal changes of droplet total number concentration and total charge content for $\bar{r} = 6.5$ μm.

[Figure]

FIG. 11. FIG. 16. Terminal velocities of droplets in an external electric field 400 V cm⁻¹. Different lines denote different droplet charge conditions. It is significant that terminal velocity of negatively charged droplets smaller than 5 μm would turn upwards, which leads to the discontinuity of the lower curve in the figure.

---

## Referee Report (RR1)

Comments to the manuscript with ID "acp-2019-1140"

General comments:

A summery of previous works and the key point of this study have been improved substantially. However, the language and presentation can still be improved. A moderate revision is needed before it can be considered for publication.

Specific comments:
1. L.42: Why "cloud top is positively charged and the cloud base is negatively charged"?
2. L.67: "researches"→ "studies". Please also check the use of "researches" in other places of the manuscript.
3. L.71: "investigate"→"investigated"
4. L.75: How do they affect climate system?
5. L.83: Stokes flow
6. L.85-88: check the tense of verbs.
7. L.88: The sentence "The roles of electric charges and fields on precipitation acceleration still needs to be studied" is very abrupt. Could you please elaborate more on this and its link to previous sentences?
8. L.98: "The more accurate method for "→"A more …method to". Please also check the application of "the" in the last paragraph of the introduction.
9. L.126: collide and coalesce …
10. L.175: to be "solved"
11. L.182: kinetic or dynamic viscosity? Also, \nu is the common symbol representing kinetic viscosity and \mu for the dynamic viscosity. Please check these symbols carefully in the manuscript to improve the readability of the manuscript.
12. L. 188: "right side"→right hand side.
13. L. 195: Please check the expression of $N\_Re$. Also, please specify that it is the particle Reynolds number.
14. L. 199-200: "U" is a scalar here and velocity is a vector.
15. L.204: Could you please estimate the $N\_Re$ using the typical values of droplet velocity, diameter, and the kinetic viscosity of air? Can it really be as large as 5000? More importantly, why Eq.6 applies to cases with $N\_Re<5000$?
16. Is Eq. 10 an empirical Eq?
17. Please clarify the assumptions employed in Eq.11.
18. $F\_e$ in Eq. 12 is a vector, which looks like a scalar in the sentence below Eq.12. Please check the font style carefully throughout the manuscript.
19. For the sentences above Eq.12, can you simply say "apply Coulomb's law to point particles" or something like that to improve the readability?
20. L.242: How does "the line connecting the centers" affect the electric force?
21. Are your $q\_1$ and $q\_2$ in Eq. 12 and 13 the same? Why do you define them twice?

22. L.251: "the first two terms of Eq. (13)". Please check the wording of explaining each item in Eqs all through the manuscript. You may explicitly write down those terms and explain the physical meaning of them.

23. L.291: "the convenience" or you may say "To save the computational power,"

24. L.293: I don't understand this "the two droplets can either collide or not depending on the initial horizontal distance"
25. L.402: Did you mean "net" effect?
26. L.409: "negatively-charged"
27. L. 407-409: I don't understand how the overall negatively charged collector and the collected droplet can attract each other. Could the author elaborate more on this?
28. L. 417: "and can be used to represent cloud droplets"→ which are the typical size of cloud droplets.

---

## Author Response (AR2)

Dear editor,

Thank you very much for the editorial handling of our manuscript. We have revised our manuscript according to the referee's report and the editor's comments. The grammar errors are corrected and the language is revised based on the suggestions of the referee and the editor.

Attached are the point-by-point response to the referee's comments, and a marked-up version of the revised manuscript. In the response to referee's comments, the sentences in blue are our responses, and the sentences in red are revised texts in the manuscript. In the marked-up version of the revised manuscript, words and sentences in blue are contents that are added, and those in red with strikethrough are contents that are deleted.

We did not prepare a point-by-point response to the editor's comments. All the changes based on the editor's suggestions are marked up in the revised manuscript. There are several things we should point out here:
1. In the last version of the manuscript, L.83: "air-breakdown limit" means the electrical breakdown of air due to an extremely strong electric field. It is changed to "electrical breakdown limit of air" now.
2. The editor commented that, in the last version of the manuscript, the discussion from L.42 to the end of this paragraph is not very clear. We have added a sentence before this part: "Previous studies also evaluated the charge amount per droplet in warm clouds." Basically, Harrison et al. (2017) used a measured charge density in cloudy air and an assumed droplet number concentration to derive the charge amount per droplet.
3. Table 1 has been deleted and the information has been added to the caption of Fig.7 as suggested.

Thank you very much and we look forward to your reply.

Sincerely,

Shian Guo, Huiwen Xue

**Response to Referee #1**

General comments: A summary of previous works and the key point of this study have been improved substantially. However, the language and presentation can still be improved. A moderate revision is needed before it can be considered for publication.

Response: We thank the reviewer for the careful reading and detailed comments on the language of the paper. We have a point-by-point revision. The marked-up version of the revised paper is also listed below.

Specific comments:

1. L.42: Why "cloud top is positively charged and the cloud base is negatively charged"?

Response:

Thanks to the reviewer for pointing this out. We probably did not explain it clearly in L. 40-42 in the last version of the manuscript.

Consider an ideal model, a stratus cloud which expands wide enough horizontally. According to the theory of electromagnetics, the current density is $\vec{j} = \sigma\vec{E}$, where $\sigma$ is the conductivity, and $\vec{E}$ is the electric field (which is vertically downward in this model). The current density is constant in a column. Since cloudy air has a much lower electrical conductivity than cloud-free air, the strength of the electric field in clouds is much stronger than that in cloud-free air. On the horizontal boundary between air and the cloud, according to Gauss' law, the difference of the electric field $\vec{E}$ between the two sides of the boundary equals to $\hat{e}_z \cdot (\vec{E}|_{upper} - \vec{E}|_{lower}) = \frac{\rho_{surf}}{\varepsilon}$, where $\rho_{surf}$ is the surface density of electric charge on the boundary, and $\varepsilon$ is the dielectric constant. Consequently, the cloud top is positively charged and the cloud base is negatively charged.

This concise explanation is added before L.42 "Therefore":

For the given current density, the fair weather electric field is typically about 1 V cm$^{-1}$ in cloud-free air, but is usually much stronger inside stratus clouds because the cloudy air has a lower electrical conductivity than cloud-free air. At cloud top, the difference of the downward electric fields on the two sides of the cloud boundary leads to a certain amount of positive charge accumulated on the cloud boundary, according to Gauss' law. In the same way, a certain amount of negative charge is accumulated on the cloud boundary at cloud base.

2. L.67: "researches" → "studies". Please also check the use of "researches" in other places of the manuscript.

Response: corrected in L.67, 79 and 189.

3. L.71: "investigate" → "investigated"

Response: corrected

4. L.75: How do they affect climate system?

Response:

L.75: "Moreover, they proposed that solar influences may change the fair weather current and droplet collision process, a possible pathway for affecting the climate system." This sentence is replaced by the following elaboration:

Moreover, Solar influences (e.g. solar modulation of high-energy particles) can modulate atmospheric electrical parameters, such as current density in the atmosphere, and then influence the amount of electric charge on the cloud-air boundary. Since electric charges enhance the collection efficiency of small droplets, this solar modulation can further affect the lifetime and radiative properties of clouds globally (Harrison et al. 2015). Therefore, it is possible that solar modulation may have an indirect influence on climate.

5. L.83: Stokes flow

Response: corrected

6. L.85-88: check the tense of verbs.

Response:

L.85: are → were

L.87: needs → need

7. L.88: The sentence "The roles of electric charges and fields on precipitation acceleration still needs to be studied" is very abrupt. Could you please elaborate more on this and its link to previous sentences?

Response: it has been changed to (L.88):

This approximation can roughly show the enhancement of droplet collision and raindrop formation by charges in warm clouds. Further studies are still needed to evaluate the electrostatic effect more accurately and for various aerosol conditions that are typical in warm clouds.

8. L.98: "The more accurate method for" → "A more …method to". Please also check the application of "the" in the last paragraph of the introduction.

Response: corrected for L.98 and L.100

9. L.126: collide and coalesce …

Response: corrected.

L.126: not all the droplets in this volume will have collision-coalesce with the collector.

Now it reads:

not all the droplets in this volume will necessarily collide or coalesce with the collector.

10. L.175: to be "solved"

Response: corrected

11. L.182: kinetic or dynamic viscosity? Also, \nu is the common symbol representing kinetic viscosity and \mu for the dynamic viscosity. Please check these symbols carefully in the manuscript to improve the readability of the manuscript.

Response:

We thank the reviewer for pointing this out. The viscosity in our paper is dynamic viscosity. Now it is specified in L.182

and 184. The symbol $\eta$ in Eq.(4) is replaced by the symbol $\mu$.

12. L. 188: "right side" →right hand side.

Response: corrected

13. L. 195: Please check the expression of N_Re. Also, please specify that it is the particle Reynolds number.

Response: the expression of Reynolds number is right.

L. 195: "…the steam function depends on Reynolds number, $N_{Re} = \frac{2rv\rho}{\mu}$". It now reads,

"…the stream function depends on Reynolds number of this spherical particle, $N_{Re} = \frac{2rv\rho}{\mu}$"

14. L. 199-200: "U" is a scalar here and velocity is a vector.

Response:

We thank the reviewer for pointing out this mistake. "U" is a scalar indeed, so we have changed this sentence into:

L.199: "$U$ is the value of droplet velocity relative to the flow,…"

Besides, we have also found a similar mistake and corrected it:

L.218: "Note that $u_\theta$ is the velocity of the fluid at droplet surface,…"

$u_\theta$ here is not a velocity vector. It should be a tangential component. So, it is changed to:

"Note that $u_\theta$ is the tangential component of the velocity of the fluid,…"

15. L.204: Could you please estimate the N_Re using the typical values of droplet velocity, diameter, and the kinetic viscosity of air? Can it really be as large as 5000? More importantly, why Eq.6 applies to cases with N_Re<5000?

Response:

In L.222-225 in the last version of the manuscript, we showed Reynolds numbers for a couple of droplets. But there is a typo error on "r=16 μm". It should be "r=32 μm" instead. Another thing we should clarify is that the Reynolds number in this paragraph is computed by an empirical formula (Pruppacher and Klett, 1997, Chapter 10.3.5). However, in our study, Reynolds numbers are exactly derived by the definition $N_{Re} = \frac{2rv\rho}{\mu}$ when we compute the stream function and drag coefficient. Note that there is some difference between the results by the empirical formula and by definition. For consistency, the Reynolds numbers and drag coefficients listed in this paragraph are now derived by definition. We also decide to show another example here with r = 2 μm, because droplet radii of 2 ~ 1024 μm are considered in this study. Now this part is changed to:

For example, the terminal velocity of a droplet of 2 μm in radius is $4.92\times10^{-4}$ cm s$^{-1}$, with $N_{Re} = 1.23\times10^{-4}$ and $C = 1.00001$; the terminal velocity of a droplet of 32 μm in radius is 11.8 cm s$^{-1}$, with $N_{Re} = 0.47$ and $C = 1.07$; the terminal velocity of a droplet of 1024 μm in radius is 715 cm s$^{-1}$, with $N_{Re} = 915$ and $C = 18.0$.

Reynolds numbers of cloud droplets with radii of 2 ~ 1000 μm are typically in the range of $10^{-4}$ ~ $10^3$. (Very large rain drops can have larger Reynolds numbers, such as 5000.)

It is known that for small $N_{Re}$ ($N_{Re} \ll 1$, typically for cloud droplet radii smaller than 30 μm), the flow field is described by the Stokes flow. But this is no longer valid when $N_{Re}$ is large. Therefore, the expression of the flow field is better dependent on $N_{Re}$. In fact, previous studies have shown many different approximate equations of stream fields for a large range of $N_{Re}$. We choose this empirical stream function of Hamielec and Johnson (1962, 1963) because it is applicable to a relatively large $N_{Re}$. Note that the parameters in $A_1, ..., B_4$ in Eq. (6) are a series of functions of Reynolds number. This sentence is added before L.205 "Therefore", to remind readers that the range of Reynolds number in our study is relatively large, and so Eq. (6) is needed:

Cloud droplets with radii ranging from 2 μm to 1024 μm typically have a Reynolds number ranging from $10^{-4}$ to $10^3$.

16. Is Eq. 10 an empirical Eq?

Response:

Yes, it is an empirical equation of Beard (1976). This sentence:

"According to an empirical equation of Beard (1976)," is added before L.220: "the drag coefficient $C$ in Eq. (4) is a function of $N_{Re}$"

17. Please clarify the assumptions employed in Eq.11.

Response:

We have explained more about this assumption. The whole paragraph which starts from L.226 has been changed to:

For droplets with $r < 10 \, \mu m$, the assumption of no-slip boundary condition is no longer valid because droplet sizes are comparable with the mean free path of air molecules. Air cannot be considered as a continuous medium. The flow slips on the droplet surface. To take this effect into consideration, the drag coefficient should multiply another coefficient (Lamb and Verlinde 2011, p386)

165

18. F_e in Eq. 12 is a vector, which looks like a scalar in the sentence below Eq.12. Please check the font style carefully throughout the manuscript.

Response: corrected

170    19. For the sentences above Eq.12, can you simply say "apply Coulomb's law to point particles" or something like that to improve the readability?

Response:

The sentence of L.235 is replaced by "For two point particles, we apply Coulomb's law,"

175    20. L.242: How does "the line connecting the centers" affect the electric force?

Response:

"the line connecting the centers" is just used for specifying a certain direction. $\theta$ is the angle between this direction and the downward electric field. It affects the electric force because the electric field can cause electrostatic induction on the two droplets. For instance, let us assume that two droplets are both neutral with a certain distance R. They are induced by

180    the downward electric field and each become an electric dipole approximately. When the line connecting the centers is perpendicular to the electric field ($\theta=90°$), the dipole-dipole interaction is repulsive, according to electromagnetics. But for the parallel case ($\theta=0$), the dipole-dipole interaction is attractive.

21. Are your q_1 and q_2 in Eq. 12 and 13 the same? Why do you define them twice?

185    Response:

Yes, the definition for Eq.(13) is redundant. The sentence "$q_1$ and $q_2$ are the charges of droplet 1 and droplet 2 respectively," has been deleted.

22. L.251: "the first two terms of Eq. (13)". Please check the wording of explaining each item in Eqs all through the

190    manuscript. You may explicitly write down those terms and explain the physical meaning of them.

Response:

"the first two terms of Eq. (13)" is just the force purely from the electric field acting on droplet 2, and it is downward (if $q_2 > 0$). The sum of the two terms, $E_0 q_2 \cos\theta \, \hat{\boldsymbol{e}}_R + E_0 q_2 \sin\theta \, \hat{\boldsymbol{e}}_{\boldsymbol{\theta}}$, is exactly the vertical vector $\boldsymbol{E_0} q_2$. The rest of Eq.(13) is the force from droplet 1. We use the polar coordinate here because the line connecting the centers of the two droplets

is not parallel with x-y-z direction. So, the force from the electric field, $\boldsymbol{E_0}q_2$, is decomposed into the first two terms in Eq. (13), so that the representations of forces are consistent. Their physical meaning has already been discussed in this paragraph.

23. L.291: "the convenience" or you may say "To save the computational power,"
Response:
Thanks for the reviewer's advice and we have revised it.

24. L.293: I don't understand this "the two droplets can either collide or not depending on the initial horizontal distance"
Response:
It is a little confusing indeed. What we mean is "the initial horizontal distance determines whether two droplets collide or not." This sentence is redundant because the next sentence has already implied this. ("We vary the initial horizontal distance using the bisection method, until we find a threshold distance $r_c$ that makes the two droplets follow the grazing trajectories and just exactly collide") Section 2 also has already explained this. So, we decide that this sentence "Following the trajectories, the two droplets…" is deleted now.

25. L.402: Did you mean "net" effect?
Response:
Yes. We have corrected it.

26. L.409: "negatively-charged"
Response: corrected.

27. L. 407-409: I don't understand how the overall negatively charged collector and the collected droplet can attract each other. Could the author elaborate more on this?
Response:
We have elaborated more on this at the end of L.409:
In other words, we can approximately consider the collected droplet as a negative monopole (since it is very small), and consider the collector as a negative monopole plus a downward dipole that is induced by the electric field. When the two droplets are relativity far, the monopole-monopole interaction is dominant so that the force is repulsive. But when they get close, the monopole-dipole interaction gets dominant in certain circumstances, so that the force changes to attractive.

28. L. 417: "and can be used to represent cloud droplets" → which are the typical size of cloud droplets.
Response:

The typical size of droplets in non-precipitation clouds is on the order of 10 μm. But cloud droplet size can vary from 1 μm to 100 μm, and even to 1000 μm if we want to include the drizzle droplets.

We decide that this sentence "and can be used to represent cloud droplets" is redundant and it has been deleted.

230

235

**The enhancement of droplet collision by electric charges and atmospheric electric fields**

Shian Guo, Huiwen Xue

School of Physics, Peking University, Beijing, China

240    *Correspondence to*: Huiwen Xue (hxue@pku.edu.cn)

**Abstract.**

The effects of electric charges and fields on droplet collision-coalescence and the evolution of cloud droplet size distribution are studied numerically. Collision efficiencies for droplet pairs with radii from 2 to 1024 μm and charges from -32 $r^2$ to +32 $r^2$ (in unit of elementary charge, droplet radius $r$ in unit of μm) in different strengths of downward electric fields (0, 200 and 400 V cm⁻¹) are computed by solving the equations of motion for the droplets. It is seen that collision efficiency is increased by electric charges and fields, especially for pairs of small droplets. These can be considered as electrostatic effects.

The evolution of cloud droplet size distribution with the electrostatic effects is simulated using the stochastic collection equation. Results show that the electrostatic effect is not notable for clouds with the initial mean droplet radius $\bar{r}$ =15 μm or larger. For clouds with the initial $\bar{r}$ = 9 μm, the electric charge without field could evidently accelerate raindrop formation compared to the uncharged condition, and the existence of electric fields further accelerates it. For clouds with the initial $\bar{r}$ = 6.5 μm, it is difficult for gravitational collision to occur, and the electric field could significantly enhance the collision process. Results of this study indicate that electrostatic effects can accelerate raindrop formation in natural conditions, particularly for polluted clouds. It is seen that the aerosol effect on the suppression of raindrop formation is significant in polluted clouds, when comparing the three cases with $\bar{r}$ = 15, 9, and 6.5 μm. However, the electrostatic effects can accelerate raindrop formation in polluted clouds and mitigate the aerosol effect to some extent.

**1 Introduction**

Clouds are usually electrified (Pruppacher and Klett 1997). For thunderstorms, several theories of electrification have been proposed in the past decades. The proposed theories assume that the electrification involves the collision of graupel or hailstones with ice crystals or supercooled cloud droplets, based on radar observational result that the onset of strong electrification follows the formation of graupel or hailstones within the cloud (Wallace and Hobbs, 2006). However, the exact conditions and mechanisms are still under debate. One charging process could be due to the thermoelectric effect between the relatively warm, rimed graupel or hailstones and the relatively cold ice crystals or supercooled cloud droplets. Another charging process could be due to the polarization of particles by the downward atmospheric electric field. The thunderstorm electrification can increase the electric fields to several thousand V cm⁻¹, while the magnitude of electric fields in fair weather air is only about 1 V cm⁻¹ (Pruppacher and Klett 1997). Droplet charges can reach $|q| \approx 42r^2$ in unit of elementary charge

in thunderstorms, with the droplet radius $r$ in unit of μm according to observations (Takahashi, 1973). For  cumulus clouds, previous studies show smaller charge amounts.

270    Liquid  stratiform clouds do not have such strong charge generation as in the thunderstorms. But charging of droplets can indeed occur at the upper and lower cloud boundaries as the fair weather current passes through the clouds (Harrison et al. 2015, Baumgaertner et al. 2014). The global fair weather current and the electric field are in the downward direction. Given the electric potential of 250 kV for the ionosphere, the exact value of fair weather current density over a location depends on the electric resistance of the atmospheric column, but its typical value is about $2 \times 10^{-12}$ A m$^{-2}$

275    (Baumgaertner et al. 2014). For the given current density, the fair weather electric field is typically about 1 V cm$^{-1}$ in  cloud-free air, but is usually much stronger inside stratus clouds because the cloudy air has a lower electrical conductivity than  cloud-free air.  At cloud top, the difference of the downward electric fields on the two sides of the cloud boundary leads to a certain amount of positive charge accumulated on the cloud boundary, according to Gauss' law. In the same way, a certain amount of negative charge is accumulated on the cloud boundary at cloud

280    base. Therefore, the cloud top is positively charged and the cloud base is negatively charged. Previous studies also evaluated the charge amount per droplet in warm clouds. Based on the in situ measurements of charge density in liquid  stratiform cloud, and assuming that the cloud has a droplet number concentration on the order of 100 cm$^{-3}$, it is estimated that the mean charge per droplet is +5e (ranging from +1e to +8e) at cloud top, and -6e (ranging from -1e to -16e) at cloud base ( Harrison et al. 2015). According to  Takahashi (1973) and Khain (1997), the mean absolute charge of

285    droplets in warm clouds is around $|q| \approx 6.6\ r^{1.3}$ (e, μm). For a droplet with radii of 10 μm, it is about 131 e.

        In general, charging of droplets can lead to the following effects on warm cloud microphysics.  First, for charged haze droplets, the charges can lower the saturation vapor pressure over the droplets and enhance  cloud droplet activation (Harrison and Carslaw, 2003, Harrison et al. 2015).  Second, the electrostatic induction effect between charged droplets can lead to strong attraction at very small distance (Davis, 1964) and higher collision-coalescence efficiencies (Beard

290    et al. 2002).  However, Harrison et al. (2015) showed that charging is more likely to affect collision processes than activation, for small droplets.

        The electrostatic induction effect can be explained by regarding the charged cloud droplets as spherical conductors. The electrostatic force between two conductors is different from the well-known Coulomb force between two point charges. When the distance between a pair of charged droplets approaches infinity, the electrostatic force converges to Coulomb force between

295    two point charges. But when the distance  between the surfaces of two droplets is small (e.g. much smaller than their radii), their interaction shows extremely strong attraction. Even when the pair of droplets carry the same sign of charges, the electrostatic force can still change from repulsion to attraction at small distances. Although there is no explicit analytical expression  for the electrostatic interaction between two charged droplets, a model with high accuracy has been developed (Davis 1964) for the interaction of charged droplets in a uniform electric field. Many different approximate

300    methods are also proposed for the convenience of computation in cloud physics (e.g. Khain et al., 2004).

Based on this induction concept, electrostatic effects on droplet collision-coalescence process have been studied in the past decades. A few experiments show that electric charges and fields can enhance coalescence between droplets. Beard et. al. (2002) conducted experiments in cloud chambers and showed that even minimal electric charge can significantly increase the probability of coalescence when the two droplets collide. Eow et. al., (2001) examined several different electrostatic effects in water-in-oil emulsion, indicating that electric fields can enhance coalescence by several mechanisms such as film drainage.

 Model simulations indicate that charges and fields can increase droplet collision efficiencies because of the electrostatic forces. Schlamp et al. (1976) used the model of Davis (1964) to study the effect of electric charges and atmospheric electric fields on collision efficiencies. They demonstrated that the collision efficiencies between small droplets (about 1~10 μm) are enhanced by an order of magnitude in thunderstorms, while collision between large droplets is hardly affected. Harrison et al. (2015) investigated the electrostatic effects in weakly electrified liquid clouds rather than thunderstorms. They calculated collision efficiencies between droplets with radii less than 20 μm and charge less than 50 e using the equations of motion in Klimin (1994). Their results indicate that electric charges at the upper and lower boundaries of warm  stratiform clouds are sufficient to enhance collisions, and the enhancement is especially significant for small droplets. Moreover, Solar influences (e.g. solar modulation of high-energy particles) can modulate atmospheric electrical parameters, such as current density in the atmosphere, and can influence the amount of electric charge on the cloud-air boundary. Since electric charges enhance the collection efficiency of small droplets, this solar modulation can further affect the lifetime and radiative properties of clouds globally (Harrison et al. 2015). Therefore, it is possible that solar modulation may have an indirect influence on climate.  Tinsley (2006) and Zhou (2009) also studied the collision efficiencies between charged droplets and aerosol particles in weakly electrified clouds, by treating the particles as conducting spheres. They considered many aerosol effects such as thermophoretic forces, diffusophoretic forces and Brownian diffusion.

As for the electrostatic effect on the evolution of droplet size distributon and the cloud system, few  studies have been conducted. Focusing on weather modification, Khain et al. (2004) showed that a small fraction of highly charged particles  can trigger the collision process, and thus accelerate raindrop formation in warm clouds or  fog dissipation significantly. In their study, the electrostatic force between the droplet pair is represented by an approximate formula. The charge limit is set to the electrical breakdown limit of air.  Stokes  flow is adopted to represent the hydrodynamic interaction for deriving the trajectories of  droplet pairs. Harrison et. al. (2015) calculated droplet collision efficiencies affected by electric charges in warm clouds. When simulating the evolution of droplet size distribution in their study, the enhanced collision efficiencies were not used. Instead, the collection cross sections  were multiplied by a factor of no more than 120% to approximately represent the electric enhancement of collision efficiency.  This approximation can roughly show the enhancement of droplet collision and raindrop formation by charges in warm clouds. Further studies are still needed to evaluate the electrostatic effect more accurately and for various aerosol conditions that are typical in warm clouds.

[revised manuscript text omitted]
. Cloud droplets with radii ranging from 2 μm to 1024 μm typically have a Reynolds number ranging from $10^{-4}$ to $10^3$. Therefore, it is needed to construct a stream function that applies to a wide range of $N_{Re}$. This work adopts a stream function that is a linear combination of $\psi_h$ and Stokes stream function $\psi_s$ (Pinsky and Khain, 2000)

$$\psi = \frac{N_{Re} \psi_h + N_{Re}^{-1} \psi_s}{N_{Re} + N_{Re}^{-1}} \tag{7}$$

which converges to stokes flow when $N_{Re} \to 0$. Then the induced flow field $\boldsymbol{u}$ is derived,

$$\boldsymbol{u} = -\frac{1}{\tilde{R}^2 \sin \theta_0} \frac{\partial \psi}{\partial \theta_0} \hat{\boldsymbol{e}}_R + \frac{1}{\tilde{R} \sin \theta_0} \frac{\partial \psi}{\partial \tilde{R}} \hat{\boldsymbol{e}}_\theta = u_R \hat{\boldsymbol{e}}_R + u_\theta \hat{\boldsymbol{e}}_\theta \tag{8}$$

where $\hat{e}_R$ and $\hat{e}_\theta$ are unit vectors in the polar coordinate $(R, \theta_0)$. It can also be expressed in the Cartesian coordinate (x, z)

$$\boldsymbol{u} = (u_R \cos\varphi - u_\theta \sin\varphi)\hat{e}_z + (u_R \sin\varphi + u_\theta \cos\varphi)\hat{e}_x \tag{9}$$

where the direction of $\hat{e}_z$ is vertically down, the same as gravitation. $\varphi$ is the angle between $\hat{e}_z$ and the droplet velocity $\boldsymbol{v}$.

Both Stokes and Hamielec stream functions satisfy the no-slip boundary condition, i.e., the fluid velocity on the surface of the droplet is equal to the velocity of the droplet. Hamielec stream function is no-slip because those functions $A_1, \ldots, B_4$ in Eq. (6) satisfy $A_1 + 2A_2 + 3A_3 + 4A_4 = 1$ and $B_1 + 2B_2 + 3B_3 + 4B_4 = 0$, as long as the droplet is considered as a rigid sphere (Hamielec, 1963). These relations ensure that $u_\theta = -U \sin\theta_0$ at the surface of the droplet, which means the no-slip boundary condition. (Note that $u_\theta$ is the tangential component of the velocity of the fluid  and $U \sin\theta_0$ is the tangential velocity of the droplet surface.)

According to an empirical equation of Beard (1976), the drag coefficient $C$ in Eq. (4) is a function of $N_{Re}$,

$$C = 1 + \exp(a_0 + a_1 X + a_2 X^2) \tag{10}$$

where $X = \ln(N_{Re})$, and fitting constants $a_0, a_1, a_2$ are from  Table 1 of Beard (1976). The drag coefficient increases with Reynolds number.  For example, the terminal velocity of a droplet of 2 μm in radius is $4.92\times10^{-4}$ cm s⁻¹, with $N_{Re}$ =$1.23\times10^{-4}$ and $C$ =1.00001; the terminal velocity of a droplet of 32 μm in radius is 11.8 cm s⁻¹, with $N_{Re}$ =0.47 and $C$ =1.07; the terminal velocity of a droplet of 1024 μm in radius is 715 cm s⁻¹, with $N_{Re}$ =915 and $C$ =18.0.

For droplets with $r < 10\ \mu m$, the assumption of no-slip boundary condition is no longer valid because droplet sizes are comparable with the mean free path of air molecules. Air cannot be considered as a continuous medium. The flow slips on the droplet surface.  To take this effect into consideration, 
[revised manuscript text omitted]
 negatively-charged collected droplet below experiences an attractive force. In other words, we can
665 approximately consider the collected droplet as a negative monopole (since it is very small), and consider the collector as a negative monopole plus a downward dipole that is induced by the electric field. When the two droplets are relativity far, the monopole-monopole interaction is dominant so that the force is repulsive. But when the two droplets get close, the monopole-dipole interaction gets dominant in certain circumstances, so that the force changes to attractive.

As for a pair with opposite-sign charges, line 5 in Fig. 7a shows that the collision efficiency is enhanced by the
670 electrostatic effect even when there is no electric field. The collision efficiency is nearly an order of magnitude higher with $r_2 < 5$ μm. Line 6 in Fig. 7a shows that, with an electric field of 400 V cm$^{-1}$, the electrostatic effect for the pairs with opposite-sign charges is even stronger. There is also an interesting feature in Fig. 7a: as the collector and collected droplets have similar sizes, collision efficiency is high for the pairs with opposite-sign charges. This is quite different from the other four lines, where collision efficiencies are very low for droplet pairs with similar sizes.

675 Figure 8 shows the collision efficiencies for droplet pairs with charge and field, and with smaller collectors. The collector droplet has a radius of 10 μm (Fig. 8a) and 20 μm (Fig. 8b) here. Collision efficiencies for these smaller collectors are much smaller than 1 when there is no charge (line 1 in Figs. 8a and 8b), which is already well known in cloud physics community. However, the electrostatic effects are so strong that the collision efficiencies could be significantly changed for these collectors. For the collector droplet with a radius of 10 μm (Fig. 8a), the
680 positively-charged pair has a very small collision efficiency that is out of the scale in the figure, due to the dominating effect of the repulsive force as discussed above. For the positively-charged pair under a downward electric field, the collision efficiencies  have a similar order of magnitude as the pair with no charge. For the negatively-charged pair under the downward electric field, and for the pairs with opposite-sign charges, the electrostatic effects  are very strong. The

negatively-charged pair even  exhibits collision efficiency increases of as much as

[revised manuscript text omitted]
  represent collision efficiencies from previous experimental studies. The numbers next to these points represent the collector drop radius.

940

[Figure]

**FIG. 7.** Collision efficiency for droplets with electric charge and field. The radius of the collector droplet $r_1$ is: (a) 30.0 μm, (b) 40.0 μm. X-axis denotes the collected droplet radius $r_2$. The two droplets carry electric charges proportional to $r^2$. The lines for droplet pairs with no charge (line 1 in Fig. 7a and 7b) are the same as the 30 μm and 40 μm lines in Fig. 6. The settings of electric charge and field is: (1) no charge and no field. (2) $q_1 = +32r_1{}^2$, $q_2 = +32r_2{}^2$, with no field. (3) $q_1 = +32r_1{}^2$, $q_2 = +32r_2{}^2$, with a downward electric field of 400 V cm⁻¹. (4) $q_1 = -32r_1{}^2$, $q_2 = -32r_2{}^2$, with a downward

945

electric field of 400 V cm $^{-1}$. (5) $q_1 = +32r_1{}^2$, $q_2 = -32r_2{}^2$, with no field. (6) $q_1 = +32r_1{}^2$, $q_2 = -32r_2{}^2$, with a downward electric field of 400 V cm $^{-1}$.

950

[Figure]

**FIG. 8.** Collision efficiency for droplets with electric charge and field. The radius of the collector droplet $r_1$ is: (a) 10.0 μm, (b) 20.0 μm. The other characteristics of the droplet pairs are similar to those in Fig. 7.

[Figure]

**FIG. 9.** The evolution of droplet size distribution with initial $\bar{r} = 15$ μm. These panels show different stages of the evolution from top to bottom. The left column shows the size distribution of droplet mass concentration, and the right column shows the size distribution of droplet number concentration, on logarithmic scales. In each panel, comparisons are made for 4 different electric conditions. Blue lines denote the uncharged cloud. Red lines denote charged cloud without electric field. Green and purple lines denote charged cloud with a field of 200 V cm⁻¹ and 400 V cm⁻¹, respectively. Dotted lines show the initial size distribution.

[Figure]

**FIG. 10.** Temporal changes of droplet total number concentration and total charge content for $\bar{r} = 15$ μm

[Figure]

**FIG. 11.** The evolution of the droplet size distribution with initial $\bar{r} = 9$ μm.

[Figure]

[Figure]

975    **FIG. 12.** Temporal changes of droplet total number concentration and total charge content for $\bar{r} = 9$ μm

[Figure]

**FIG. 13.** Comparison of evolutions of 2-dimensional distribution of droplet mass concentration with different electric conditions at 60 min (initial $\bar{r}$ = 9 μm).

[Figure]

**FIG. 14.** The evolution of the droplet size distribution with initial $\bar{r}$ = 6.5 μm.

[Figure]

985

**FIG. 15.** Temporal changes of droplet total number concentration and total charge content for $\bar{r} = 6.5$ μm.

[Figure]

**FIG. 16.** Terminal velocities of droplets in an external electric field 400 V cm$^{-1}$. Different lines denote different droplet charge conditions. It is significant that terminal velocity of negatively charged droplets smaller than 5 μm would turn upwards, which leads to the discontinuity of the lower curve in the figure.